# How Does the Pretraining Distribution Shape In-Context Learning? A Fundamental Trade-Off

**Waïss Azizian** [1] [*]    **Ali Hasan** [2]

## Abstract

The factors driving the performance of in-context learning (ICL) in large language models (LLMs) remain poorly understood despite ICL's surprising effectiveness, enabling models to adapt to new tasks from only a handful of examples. To clarify and improve these capabilities, we characterize how the statistical properties of the pretraining distribution (e.g., tail behavior, coverage) shape ICL. We develop a theoretical framework that encompasses generalization and task selection and show how distributional properties govern sample efficiency, task retrieval, and robustness. To this end, we generalize existing concentration results to heavy-tailed priors and dependent sequences, better reflecting the structure of LLM pretraining data. Our framework reveals a fundamental design trade-off: heavy-tailed pretraining distributions facilitate robust task selection under distribution shifts but are detrimental to generalization, especially in low-data regimes. We then empirically evaluate our predictions by studying how ICL performance varies with the pretraining distribution on challenging tasks such as stochastic differential equations and stochastic processes with memory. Together, these findings suggest that controlling key statistical properties of the pretraining distribution is essential for building ICL-capable and reliable LLMs.

## 1. Introduction

In-context learning (ICL) is the phenomenon whereby a model generalizes to a new task from a handful of examples provided in the input context without any model weight updates. This emergent behavior has been observed across models in multiple domains, including in language (Brown et al., 2020), vision (Radford et al., 2021), and reinforcement learning (Moeini et al., 2025). ICL is a particularly appealing feature in domains where data for a specific task is scarce such as robotics (Ahn et al., 2023b), healthcare (Singhal et al., 2023), or chemistry (Stokes et al., 2020).

Despite the importance of this property, the conditions under which ICL emerges are still poorly understood. Several lines of works have emerged to address this question. The algorithmic view focuses on studying which learning algorithms over the context can be implemented by transformer and thereby perform ICL (Garg et al., 2022; Akyürek et al., 2023). Others have suggested modeling ICL as Bayesian inference (Xie et al., 2021; Lin & Lee, 2024; Zhang et al., 2025b; Jeon et al., 2024). Empirical works have sought to design controlled settings in which ICL can be carefully studied, and these works highlight how sensitive to pretraining choices ICL is (Chan et al., 2022; Raventós et al., 2023), indicating that distributional aspects of pretraining play a central role. A crucial line of work also seeks to assess ICL performance on numerical tasks through out-of-distribution robustness of ICL (Wang et al., 2025b; Kwon et al., 2025; Goddard et al., 2025).

Taken together, these perspectives suggest that ICL is shaped not only by architecture and learning dynamics, but also by what the model sees during pretraining. What remains missing is a principled way to translate pretraining distributional properties into test-time ICL behavior. Indeed, several aspects remain particularly underexplored: (i) heavy-tailed task priors capturing long-tail effects that have been implicated empirically in ICL (Chan et al., 2022; Singh et al., 2023), (ii) non-i.i.d. and dependent context structure (e.g., long-range dependencies) beyond standard i.i.d. / Markov assumptions (Alabdulmohsin et al., 2024), and (iii) how these distributional properties govern ICL behavior under test-time shifts, a key motivation for ICL (Wang et al., 2025b; Kwon et al., 2025; Goddard et al., 2025).

---

[*]Work done during an internship at Morgan Stanley Machine Learning Research. [1]Univ. Grenoble Alpes, CNRS, Inria, Grenoble INP, LJK, 38000 Grenoble, France [2]Machine Learning Research, Morgan Stanley, New York, USA. Correspondence to: Waïss Azizian <waiss.azizian@univ-grenoble-alpes.fr>, Ali Hasan <ali.hasan@morganstanley.com>.

*Proceedings of the 43$^{rd}$ International Conference on Machine Learning*, Seoul, South Korea. PMLR 306, 2026. Copyright 2026 by the author(s).

We thus develop a study of ICL with a focus on the influence of the pretraining distribution. We decompose ICL performance into two components: *task selection* (identifying the right task from the context) and *generalization* (performing well on tasks and sequences unseen during training) and focus on the following questions:

> *How does the pre-training distribution shape ICL performance on new tasks? How does it affect generalization and task selection errors?*

Our contributions are as follows:

- **Theoretical framework under heavy tails and dependence.** We develop a general theoretical framework for ICL that focuses on the role of pretraining *distributional* properties, handling both the task selection error and the ICL generalization error. We cover *heavy-tailed* priors and *dependent* sequences, providing conditions that better reflect pretraining data used for LLMs and highlighting the role of these key distributional properties.

- **A trade-off in pretraining distribution design.** Our theory reveals a fundamental trade-off in pretraining distribution design for ICL: heavier tails improve task selection at test time, especially under distribution shift, but they degrade generalization when training data is limited.

- **Empirical validation on numerical tasks.** We validate the predictions of our framework on challenging numerical tasks—including stochastic differential equations and processes with memory, assessing ICL via robustness to new tasks and distribution shift.

Together, our results suggest that controlling key statistical properties of the pretraining distribution is essential for building ICL-capable and reliable transformer models.

## 2. Related Work

A growing number of works aim to understand in-context learning (ICL) from complementary perspectives. We focus here on those most relevant to our setting, see Table 1 for a summary comparison and App. A for additional discussion.

**Bayesian Perspectives.** One of the most influential perspectives on ICL is Bayesian: the pretraining distribution is viewed as a prior over tasks, and ICL corresponds to Bayesian inference on a new task given the context (Xie et al., 2021). Lin & Lee (2024) adopted this perspective to analyze ICL on linear regression tasks, characterizing which task is effectively retrieved at inference time, while Wang et al. (2025b) refined this analysis for out-of-distribution tasks. However, these frameworks are restricted to Gaussian linear task families and therefore do not isolate how the shape of the pretraining distribution impacts ICL. Jeon et al. (2024) provide an information-theoretic perspective

on task retrieval for ICL but do not model the distribution of tasks. Park et al. (2025b); Wurgaft et al. (2025) study the competition and transition between in-weight learning and ICL, and obtain scaling laws for the emergence of ICL in transformers, while Nguyen & Reddy (2025) investigate this transition via a differential kinetics model. Though offering valuable insights into ICL mechanisms, these works do not provide guarantees that connect key properties of the pretraining distribution to ICL performance. Zhang et al. (2025b) introduced a broad Bayesian framework, with results on both task identification and generalization for Markovian sequences. It is the closest to our work in scope, but it focuses on light-tailed priors and does not characterize how properties of the pretraining distribution affect ICL and the resulting trade-offs, in contrast to our work.

**Conditions for the emergence of ICL.** Raventós et al. (2023) studied how training choices affect the emergence of ICL on linear regression tasks, and in particular how these factors influence the number of pretraining tasks required for strong in-context performance. Chan et al. (2022) empirically studied properties of the pretraining distribution that promote ICL on international character recognition, which was extended by Singh et al. (2023), who showed that ICL can be transient.Together, these works suggest that heavier-tailed pretraining distributions can improve ICL performance only up to a point, beyond which performance degrades. Our work provides a complementary theoretical framework that predicts and explains this trade-off via explicit task-identification and generalization guarantees.

**Generalization in ICL.** Li et al. (2023) derive generalization guarantees via stability of the transformer architecture, but in a setting where the task distribution is fixed and finite and is the same at pretraining and test time. Zhang et al. (2025b); Zekri et al. (2024) provide generalization bounds for ICL on Markov chains, but do not model a pretraining task distribution whose shape can vary and affect performance. Lotfi et al. (2024) derive generalization bounds for transformers on arbitrary sequences, yet their notion of generalization only covers new tokens as continuations of existing sequences and not new sequences, which does not correspond to the ICL setting. In contrast, our generalization guarantee handles a general pretraining task distribution, allowing heavy-tailed priors and dependent contexts,thereby quantifying how properties of the pretraining distribution control ICL generalization.

**Numerical Tasks.** A related line of work uses controlled numerical tasks (e.g., linear regression or dynamical systems) as probes to study ICL in simplified transformer models and in pretrained LLMs. Zhang et al. (2024); Wu et al. (2024) analyze ICL on linear regression with single-layer or linear-attention models, characterizing the ICL error of the trained model.More recently, Lu et al. (2025) obtain a

precise characterization of the emergence of ICL for linear regression in a linear-attention model, including certain out-of-distribution regimes. Chan et al. (2025) study a simple Bayesian predictor model to understand the different modes of in-weight learning and ICL while Liu et al. (2024) investigate ICL of pretrained large language models on Markov processes and report power-law scaling behavior. Finally Wang et al. (2025b); Kwon et al. (2025); Goddard et al. (2025) all show that ICL can extrapolate to out-of-distribution tasks only to a limited extent. These works primarily focus on architectural mechanisms and scaling behavior in specific probe settings, whereas our focus is complementary: we quantify how properties of the pretraining task distribution shape ICL.

**General Concentration.** The pioneering work of Yu (1994) provides concentration inequalities for dependent processes under mixing-type conditions, opening up a fruitful line of research; see, e.g., Kontorovich & Ramanan (2008); Mohri & Rostamizadeh (2008; 2010); Maurer (2023); Abélès et al. (2025), and for related coupling techniques Chazottes et al. (2007); Paulin (2015). While these frameworks can handle general dependent sequences, they typically rely on sub-Gaussian-type assumptions that are incompatible with the heavy-tailed task priors considered here. Another line of work derives concentration for sums of stationary dependent sequences (Wu, 2005; 2011; Liu et al., 2013). In contrast, our ICL analysis requires concentration for more general function classes and for non-stationary dependence along the context; we are not aware of existing results that simultaneously accommodate these requirements together with heavy tails. For heavy-tailed concentration in the independent case, the recent frameworks of Bakhshizadeh et al. (2023); Li & Liu (2024b); Li et al. (2024), provide concentration for non-linear functions of i.i.d. heavy-tailed random variables. Our framework extends this line by handling non-linear functions of dependent heavy-tailed sequences, which underpins our generalization guarantees.

## 3. Theoretical framework

To connect ICL behaviour at test time to the properties of the pretraining distribution, we model the training data as a mixture of tasks, in line with existing works on ICL (Garg et al., 2022; Lin & Lee, 2024; Jeon et al., 2024; Zhang et al., 2025b; Wang et al., 2025b). In § 3.1, we present the ICL setting. The error of ICL is decomposed into two components: the generalization error of the trained model (§ 3.2), and the ability of the model to identify the correct task given some in-context examples (§ 3.3).

### 3.1. In-context learning setting

We model the training data as a mixture of tasks, with each task defining its own distribution. Formally, denote by

$\Theta \subset \mathbb{R}^d$ the space of tasks $\theta$ and by $\pi(\theta)$ the density of the pretraining task distribution. Given a task $\theta$, the data is generated according to a task-specific distribution with density $p(\cdot \mid \theta)$ The training data is then generated by first sampling a task $\theta$ from the task distribution $\pi$, and then sampling data points $(x_t)_{t \geq 1}$ according to

$$x_{t+1} \sim p_{t+1}(\cdot \mid x_{1:t}, \theta), \quad \text{where } x_{1:t} = (x_1, \ldots, x_t).$$

We first illustrate the setting with several examples.

**Example 3.1** (Classification). Several ICL benchmarks for LLMs such as Bertsch et al. (2025); Zou et al. (2025); Li et al. (2025b) are built on classification tasks. Each task $\theta$ represents a small subset of classes from a larger classification problem and the data sequence $x_1, \ldots, x_t$ is a sequence of inputs and labels from these classes. The challenge is therefore to both identify the classes and learn to classify them from the in-context examples.

**Example 3.2** (Linear Regression). Introduced by Garg et al. (2022), the regression setting is a popular testbed for ICL. Each task $\theta \in \mathbb{R}^d$ defines a linear model $y = \theta^T q + \epsilon$ where $\epsilon$ is some noise. The data sequence $x_1, \ldots, x_{2t}$ is a sequence of input-output pairs $q_1, y_1, \ldots, q_t, y_t$ generated according to the linear model defined by $\theta$.

**Example 3.3** (Ornstein-Uhlenbeck process). More generally, we can consider the setting where each task $\theta$ defines a stochastic process $x_{t+1} \sim p_{t+1}(\cdot \mid x_{1:t}, \theta)$. We will consider later the specific case of the Ornstein-Uhlenbeck process: each task $\theta = (\tau, \mu)$ defines a mean-reverting stochastic process with mean $\mu$ and reversion speed $\tau$:

$$dX_t = \tau(\mu - X_t)dt + \sigma dW_t, \tag{1}$$

where $W_t$ is a standard Brownian motion and $\sigma$ is the volatility parameter. The data sequence $x_1, \ldots, x_t$ is then a discretization of the stochastic process defined by $\theta$. In this setting, the learning objective is predict the next sample given the previous ones, implicitly requiring the identification of the parameters $\theta$.

We present next examples of prior distributions $\pi$ over tasks that will illustrate our theoretical results.

**Example 3.4** (Priors in 1D). For simplicity, consider the case where tasks are one-dimensional, i.e., $\Theta \subset \mathbb{R}$. Student's $t$-distributions with $\nu > 1$ degrees of freedom are an example of heavy-tailed priors with polynomially decaying tails: for large $\theta$, $\pi(\theta) \propto 1/|\theta|^{\nu+1}$. $\pi(\theta)$ thus decays more slowly as $\nu$ decreases, leading to heavier tails. By convention, Student's $t$-distribution with $\nu = \infty$ degrees of freedom corresponds to the Gaussian distribution, whose tails decay exponentially. Generalized Normal distributions, by contrast, still retain exponentially decaying tails but allow to control the rate of decay: for a scale parameter $\alpha > 0$ and a shape parameter $\beta \geq 1$, it has density $\pi(\theta) \propto \exp(-|\theta/\alpha|^\beta)$. $\pi(\theta)$ thus decays more slowly as $\beta$ decreases, leading to heavier tails.

**Table 1:** Positioning relative to selected prior work on in-context learning. ✓/✗ indicate whether an explicit guarantee is provided. *Task model:* ARBITRARY(arbitrary task structure), FINITE(finite reuse), LIN-G(Gaussian linear), —(no task parameter). *Task selection:* FINAL ($t = T$) or AVG (avg. over $t = 1{:}T$). *Dependent seq.:* dependence class (e.g., IID/Markov/Ergodic/Arbitrary). Other columns indicate whether generalization bounds, heavy-tailed priors, and explicit dependence on the pretraining distribution are covered.

| | Task model | Task selection | Gen. bounds | Dependent seq. | Heavy-tailed prior | Influence of pretrain |
|---|---|---|---|---|---|---|
| **Ours** | ARBITRARY | FINAL | ✓ | ARBITRARY | ✓ | ✓ |
| Li et al. (2023) | FINITE | ✗ | ✓ | MARKOV | ✗ | ✗ |
| Zhang et al. (2025b, §5) | — | ✗ | ✓ | ERGODIC | ✗ | ✗ |
| Zhang et al. (2025b, §6) | FINITE | AVG | ✗ | ARBITRARY | ✗ | ✗ |
| Zekri et al. (2024) | — | ✗ | ✓ | MARKOV | ✗ | ✗ |
| Lin & Lee (2024); Wang et al. (2025b) | LIN-G | FINAL | ✗ | IID | ✗ | ✗ |
| Chan et al. (2022); Singh et al. (2023) | — | ✗ | ✗ | IID | ✓ | ✓ |

Given a dataset of tasks $\theta_1, \ldots, \theta_N$ and associated samples $x_{1:T}^{(1)}, \ldots, x_{1:T}^{(N)}$, a model $f$ is trained by minimizing the next-sample prediction loss

$$\widehat{L}(f, (\theta_n, x_{1:T}^n)_{n \leq N}) = \frac{1}{NT} \sum_{n=1}^{N} \sum_{t=1}^{T} \ell_t(f(x_{1:t-1}^n), x_t^n), \quad (2)$$

where $\ell_t$ is a per-sample loss which depend on $t$ to encompass regression and classification tasks. Note that the model is trained to predict the next sample $x_t$ given the previous samples $x_{1:t-1}$, without any explicit supervision on the task $\theta$. This is why ICL is referred to as an emergent ability of large models (Wei et al., 2022). When evaluating the performance of ICL on new tasks, two kinds of error come into play: (i) the *generalization error* of the trained model $\hat{f}$ obtained by minimizing (2) on a training dataset, and(ii) the ability of the model to identify the correct task given some in-context examples, which we refer to as *task selection*.

### 3.2. Generalization error

The first key statistical question for ICL is its generalization error. We therefore study the generalization error of the trained model $\hat{f}$ obtained by minimizing (2) on a training dataset. We consider a dataset consisting of $N$ tasks $\theta_1, \ldots, \theta_N$ sampled independently from the prior $\pi$, and for each task $\theta_n$, a sequence of $T$ samples $x_{1:T}^n$ generated according to the task-specific distribution $\mathrm{p}_T(\cdot \mid \theta_n)$: for $n \leq N$, for $t < T$, $x_{t+1}^{(n)} \sim \mathrm{p}_{t+1}(\cdot \mid x_{1:t}^{(n)}, \theta_n)$.

Motivated by LLMs pre-trained on large corpora of text, we consider here the challenging setting where the data sequence $(x_t)_{t \leq T}$ within each task is dependent and possibly non-Markovian and the task distribution $\pi$ can be heavy-tailed. To the best of our knowledge, existing concentration for dependent sequences do not cover this case. We thus develop our own framework: we encompass non-independent and identically distributed (i.i.d.) and non-Markovian data sequences through a weak dependence assumption in Wasserstein distance, and we handle heavy-

tailed task distributions by taking inspiration from the recent framework of Li & Liu (2024a); Li et al. (2024). The resulting framework is therefore quite general and can be of independent interest beyond ICL, see App. D.

We present here a simplified version of our assumptions, where we focus on the few key quantities that are relevant in our study: how dependent the data sequence is and how heavy-tailed the prior $\pi$ is, quantified through the maximal moment of $\pi$ that exists[1] We refer to App. D.3 for the complete version of the assumptions. We consider $\mathcal{F}$ a class of models $f : \cup_t (\mathbb{R}^k)^t \to \mathbb{R}^k$ and $\ell_t : \mathbb{R}^k \times \mathbb{R}^k \to \mathbb{R}_+$ a per-sample loss function that can depend on time $t$.

**Assumption 1** (Moment condition). There is $q \geq 2$ an integer such that $\mathbb{E}_{\theta \sim \pi}[\|\theta\|^q] < \infty$.

This assumption quantifies how "heavy-tailed" the prior $\pi$ is: the smaller the exponent $q$, the heavier the tail of $\pi$. This exponent $q$ will play a key role in the generalization error of ICL. We now introduce the assumptions on the dependence structure of the data sequence, where $W_1(\cdot, \cdot)$ is the 1-Wasserstein distance.

**Assumption 2** (Dependence structure).

(i) **Weak dependence.** There is $B_T > 0$ such that, for any $s < t \leq T$, any $\theta \in \Theta$, any $x_{1:s}, x_s'$,

$$W_1(\mathrm{p}_t(dx_t \mid x_{1:s}, \theta), \mathrm{p}_t(dx_t' \mid x_{1:(s-1)}, x_s', \theta)) \leq B_T(1 + \|\theta\|).$$

(ii) **Influence of the task.** There is $A_T > 0$ such that, any $t \leq T$, any $\theta, \theta' \in \Theta$,

$$W_1(\mathrm{p}_t(dx_t \mid \theta), \mathrm{p}_t(dx_t' \mid \theta')) \leq A_T \|\theta - \theta'\|.$$

The first assumption quantifies how dependent the data sequence: the higher $B_T$, the more influence past samples have on future samples; while second assumption quantifies how much the task influences the data distribution. In the extreme case of an i.i.d. sequence, both $A_T$ and $B_T$ are bounded w.r.t. $T$, which might not be the case in general.

---

[1]We focus here on priors with polynomially decaying tails, such as the Student-$t$ family since it is the most representative. A similar result could be established for subexponential tails.

Finally, we require some regularity on the model class $\mathcal{F}$.

**Assumption 3** (Model regularity)**.**

(i) **Average Lipschitzness.** There is an $L_T > 0$ such that, for any $f \in \mathcal{F}$, any $x_{1:T}, x'_t$,

$$\frac{1}{T} \sum_{s=1}^{T} \|f(x_{1:s-1}) - f(x_{1:t-1}, x'_t, x_{t+1:s-1})\| \leq L_T \|x_t - x'_t\|,$$

(ii) **Usual conditions.** The losses $\ell_t$ are 1-Lipschitz; the class of models $\mathcal{F}$ is bounded and uniformly Lipschitz with respect to some metric and $x_t$ conditioned on $x_{1:t-1}, \theta$ is uniformly sub-Gaussian.

In addition to assumptions common in learning theory, Asm. 3-(i) requires that the model class $\mathcal{F}$ be "on average" Lipschitz with respect to changes in the input sequence. Thus $L_T$ quantifies how much the model $f$ uses the older examples in context: for transformer with context length at least $T$, $L_T$ is typically bounded. If, on the contrary, the context length is kept constant and smaller than $T$, as in Zekri et al. (2024), $L_T$ can decay as $1/T$.

Given $\hat{f}$ the trained model obtained using the empirical distribution $(\theta_n, x^n_{1:T})_{n \leq N}$ the central quantity that our main result bounds is the generalization error:

$$\widehat{\text{gen}} := \mathbb{E}_{\theta \sim \pi} \left[ \mathbb{E}_{x_{1:T} \sim p_T(\cdot|\theta)} \left[ \frac{1}{T} \sum_{t=1}^{T} \ell_t(\hat{f}(x_{1:t-1}), x_t) \right] \right]$$
$$- \widehat{L}(\hat{f}, (\theta_n, x^n_{1:T})_{n \leq N}).$$

**Theorem 1.** *Under Asms. 1–3, for any $\delta \in (0, e^{-2})$, with probability at least $1 - \delta$, it holds:*

(i) *If $\delta \geq Ne^{-q}$, then*
$$\widehat{\text{gen}} \leq \mathcal{O}\left( \frac{(\log 1/\delta)^{3/2} L_T \sqrt{T}}{\sqrt{N}} \left( 1 + A_T \sqrt{T} + B_T T \right) \right),$$

(ii) *If $\delta < Ne^{-q}$, then*
$$\widehat{\text{gen}} \leq \mathcal{O}\left( \frac{L_T \sqrt{T}}{\delta^{1/q} \sqrt{N}} \left( 1 + A_T \sqrt{T} + B_T T \right) \right),$$

*where the terms in $\mathcal{O}(\cdot)$ depend polynomially on $q$, $\log N$, the scale of $\pi$ and the size of $\mathcal{F}$.*

Like standard concentration inequalities for sums of independent heavy-tailed random variables, Thm. 1 provides two regimes. For small deviations, i.e., $\delta$ not arbitrarily small, the generalization error behaves like in a sub-exponential setting. However, for large deviations, i.e., $\delta$ very small, the behaviour of the generalization error worsens and depends on the moment $q$ of the prior $\pi$. The generalization thus depends critically on the moment $q$ of the prior $\pi$: the smaller the moment $q$, the heavier the tail of the prior $\pi$ and the worse the generalization error. Indeed, the smaller $q$, the higher the threshold $Ne^{-q}$ separating the two regimes, leading to worse generalization for small $\delta$. Moreover, the dependence on $\delta$ in the second regime also worsens as $q$ decreases.

This can be observed on the examples of priors presented in Ex. 3.4 and in particular Student's $t$-distributions: with $\nu$ degrees of freedom, the maximal moment is $q = \lceil \nu - 1 \rceil$ so that smaller values of $\nu$, i.e., heavier tails, lead to smaller values of $q$ and worse generalization.

This bound also highlights how much larger the number of tasks must be compared to the number of in-context examples to ensure good generalization: in general, one needs $N$ to be at least much larger than $T$ to ensure a small generalization error. This is in line with our experiments and previous empirical studies. Raventós et al. (2023) shows that to obtain optimal ICL performance with a context length of 16 or 64 in linear regression, one needs thousands of tasks[2]. Moreover, if the data sequence is highly dependent, i.e., $A_T$ and $B_T$ are large, the requirement on the number of tasks $N$ for ICL to generalize well also increases. This will be demonstrated in § 4.3.

Note that the guarantee of Thm. 1 can be translated into a bound on out-of-distribution generalization, see App. D.6. Also, in App. D.7, we extend this result to the case where tasks are repeated in the training dataset, which is often the case in practice and improves the dependence on $N$.

> ***Takeaway #1:*** *Heavier-tailed priors and stronger temporal dependences increase the number of tasks required for reliable ICL generalization.*

**Connection to the IWL-ICL transition.** Although studying the IWL-ICL transition is not our primary focus, Thm. 1 provides some insight into it. Consider $N$ tasks $\theta_1, \ldots, \theta_N$ sampled from $\pi$. The IWL regime corresponds to the Bayes-optimal predictor with respect to the discrete empirical distribution $\hat{\pi}_N = \frac{1}{N} \sum_{i=1}^{N} \delta_{\theta_i}$, while the ICL regime corresponds to the Bayes-optimal predictor with respect to the true distribution $\pi$ (Raventós et al., 2023). A trained model will be closer to the ICL regime when it minimizes not only the training error but also the population loss. Our generalization guarantee can thus be seen as a guarantee on when the model enters the ICL regime: when the generalization error (Thm. 1) is small, the trained model is close to the Bayes-optimal predictor for $\pi$, and therefore operates in the ICL regime rather than the IWL regime.

Though it ensures good performance on out-of-sample tasks from $\pi$, this does not guarantee good performance under distribution shift: understanding how the Bayes-optimal predictor itself performs on tasks far from the bulk of $\pi$ is the subject of the next section.

---

[2]Note however that Park et al. (2025b); Wurgaft et al. (2025) highlight that these numbers significantly vary across settings.

### 3.3. Task selection

Our second main result concerns the ability of a trained model to perform ICL and in particular to retrieve the correct task given some input sequence. For this, we adopt the Bayesian point of view: if $f$ is arbitrarily powerful, trained to optimality and generalization is negligible, $f$ learns the *Bayesian optimal predictor*. If we denote the posterior $\widehat{p}_t(\theta \mid x_{1:t-1})$ the posterior distribution over tasks given the input sequence $x_{1:t-1}$, the Bayesian optimal predictor $f(x_{1:t-1})$ is given by

$$\arg \min_{\hat{x}_t} \mathbb{E}_{\theta \sim \widehat{p}_t(\cdot \mid x_{1:t-1})} \left[ \mathbb{E}_{x_t \sim p_t(\cdot \mid x_{1:t-1}, \theta)} [\ell_t(\hat{x}_t, x_t)] \right] . \quad (3)$$

Assuming that transformer models learn this Bayesian is a common assumption in the literature on ICL (Lin & Lee, 2024; Zekri et al., 2024; Jeon et al., 2024; Zhang et al., 2025b; Wang et al., 2025b) supported by empirical evidence (Chan et al., 2022; Raventós et al., 2023; Wurgaft et al., 2025; Nguyen & Reddy, 2025; Park et al., 2025b).

For a model to perform ICL given in-context examples $x_{1:t-1}$ generated from a task $\theta^*$, it is therefore necessary that the posterior $\widehat{p}_t(\theta \mid x_{1:t-1})$ concentrates around the true task $\theta^*$ as the number of in-context examples $t$ increases. Our main result provides a quantitative guarantee of this concentration and highlights the role of the properties of the pretraining distribution $\pi$.

For this, we require some mild assumptions on the data generation process only; they do not restrict the prior $\pi$. Since our focus is on the influence of the prior $\pi$ on task identification, in the main text we mainly focus on assumptions and quantities that involve $\pi$, and defer the detailed assumptions to App. E. We will therefore use the notation poly$(x)$ to denote a quantity that is polynomial in $x$ with coefficients independent of the prior $\pi$ and the number of samples $T$.

**Assumption 4** (Data generation, informal). Let $\theta^* \in \Theta$ be the true task. We assume:

(i) **Tail control.** Sequences $x_{1:t}$ generated under the true task $\theta^*$ have controlled tails, at most poly$(T)$ on typical tail events and $\pi$ admits a second moment.

(ii) **Moment bound.** For any $T \geq 1$, $\mathbb{E}_{X \sim p_T(\cdot \mid \theta^*)} \left[ \log^2 \left( \sup_{\theta \in \Theta} \frac{p_T(x_{1:T} \mid \theta)}{p_T(x_{1:T} \mid \theta^*)} \right) \right]$ is at most poly$(T)$.

(iii) **Local regularity.** The prior density $\pi$ is continuous and, for any $R > 0$, $t \leq T$,

$$\log \frac{p_t(x_t \mid x_{1:t-1}, \theta)}{p_t(x_t \mid x_{1:t-1}, \theta')} \leq \text{poly}(R) \|\theta - \theta'\|$$

for all $x_{1:t}, \theta, \theta'$ such that $\|x_s\|, \|\theta\|, \|\theta'\| \leq R$

These assumptions are quite mild and are satisfied by our examples, see App. F.2. As a metric to assess the quality of a given retrieved task $\theta$ w.r.t. the true task $\theta^*$, we consider $D_\rho(\theta \| \theta^*)$ the Rényi divergence (Rényi, 1961) of order $\rho \in (0, 1)$ between the distributions $p_T(\cdot \mid \theta)$ and $p_T(\cdot \mid \theta^*)$:

$$-\frac{1}{T(1-\rho)} \log \mathbb{E}_{X \sim p_T(\cdot \mid \theta^*)} \left[ \prod_{t=1}^{T} \left( \frac{p_t(x_t \mid x_{1:t-1}, \theta)}{p_t(x_t \mid x_{1:t-1}, \theta^*)} \right)^\rho \right] .$$

We divide by $T$ to obtain a per-sample divergence that does not trivially diverge as $T$ increases. This metric is standard in the Bayesian consistency literature (Zhang, 2003; 2006; Ghosal & van der Vaart, 2017) and in practical examples it bounds the error of the Bayesian optimal predictor, see App. F.2.

Our main theorem below shows that, under Asm. 4, the posterior distribution over tasks concentrates around the true task $\theta^*$ as the number of in-context examples $T$ increases, at a rate that depends on the properties of the pretraining distribution $\pi$.

**Theorem 2** (Task selection). *Let $\rho \in (0, 1)$, under Asm. 4, with $\pi(\theta^*) > 0$ and $x_{1:T} \sim p_T(\cdot \mid \theta^*)$, the posterior distribution over tasks satisfies*

$$\mathbb{E}_{x_{1:T}} \left[ \mathbb{E}_{\theta \sim \widehat{p}_T(\cdot \mid x_{1:T})} [D_\rho(\theta \| \theta^*)] \right]$$
$$\leq \frac{1+\rho}{(1-\rho)T} \log 1/\pi(\theta^*) + \mathcal{O}\left( \frac{\log T}{T} \right),$$

*where the terms in $\mathcal{O}\left( \frac{\log T}{T} \right)$ do not depend on the prior $\pi$ or are negligible compared to the first term.*

Thm. 2 provides a guarantee on how close the posterior distribution over tasks is to the true task $\theta^*$ as the number of in-context examples $T$ increases. The right-hand-side decays as $\mathcal{O}(1/T)$, which shows that the posterior concentrates around the true task as the number of examples in-context increases. The speed of convergence is governed by the coefficient $\log 1/\pi(\theta^*)$, which quantifies how well the prior $\pi$ covers the true task $\theta^*$: the smaller $\pi(\theta^*)$, the slower the convergence. Since in ICL we wish to study the capabilities of learning a new task from in-context examples, this result quantifies the speed at which ICL learns this new task $\theta^*$: the further $\theta^*$ is from the bulk of the prior $\pi$, the slower ICL learns this new task. Thus, the ability to learn a new task and its robustness to new tasks crucially depends on the tail of the prior $\pi$: the slower the tail of $\pi$ decays, the larger $\pi(\theta^*)$ is for tasks $\theta^*$ far from the modes of $\pi$, and the faster ICL learns these new tasks. This can be observed on the examples of priors presented in Ex. 3.4. For a fixed task $\theta^*$ far from the modes of $\pi$, the error for Student's $t$-distributions with $\nu$ degrees of freedom behaves as $(\nu + 1) \log |\theta^*|/T$ for large $|\theta^*|$ so that lower values of $\nu$, i.e. heavier tails, lead to smaller errors. For Generalized Normal distributions with shape parameter $\beta$, it behaves as $|\theta^*|^\beta/T$ so lower values $\beta$ also lead to smaller errors.

From a technical viewpoint, Thm. 2 is proven in App. E using ideas from Bayesian statistics (Zhang, 2003; 2006)

and is extremely general, covers discrete and continuous task spaces, and does not require any probabilistic structure on the data sequencenor specific data distributions. Moreover, unlike some existing results, Thm. 2 provides a guarantee on the posterior distribution given all $T$ in-context examples, and not only on the regret i.e., the average error of the posterior distributions given $1, \ldots, T$ examples. This better reflects the practical use of ICL, where the user typically only considers the output of the model after all in-context examples have been provided.

Finally, we provide, in App. E.4, a more refined and sharper version of Thm. 2, which also encompasses the case where $\pi(\theta^*) = 0$, in which the ICL error is not vanishing anymore. In this scenario, it shows that ICL can struggle on out-of-distribution tasks, as empirically studied previously (Goddard et al., 2025; Kwon et al., 2025; Yadlowsky et al., 2023). Also note that this more general result can also be used to obtain task identification guarantees for discrete priors.

> ***Takeaway #2:*** *Heavier-tailed priors are beneficial for task identification : they improve the learning speed on new tasks, especially far from the bulk of the pretraining distribution.*

### 3.4. Summary of theoretical predictions

Our theory shines a new light on the role of the pretraining distribution in ICL.We show that heavier-tailed priors actually lead to a trade-off in ICL performance: heavier-tailed priors are beneficial for task identification and robustness to distribution shifts, but harm generalization. More precisely, our theory makes the following predictions:

- **Task selection under distribution shift (Thm. 2):** Heavier-tailed pretraining distributions lead to better ICL performance under larger distribution shifts.
- **Generalization (Thm. 1):** The generalization penalty of heavier-tailed pretraining distributions becomes significant when the number of pretraining tasks is small.
- **Temporal dependence (Thm. 1):** Stronger temporal dependence harms generalization, especially when the number of pretraining tasks decreases.

These predictions are coherent with existing empirical observations in the literature: Chan et al. (2022); Singh et al. (2023) show that using heavier-tailed pretraining distributions improves ICL performance up to a certain point. Our framework explains this phenomenon as a trade-off between task identification and generalization. We validate these predictions empirically in the next section.

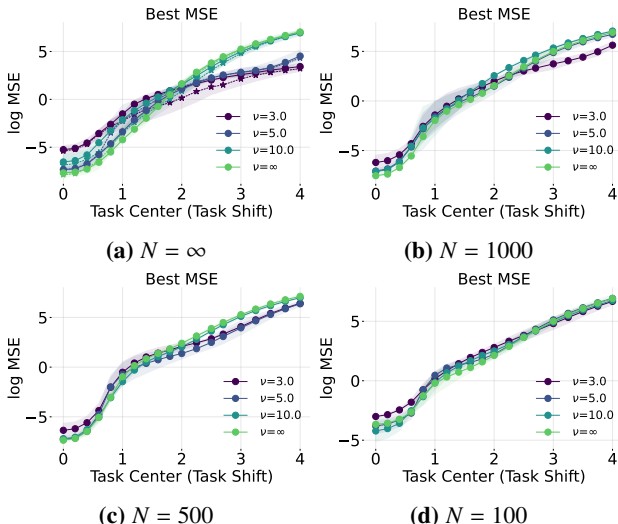

**(a)** $N = \infty$  **(b)** $N = 1000$

**(c)** $N = 500$  **(d)** $N = 100$

**Figure 1:** Influence of the degree of freedom parameter $\nu$ of a Student-$t$ pretraining distribution (lower $\nu$ corresponds to heavier tail) and of the number of tasks $N$ on the ICL error for different task shifts for linear regression.

## 4. Experimental Validation

Our theoretical framework yields several predictions on how the choice of pretraining distribution affects in-context learning (ICL) performance, in particular under distribution shift. We now conduct a series of experiments to demonstrate and validate these prediction.We thus train transformer models under different pretraining distributions to solve different ICL tasks, and evaluate their performance on shifted tasks.

**ICL evaluation through robustness to distribution shift.** The transformer is trained on tasks $\theta$ sampled from a pretraining distribution $\pi$ centered at 0.We will either use a finite number of tasks $N$ sampled from $\pi$ or an unbounded number of tasks, i.e., a new task from $\pi$ is sampled at each training iteration. To assess the ICL performance, we evaluate the trained model on tasks $\theta^* \sim \mathcal{N}(\Delta, I_d)$ where $\Delta$ is a deterministic shift and report the ICL error on these shifted tasks as a function of the shift magnitude $\|\Delta\|$. Note that these evaluations tasks are independent of the choice of pretraining distribution. Studying this error as a function of the shape of the pretraining distribution allows us to validate the theory in § 3.4. We also study the performance of ICL as a function of the number of pretraining tasks to evaluate our predictions regarding generalization.

**Distributions and Metrics.** We experiment with two different families of pretraining distributions: the Student-$t$ distribution with varying degrees of freedom $\nu \in \{3, 5, 10, \infty\}$ (where $\nu = \infty$ corresponds to the normal distribution) and the generalized normal distribution with varying shape parameter $\beta \in \{1, 1.5, 2, 2.5\}$ (where $\beta = 2$ corresponds to the normal distribution). In both cases, lower parameter values indicate heavier tails of the distribution. Note that the scale parameter is chosen such that all distributions keep

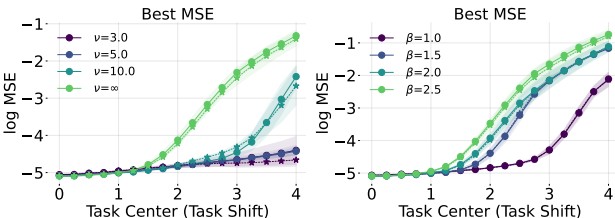

Figure 2: (Left) Influence of the degree of freedom parameter $\nu$ of a Student-$t$ pretraining distribution (lower $\nu$ corresponds to heavier tail) on the ICL error for different task shifts for predicting the next step in an OU process with context length of 32. (Right) Influence of the shape $\beta$ of a generalized normal distribution (lower $\beta$ corresponds to heavier tail) on the ICL error for different task shifts for predicting the next step in an OU process.

the same variance. We refer to App. C for details on the data generation, model architecture and optimization. For all experiments, we consider mean squared error (MSE) and report the best MSE over the context length, which is given by $\min_t (\hat{f}(x_{1:t}) - x_{t+1})^2$. The mean MSE and MSE at full context length behave similarly and are reported in App. B .

### 4.1. Linear Regression

We first consider the linear regression setting (Ex. 3.2) where each $\theta \in \mathbb{R}^d$ defines a linear regression task $y_i = \theta^T q_i + \epsilon_i$ for $i = 1, ..., 64$ where 64 is the context length. During pretraining, we sample $\theta$ according to different Student-$t$ distributions, with the same location and variance but different shape parameters $\nu$ and thus different tail heaviness.

**Task identification.** The first prediction from § 3.4 is that heavier-tailed pretraining distributions should lead to better performance under larger distribution shifts. To empirically validate this prediction without confounding effects from generalization, we first consider the case where the number of tasks used for pretraining is unbounded ($N = \infty$). The results in Fig. 1a show that for small distribution shifts, the lighter-tailed prior (higher $\nu$) performs best, but as the shift increases, the heavier-tailed priors (lower $\nu$) outperform the lighter-tailed ones, confirming the prediction of § 3.4. To further investigate, we also explore the effect of reweighting the pretraining distribution, see App. B.4 for details.

**Generalization.** To validate our second prediction from § 3.4, we now consider the case where the number of pretraining tasks is finite and study how well the model generalizes to unseen tasks as a function of the number of pretraining tasks. § 3.4 predicts that heavier-tailed priors require more samples to generalize well, so we expect that for small number of pretraining tasks, heavier-tailed priors will loose their advantage over lighter-tailed priors on out-of-distribution tasks. The results are presented in Fig. 1, which quantitatively confirms this prediction: for small $N$, light-tailed priors eliminate the performance gap with heavy-tailed priors on out-of-distribution tasks tasks, precisely as

predicted. Thus, for small number of pretraining tasks, the advantage of heavier-tailed priors for task selection is offset by their worse generalization.

### 4.2. Linear Stochastic Differential Equations

In the next experiment, we follow the setup in Ex. 3.3 with a stochastic process satisfying (1). Our metric of success is the MSE $(\hat{X}_{t+1} - \mathbb{E}[X_{t+1} \mid X_t])^2$ where $\hat{X}_{t+1}$ is the prediction with the context of $X_{1:t}$. The task parameters $\theta, \mu$ are sampled from different pretraining distributions and we again compare the performance of ICL on different test tasks. We focus on validating the first prediction from § 3.4 regarding task selection under distribution shift, and thus consider an unbounded number of pretraining tasks. In addition to the Student-$t$ distribution, we also report results with the generalized normal distribution as a pretraining prior, see Fig. 2. In both instances, our prediction is quantitatively confirmed: the heavier tailed pretraining distributions (lower $\nu$ or $\beta$) perform better for larger task shifts.

### 4.3. Stochastic Volterra Equations

In § 3, we predicted that temporal dependencies in the data would negatively impact generalization in ICL. To quantitatively validate this prediction, we finally consider stochastic Volterra equations as a model of nonlinear stochastic processes that have long range dependencies. These processes are, under certain conditions, known to model fractional Brownian motion, which exhibit self-similarity which has been thought to represent the distribution of tokens in LLMs (Alabdulmohsin et al., 2024). Each task $\theta$ parametrizes a multi-layer perceptron $b_\theta$ and induces the process: $X_t = X_0 + \int_0^t (t-s)^{-\alpha} b_\theta(X_s) ds + \int_0^t (t-s)^{-\alpha} dW_s$, where $W_t$ is a standard Brownian motion and the kernel exponent $\alpha > 0$ controls the temporal dependence of the process: the smaller $\alpha$ is, the more past values influence the current value. The dependency coefficients in Thm. 1 thus depend explicitly on $\alpha$, they are larger for smaller $\alpha$, see App. F.1. We consider the generalization capabilities as a function of the number of pretraining tasks in Fig. 3 and as a function of $\alpha$. Thm. 1 predicts that generalization should suffer for smaller $\alpha$ due to the increased dependencies, which is validated in the experiments: the performance gap between the different $\alpha$ is larger for smaller number of tasks. More precisely, sequences with lower kernel exponents such as 1.0 (higher dependence) have worse performance and degrades faster as the number of tasks decreases compared to sequences with higher kernel exponents such as 2.0 (lower dependence). For instance, for kernel exponent 1.0, the MSE at shift 0 is multiplied by 3 when $N$ goes from 5000 to 500 while for kernel exponent 2.0, the MSE at shift 0 is barely changes. These results thus validate our predictions.

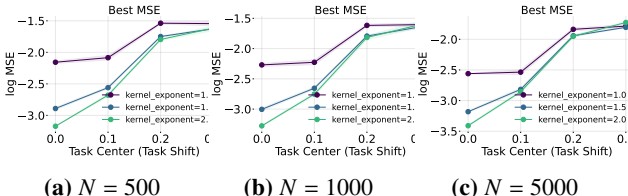

**(a)** $N = 500$      **(b)** $N = 1000$      **(c)** $N = 5000$

**Figure 3:** Generalization of a transformer trained to predict the next step of the Volterra as a function of $N$ the number of tasks with context length of 32.

## 5. Discussion

**Connections to practical pretraining settings.** Our theoretical framework is developed in a controlled mixture-of-tasks setting, but its predictions connect to practical pretraining pipelines. For numerical and tabular data, our conclusions apply directly. A natural example is tabular foundation models (Hollmann et al., 2023; Qu et al., 2025), where one samples a class of functions and inputs from some distribution and then trains a large-scale model on this data. Our work suggests that, in such settings, the choice of pretraining distribution can directly affect the robustness/generalization trade-off. In more realistic NLP pretraining settings, the natural analogue is a long-tailed mixture over latent tasks, domains, or input-output patterns—rather than a mixture concentrated almost entirely on highly frequent task types. This connection is consistent with empirical observations: Chan et al. (2022) and Singh et al. (2023) show that heavier-tailed pretraining distributions improve ICL performance up to a point, beyond which performance degrades, in line with the trade-off our framework identifies. Studying this phenomenon with natural language tasks and large language models is an important topic and a natural follow-up to our work.

**Computational implications.** Our framework focuses primarily on the statistical trade-off between task identification and generalization. That said, our results suggest an indirect computational implication: if a pretraining distribution enables faster task identification, then fewer in-context examples may be needed at test time, which reduces the required context length. Since attention cost grows quadratically with context length, this can translate into computational savings. A combined study of statistical and computational trade-offs is an interesting direction for future work.

### 5.1. Limitations

**Sub-Gaussian assumption.** Thm. 1 assumes uniform sub-Gaussianity of the per-sample loss. We choose this simplifying assumption to focus on highlighting the impact of the properties of $\pi$ on the ICL error. Note that our analysis can be extended to cover the case where the constant in the sub-Gaussian assumption grows with $\theta$: at the cost of a more intricate proof, this would yield the same theorem as Thm. 1. The fully general case is a subject of future work.

**Bayesian-optimality gap.** Our task-selection result (Thm. 2) characterizes the Bayes-optimal predictor with respect to $\pi$. Several works have shown empirically that sufficiently expressive trained transformers closely track the Bayes-optimal predictor (Chan et al., 2022; Raventós et al., 2023; Wurgaft et al., 2025; Nguyen & Reddy, 2025; Park et al., 2025a). We therefore interpret Thm. 2 as characterizing the regime that actual trained transformers are expected to approach when they are sufficiently powerful and well trained. Importantly, our empirical study is conducted with actual trained transformers, and the qualitative trends predicted by the theory are borne out in those experiments.

**Empirical scope.** Our experiments are limited to controlled numerical tasks (linear regression, Ornstein-Uhlenbeck processes, Volterra equations), which allow precise control over the statistical variables of interest. Our empirical support for the task-selection mechanism is thus indirect: rather than directly measuring posterior concentration, we evaluate prediction error under task shift, and show that the observed behavior is consistent with the task-selection effect predicted by the theory. Validation on large-scale natural language datasets, and a more direct empirical proxy for task retrieval, remain important directions for future work.

## 6. Conclusion

In this work, we characterize how statistical properties of the pretraining distribution, such as tail behavior, coverage, and temporal dependence, shape ICL performance. Our theory covers both task identification and generalization, extends to heavy-tailed priors and dependent sequences, and exposes a fundamental design trade-off: heavier-tailed pretraining distributions improve ICL performance under distribution shifts, but can degrade generalization in low-data regimes, while stronger dependence increases the amount of data needed to generalize reliably. We validate these predictions on challenging numerical probes, including stochastic differential equations and stochastic processes with memory, highlighting practical guidelines for designing pretraining distributions that enhance ICL capabilities for transformers. A prominent direction for future work is to extend these insights to LLMs trained on text data, where the pretraining distribution can be shaped through data curation and fine-tuning strategies.

## Impact Statement

This paper presents work whose goal is to advance the field of machine learning. There are many potential societal consequences of our work, none of which we feel must be specifically highlighted here.

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

# A. Additional Related Work

**Training dynamics of ICL**    Varre et al. (2025) shows that *n*-grams are approximate stationary points in the training of two-layers transformers. Zhang et al. (2025a) studies the training dynamics of a one-layer linear transformer with linear attention on linear regression tasks. Sander et al. (2024) characterize the training dynamics of a one-linear layer transformer on auto-regressive tasks, showing how ICL emerges. Ahn et al. (2023a) show that for linear regression problems and a linear transformer, the global minimizer of the training loss corresponds to performing one step of preconditioned gradient descent. In contrast, our approach focuses on the influence of the pre-training distribution on ICL. We therefore assume that the model is sufficiently expressive and trained optimally enough to approximate the Bayes optimal predictor. We refer to recent works on optimization dynamics of transformers Gao et al. (2024); Barboni et al. (2025); Azizian et al. (2025) and on the approximation capabilities of transformers.

**Approximation capabilities of transformers**    The foundational works of Von Oswald et al. (2023); Akyürek et al. (2023) demonstrate that transformers can implement gradient descent. This has led to a fruitful line of work studying the algorithmic capabilities of transformers. Bai et al. (2023) show that transformers can implement a wide variety of statistical methods. Wang et al. (2025a) shows how transformers can implement functional gradient descent on categorical data, generalizing previous works. Wu et al. (2025) shows how attention transformers can implement gradient descent on a ReLU network. Sander & Peyré (2025) explicitly constructs a transformer that implements kernel causal regression. On a more abstract perspective, Furuya et al. (2025); Kratsios & Furuya (2025) show that (causal) transformers can approximate any (causal) map between measures. Wang & Weinan (2024) studies quantitatively the approximation properties of transformers on "sparse memory" target functions. Li et al. (2025a) obtains explicit approximation bounds for numerical ICL tasks.

# B. Additional Experimental Results

## B.1. Linear Regression

We provide comprehensive experimental results for linear regression tasks (detailed in § 4.1) using Student-$t$ and generalized normal pretraining distributions. This section presents the ICL error as a function of context length (ICL step) for Student-$t$ priors with degrees of freedom $\nu \in \{3, 5, 10, \infty\}$ and generalized normal priors with shape parameters $\beta \in \{1, 1.5, 2, 2.5\}$, see Table 2. For all experiments, we consider mean squared error (MSE). We first report the ICL performance as a function of the number of in-context examples for different levels of distribution shift in Figs. 4 and 5.

The results in Fig. 4 clearly demonstrate the fundamental trade-off in selecting pretraining distributions for ICL: heavy-tailed priors (small $\nu$) achieve superior performance under distribution shift, while light-tailed priors (large $\nu$) excel on in-distribution tasks. In contrast, Fig. 5 shows that varying the shape parameter of generalized normal priors produces more subtle effects on ICL performance in the linear regression setting.

We also notice on Figs. 4 and 5 that longer context lengths are mostly beneficial for in-distribution tasks: as the perturbation magnitude increases, the performance gains from longer contexts diminish. This is in line with § 3.3: the performance gain per new example is determined by the prior probability of the task, which decreases with larger perturbations.

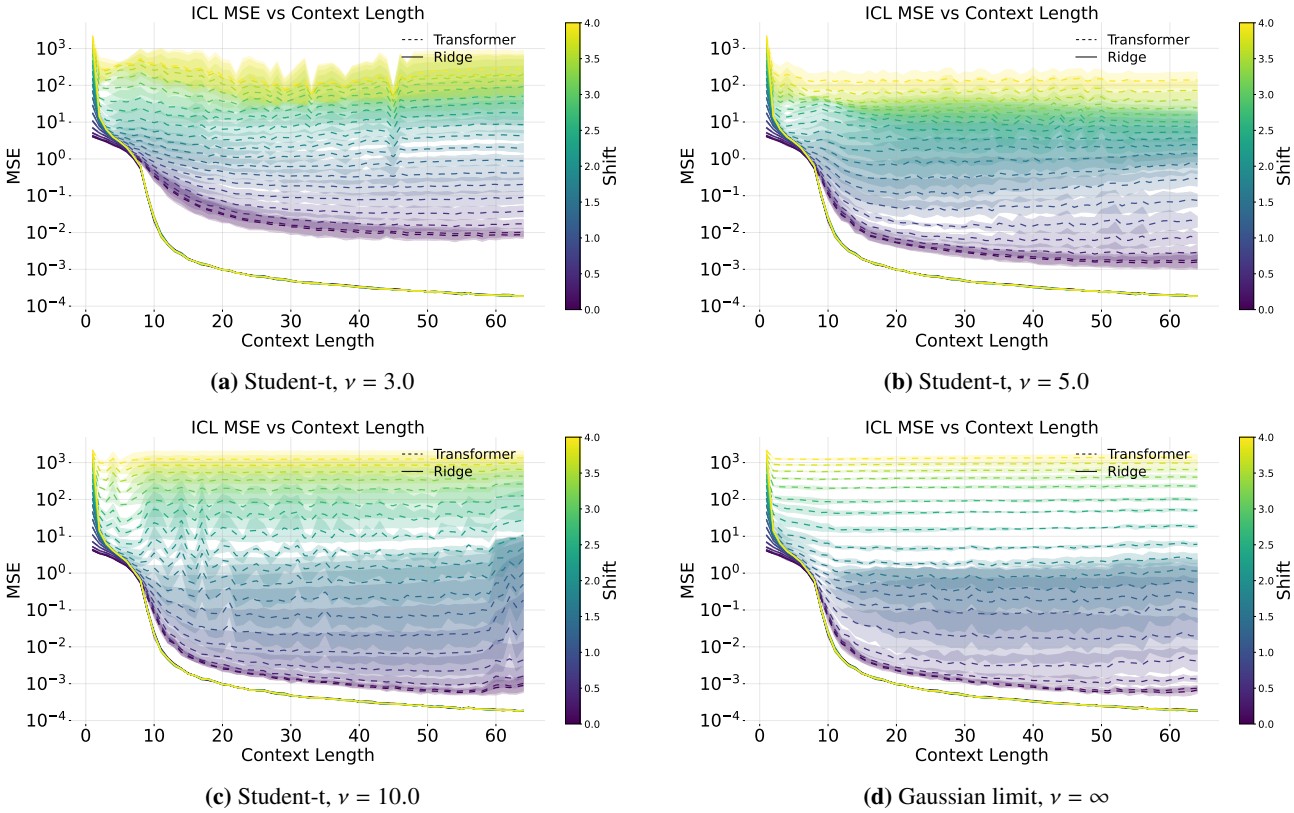

**(a)** Student-t, $\nu = 3.0$      **(b)** Student-t, $\nu = 5.0$

**(c)** Student-t, $\nu = 10.0$      **(d)** Gaussian limit, $\nu = \infty$

**Figure 4:** Linear regression with Student-$t$ pretraining distributions: MSE as a function of ICL step for different task shift magnitudes. Heavy-tailed priors ($\nu = 3$) show superior robustness to distribution shift, while light-tailed priors ($\nu = \infty$, Gaussian) perform better on unperturbed tasks. The Ridge regression baseline provides a reference that remains constant across perturbation magnitudes.

We now present an extended analysis of the results from § 4.1. We report the best MSE over the context length, which is given by $\min_t (\hat{f}(x_{1:t}) - x_{t+1})^2$, and, additionally, the mean MSE given by $\frac{1}{T} \sum_{t=1}^{T} (\hat{f}(x_{1:t}) - x_{t+1})^2$; and finally the full

**Table 2:** Pre-training distribution parameters.

| Dist. | Param. |
|---|---|
| Generalized Normal | $\beta \in \{1, 1.5, 2, 2.5\}$ |
| Student-$t$ | $\nu \in \{3, 5, 10\}$ |

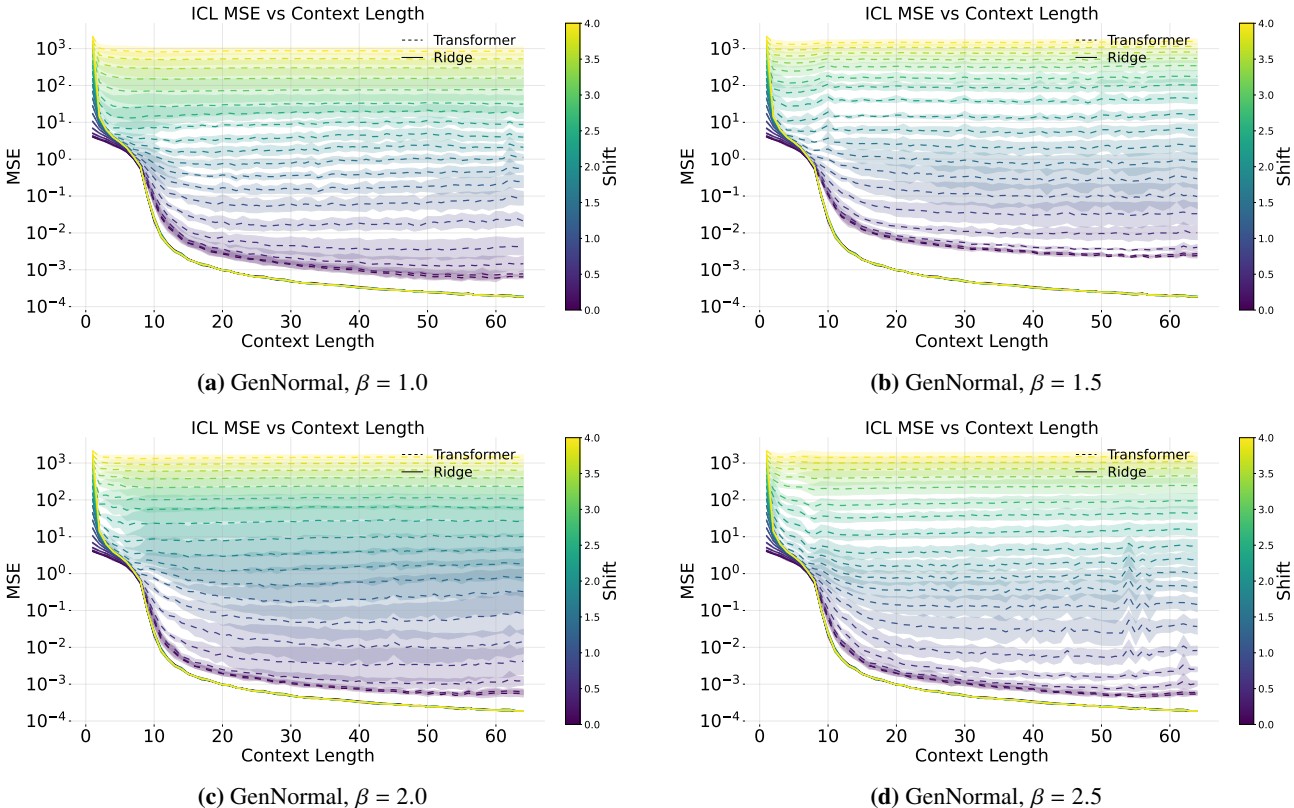

**(a)** GenNormal, $\beta = 1.0$

**(b)** GenNormal, $\beta = 1.5$

**(c)** GenNormal, $\beta = 2.0$

**(d)** GenNormal, $\beta = 2.5$

**Figure 5:** Linear regression with generalized normal pretraining distributions: MSE as a function of ICL step for different task shift magnitudes. The shape parameter $\beta$ has a more modest impact on performance compared to Student-$t$ distributions, with all variants showing similar convergence patterns across perturbation levels.

context length MSE given by $(\hat{f}(x_{1:T-1}) - x_T)^2$. These allow us to see how the different priors perform while taking into consideration the full context length.

We first provide an extended version of Fig. 1a in Fig. 6 for Student-$t$ priors with varying degrees of freedom $\nu$ with these additional metrics.

We present an extended analysis of the results from Fig. 1 in Fig. 7, examining how the number of pretraining tasks $n$ affects performance across different Student-$t$ tail parameters $\nu$. These results validate Thm. 1, showing that heavy-tailed priors require more training tasks to achieve comparable performance to light-tailed priors.

Finally, we provide an ablation study on the effect of the variance. All other experiments are designed so that the pretraining distribution has unit variance in each dimension. In Fig. 8, we vary the variance of a standard Gaussian pretraining distribution and observe it only changes the ICL performance for in-distribution tasks.

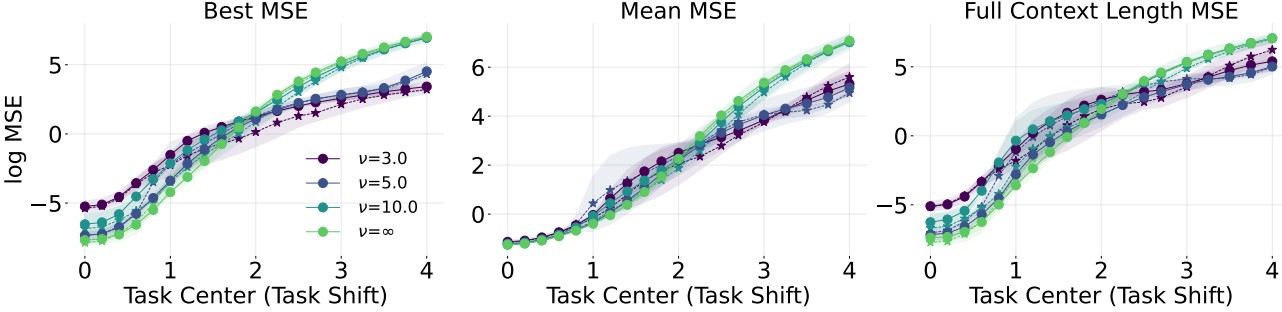

**Figure 6:** Influence of the degree of freedom parameter $\nu$ of a Student-$t$ pretraining distribution (lower $\nu$ corresponds to heavier tail) on the ICL error for different task shifts with context length of 64 for linear regression.

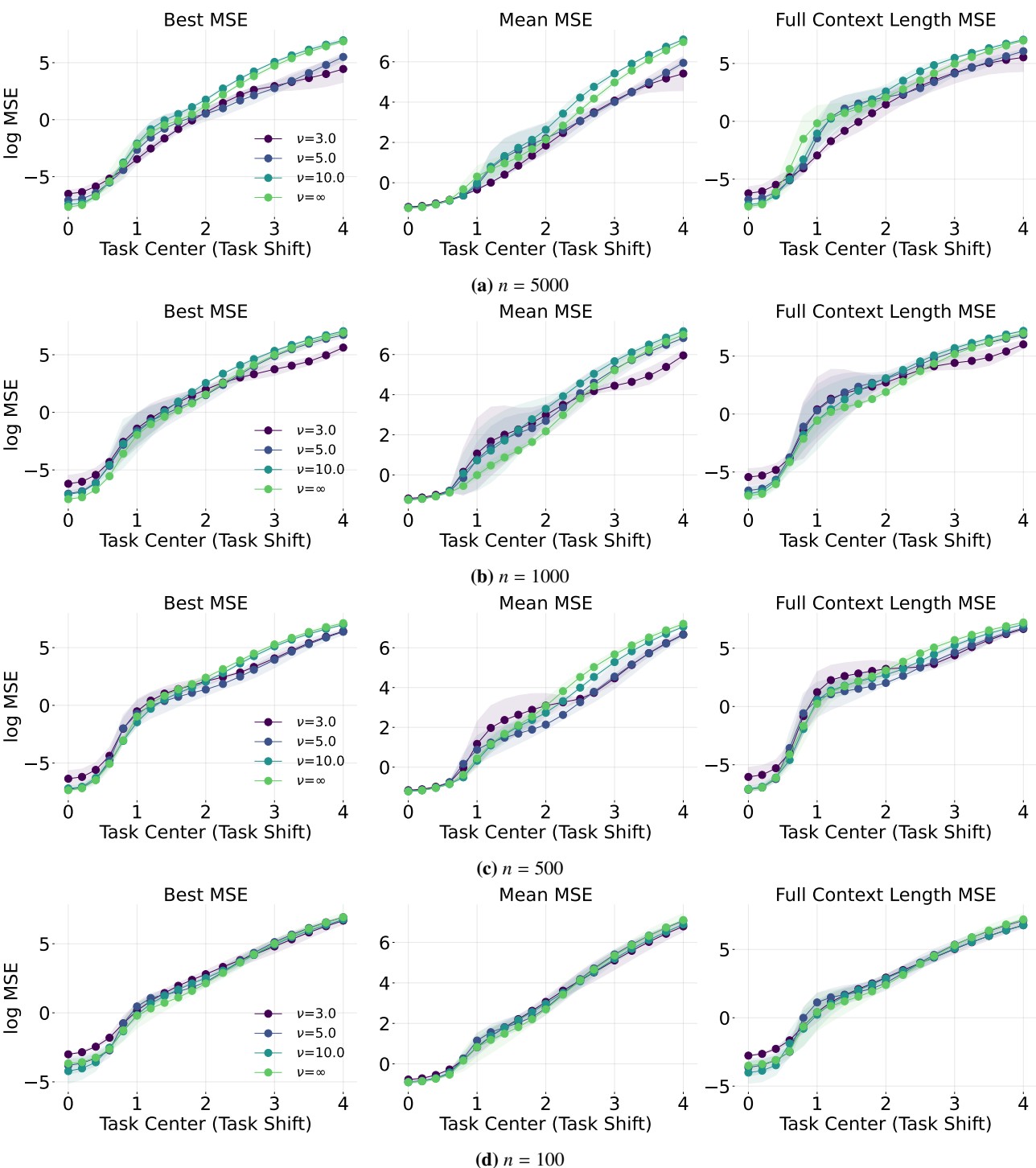

**Figure 7:** Generalization analysis for linear regression across different numbers of pretraining tasks $n$ for a context length of 64. As predicted by Thm. 1, heavy-tailed priors (small $\nu$) require more tasks to achieve performance comparable to light-tailed priors, but eventually outperform them under distribution shift. The crossover point shifts to larger $n$ for heavier-tailed distributions.

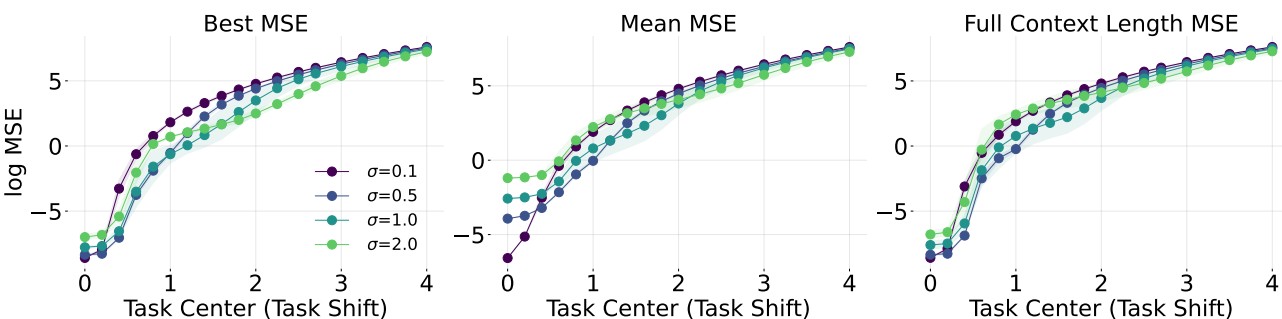

**Figure 8:** Ablation on the effect of variance for Gaussian pretraining distributions in linear regression. Only in-distribution performance is affected by the variance, with larger variances leading to worse performance.

### B.2. Ornstein–Uhlenbeck Processes

We present detailed experimental results for Ornstein–Uhlenbeck (OU) stochastic processes (described in § 4.2) using both Student-$t$ and generalized normal pretraining distributions. The figures show ICL error as a function of context length for Student-$t$ priors with degrees of freedom $\nu \in \{3, 5, 10, \infty\}$ and generalized normal priors with shape parameters $\beta \in \{1, 1.5, 2, 2.5\}$ (see Table 2) in Figs. 9 and 10, respectively.

Notably, OU processes exhibit different behavior compared to linear regression: the trade-off between in-distribution and out-of-distribution performance is less pronounced. As shown in both Figs. 9 and 10, heavy-tailed priors maintain competitive in-distribution performance while still providing improved robustness to distribution shift.

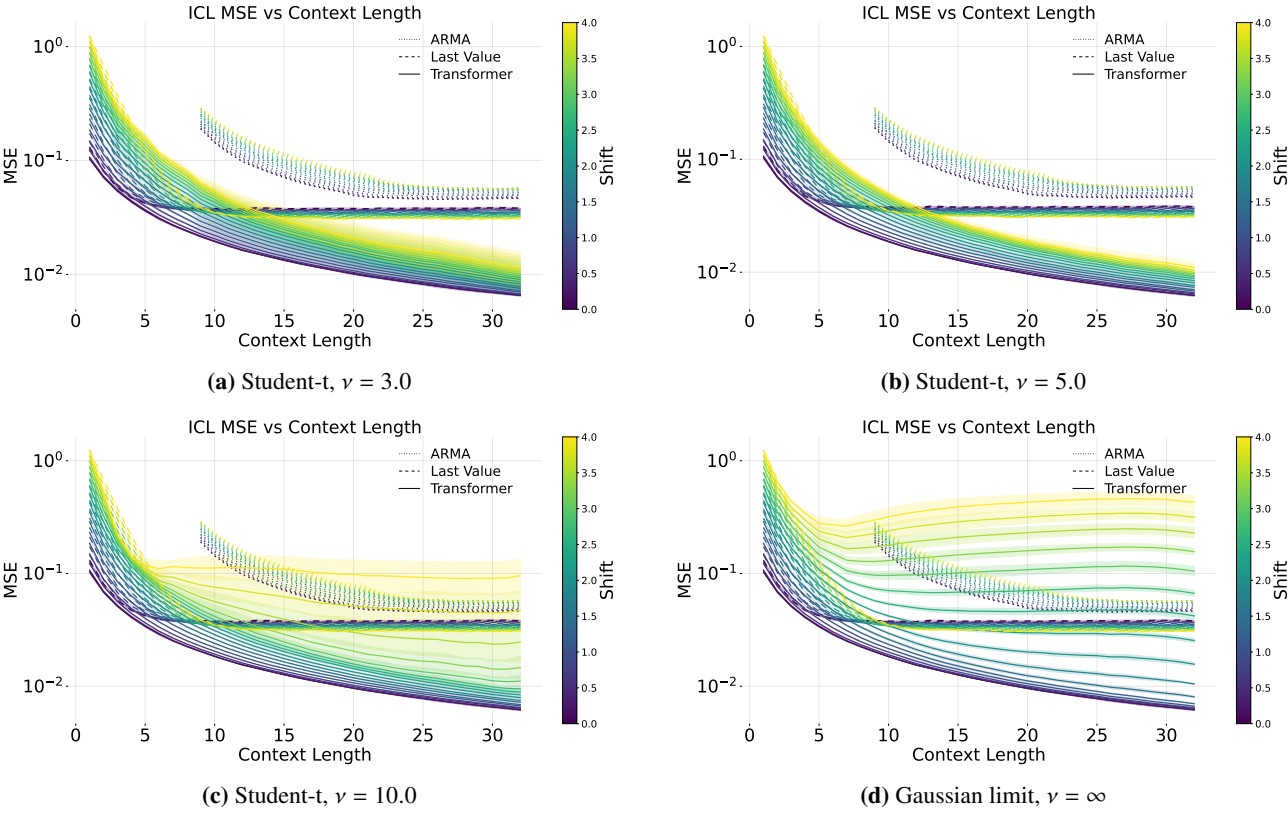

**Figure 9:** Ornstein–Uhlenbeck processes with Student-$t$ pretraining distributions: MSE as a function of ICL step for different task shift magnitudes. Unlike linear regression, heavy-tailed priors maintain strong in-distribution performance while providing superior robustness to perturbations. Baselines include predicting the last observed value and fitting an ARMA(5) model to the context.

We now present an extended version of Fig. 2 in Fig. 11 for Student-$t$ priors with varying degrees of freedom $\nu$ with the additional metrics of mean MSE and full context length MSE and Fig. 12 for generalized normal priors with varying shape parameters $\beta$.

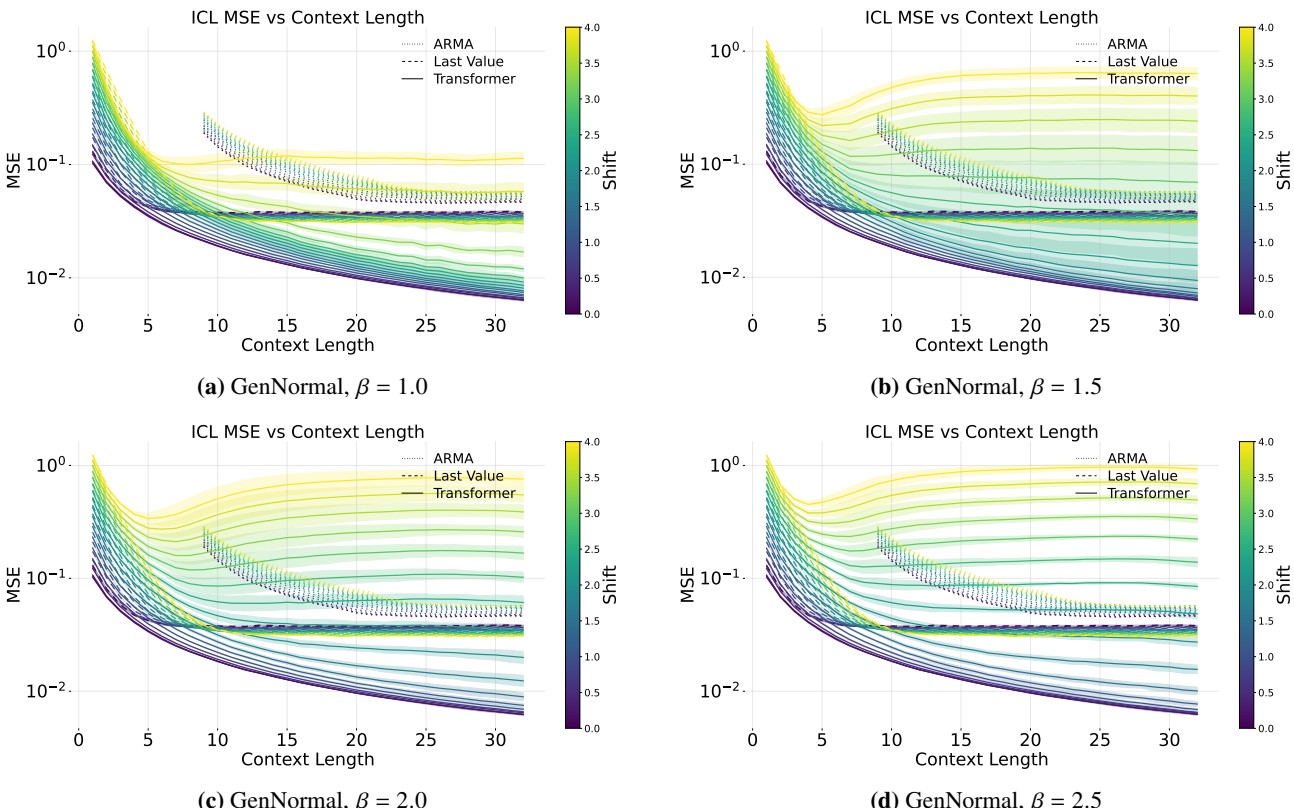

**Figure 10:** Ornstein–Uhlenbeck processes with generalized normal pretraining distributions (importance weighted): MSE as a function of ICL step for different task shift magnitudes. The shape parameter $\beta$ shows consistent effects across perturbation levels, with all variants significantly outperforming simple baselines. Importance weighting provides modest improvements in robustness.

### B.3. Volterra Processes

We present comprehensive results for stochastic Volterra equations (detailed in § 4.3), which model nonlinear processes with long-range dependencies and connections to fractional Brownian motion. Fig. 13 shows ICL error as a function of context length for different kernel exponents $\alpha \in \{1, 1.5, 2\}$, where smaller $\alpha$ values correspond to stronger temporal dependencies.

The results confirm our theoretical predictions from § 3: as the kernel exponent $\alpha$ increases (weaker dependencies), both convergence speed and final performance improve significantly. This validates the dependency structure analysis in Thm. 1.

Fig. 14 extends the generalization analysis from Fig. 3, demonstrating how the number of pretraining tasks $n$ interacts with the temporal dependency parameter $\alpha$. The results show that processes with stronger dependencies (smaller $\alpha$) require substantially more training data to achieve comparable performance.

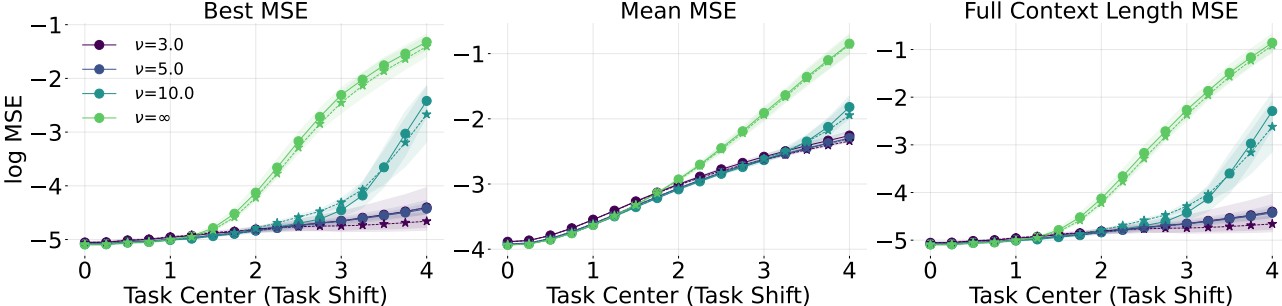

**Figure 11:** Influence of the degree of freedom parameter $\nu$ of a Student-$t$ pretraining distribution (lower $\nu$ corresponds to heavier tail) on the ICL error for different task shifts for predicting the next step in an OU process with context length of 32.

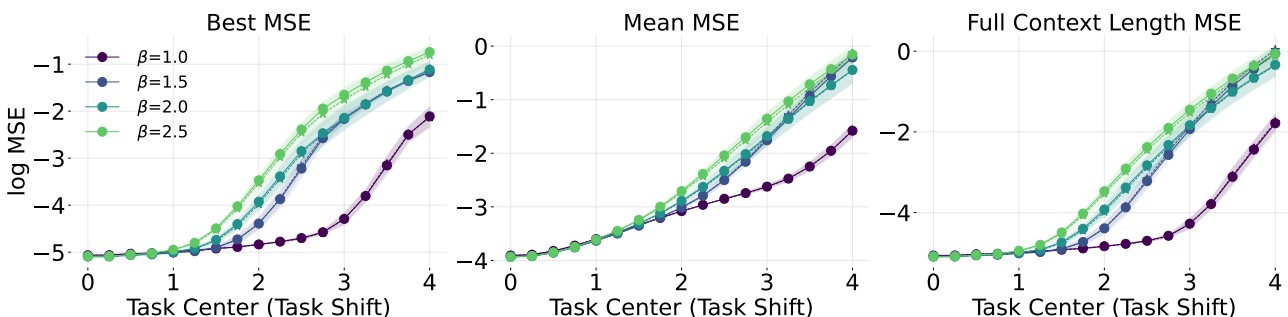

**Figure 12:** Influence of the shape $\beta$ of a generalized normal distribution (lower $\beta$ corresponds to heavier tail) on the ICL error for different task shifts for predicting the next step in an OU process.

### B.4. Reweighting

To further investigate the predictions of Thm. 2, we consider reweighting the pretraining distribution: if we are given tasks sampled from a prior distribution $\pi$, but know that another pretraining distribution $\rho$ exhibits strong performance, can we improve the performance of distribution $\pi$ by matching $\rho$ via importance sampling i.e. $\mathbb{E}_\rho[\ell(X)] = \mathbb{E}_\pi\left[\ell(Y)\frac{d\rho}{d\pi}\right]$? To test this approach, we reweigh samples such that they are approximately uniform over the support of the empirical distribution. More precisely, we set $\rho$ to be the uniform distribution over the range of values observed in the pretraining tasks, and set the weights to be proportional to the ratio of the density of $\rho$ to that of $\pi$ evaluated at each pretraining task, where $\pi$ is a Student-$t$ distribution with varying degrees of freedom $\nu$. For linear regression, results are presented in Fig. 6 where the reweighting results indicated by the $-\star$ markers. The results indicate small improvement in the performance under large shifts using the reweighting as compared to without reweighting. Similarly, for Ornstein–Uhlenbeck processes of § 4.2, the results are presented in Fig. 11 and Fig. 12. In the generalized normal case, the effect of reweighting is practically negligible, but in the Student-$t$ case, we see some benefit, particularly in the large shift regime.

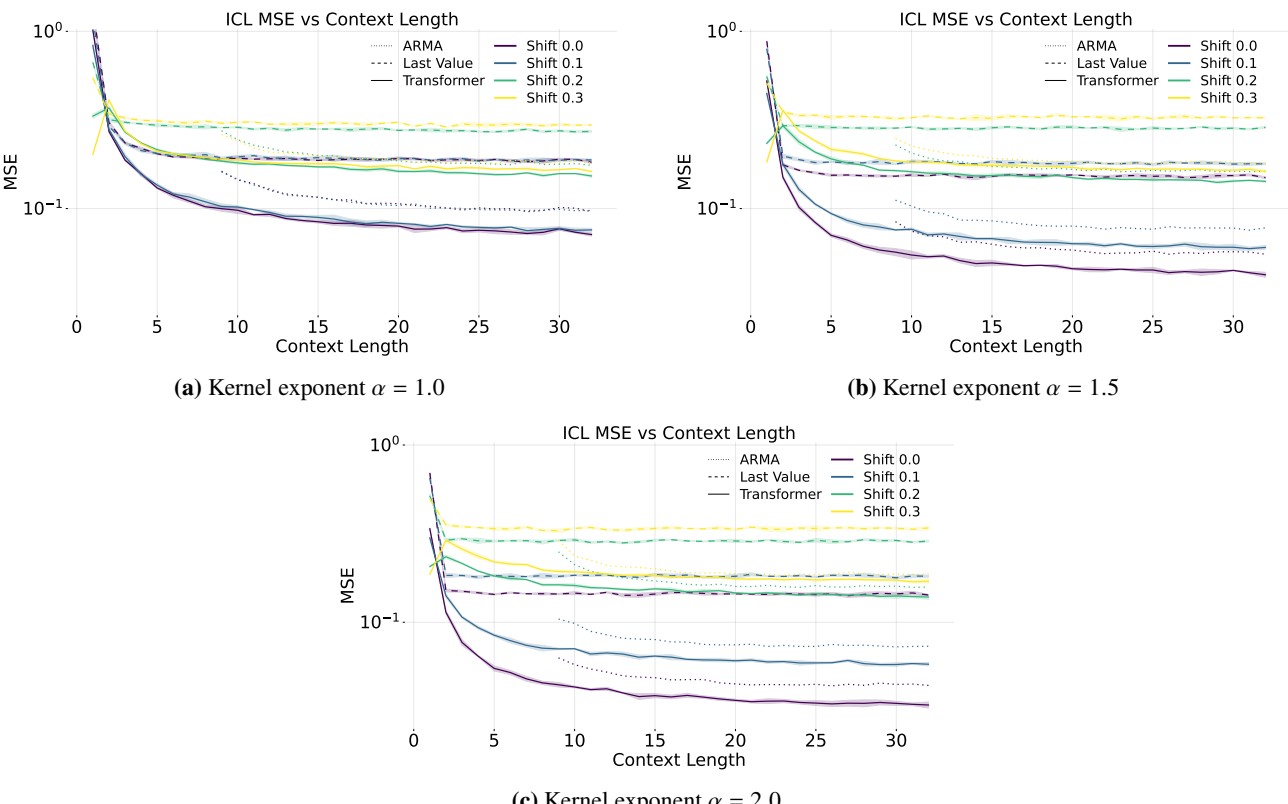

**(a)** Kernel exponent $\alpha = 1.0$

**(b)** Kernel exponent $\alpha = 1.5$

**(c)** Kernel exponent $\alpha = 2.0$

**Figure 13:** Stochastic Volterra equations: MSE as a function of ICL step across different kernel exponents $\alpha$. Smaller $\alpha$ values correspond to stronger long-range dependencies, leading to slower convergence and higher final error. The performance gap between different $\alpha$ values demonstrates the impact of temporal dependency structure on ICL learning. Simple baselines provide reference points for comparison.

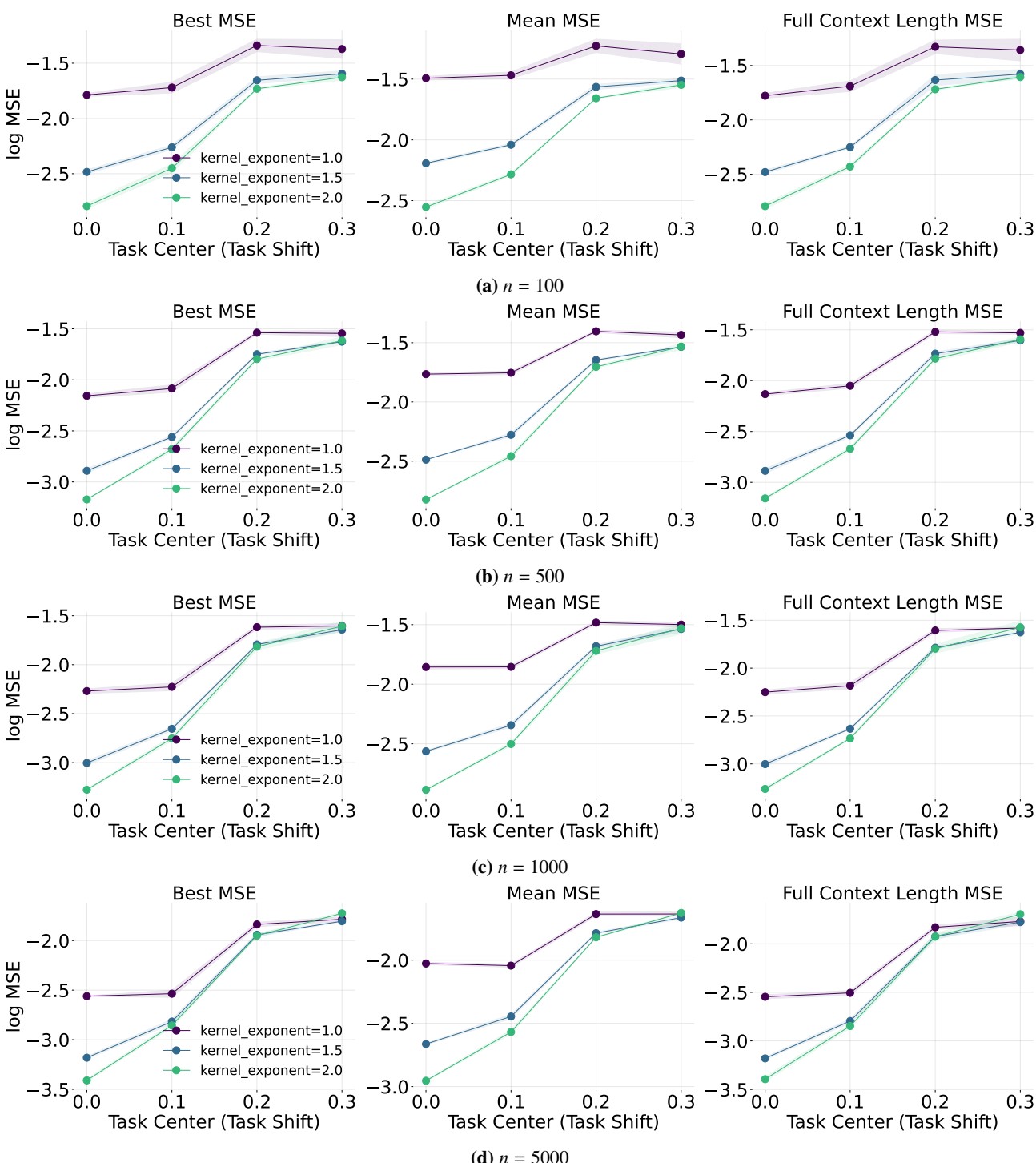

**(a)** $n = 100$

**(b)** $n = 500$

**(c)** $n = 1000$

**(d)** $n = 5000$

**Figure 14:** Generalization analysis for Volterra processes across different numbers of pretraining tasks $n$. Processes with stronger temporal dependencies (smaller $\alpha$) exhibit larger performance gaps at low $n$, consistent with Thm. 1. The dependency coefficients in our theory scale with $\alpha$, explaining why more training tasks are needed to achieve good performance for smaller $\alpha$ values.

# C. Experimental Details

We roughly follow the experimental setup used by Raventós et al. (2023).

## C.1. Data Generation

In all experiments, task parameters $\theta \in \mathbb{R}^d$ are sampled from the distribution mentioned in the main text, data sequences are sampled according to the task. All task distributions during training are zero mean and unit variance in each dimension, except for the Volterra experiments where they are normalized to have standard deviation 0.2. For testing, we sample $\theta$ from $\mathcal{N}(\mu \mathbb{1}, I)$ where $\mu \in \mathbb{R}$ is the shift value and $\mathbb{1}$ is the all ones vector, and the data is sampled according to this task. Unless otherwise specified, a new set of tasks $\theta$ is sampled for each training iteration. Otherwise, when the number of tasks is specified, we sample that many tasks at the start of training and use those same tasks throughout training.

**Linear Regression**   Given a task parameter $\theta \in \mathbb{R}^8$, we sample $x_i \sim \mathcal{N}(0, I_8)$ and $y_i = \langle x_i, \theta \rangle + \epsilon_i$ where $\epsilon_i \sim \mathcal{N}(0, 0.5^2)$. Given a context of $(x_1, y_1), \ldots, (x_k, y_k)$, the model is trained to predict $y_{k+1}$ given $x_{k+1}$ with the MSE loss. At evaluation, we evaluate the model output against $x_i^\top \theta$. We refer to the linear regression experiments in Raventós et al. (2023) for details.

**Ornstein-Uhlenbeck (OU) Process**   The OU process is given by $\mathrm{d}X_t = \tau(\mu - X_t)\mathrm{d}t + \sigma \mathrm{d}W_t$ and has two parameters: $\theta$ and $\mu$. We study a 8-dimensional process where $X_t \in \mathbb{R}^8$ and $\sigma = 0.5I_8$. We consider the initial distribution of $x_0 \sim \mathcal{N}(0, I_8)$. Full paths of $X_t$ are sampled using the Euler-Maruyama method with a step size of $\Delta t = 0.8$. For the sampling of tasks, $\theta \in \mathbb{R}^9$ is sampled from the described distribution, $\mu$ is then set to be the first 8 components of $\theta$ and $\tau$ is set to $0.3 + 0.2 \times \sigma(-0.4\theta_9)$ where $\sigma$ is the sigmoid function. The model is trained to predict $X_{(k+1)\Delta t}$ given $X_0, X_{\Delta t}, \ldots, X_{k\Delta t}$ with the MSE loss with a maximum context length of 32. For evaluation, we evaluate the model output against $\mathbb{E}[X_{(k+1)\Delta t}|X_0, X_{\Delta t}, \ldots, X_{k\Delta t}]$ which is computable in closed form.

**Volterra Process**   We study a Volterra process in dimension 8 given by

$$X_t = X_0 + \int_0^t (t - s)^{-\alpha} b_\theta(X_s)\mathrm{d}s + \int_0^t (t - s)^{-\alpha} \sigma \mathrm{d}W_s, \tag{C.1}$$

where the parameter $\alpha$ is chosen according to discrete values in $\{1, 1.5, 2\}$ and $\sigma = 0.6I_8$. $X_0$ is sampled from $\mathcal{N}(0, I_8)$ again. $b_\theta$ a clipped two-layer neural network and hidden dimension 16: formally, with $\theta = (W_1, b_1, W_2, b_2)$ then $b_\theta(x) = \mathrm{clip}(10(W_2 \tanh(W_1 x + b_1) + b_2), -2, 2) - 0.1x$.

We subsample the paths $(X_t)_t$ with step size $\Delta t = 2$ to obtain discrete samples $(X_0, X_{\Delta t}, X_{2\Delta t}, \ldots,)$ and each $X_{k\Delta t}$ is computed from past samples using 10 steps of the Euler-Maruyama method with step size $\Delta t/10$. The model is trained to predict $X_{(k+1)\Delta t}$ given $X_0, X_{\Delta t}, \ldots, X_{k\Delta t}$ with the MSE loss with a maximum context length of 32. For evaluation, we evaluate the model output against $\mathbb{E}[X_{(k+1)\Delta t}|X_0, X_{\Delta t}, \ldots, X_{k\Delta t}]$ which is computable in closed form.

## C.2. Architecture and Optimization Details

For all experiments, we consider the architecture inspired by GPT-2 as used in Raventós et al. (2023). For linear regression experiments, we use a context length of 64 points, 6 layers, embedding dimension of 32, 8 attention heads and an output dimension of 1. For the other experiments, we use a context length of 32 points, 8 layers, embedding dimension of 128, 2 attention heads and an output dimension of 8.

All models were trained for $5 \times 10^5$ iterations. Experiments are run with AdamW optimizer with a weight decay of 0.1 with a cosine learning rate schedule and 50,000 warmup steps. All experiments were run on NVIDIA H100 GPUs. We performed a hyperparameter sweep over learning rate where we considered two learning rates and chose the best model. Experiments are repeated 3 different times with different seeds.

# D. Generalization bounds

## D.1. Moment bounds for general functions

In this subsection, we generalize the heavy-tail concentration results of Li & Liu (2024a) to allow for non-i.i.d. data. This section can also be seen as extending concentration results for dependent sequences to the case where the function of interest does not necessarily admit bounded differences but only bounded moments. In particular, Lem. D.1 extends the coupling argument of Chazottes et al. (2007) to our setting, in particular not requiring bounded differences but only bounded moments. Indeed, for this, we replace the total variation distance by the Wasserstein-1 distance. It can also be seen as an extension of the bounded differences result of Kontorovich & Ramanan (2008) to our setting (see Mohri & Rostamizadeh (2010) for a presentation of the results of Kontorovich & Ramanan (2008) in a setting closer to ours). Moreover, note that even the handling of the subGaussian increments is much more trickier than in Kontorovich (2014), since we have to carefully apply a convex domination argument to handle the conditional dependence. The main result of this section is Thm. D.1, which is of independent interest.

As in the previous section, $\|\cdot\|$ denotes the Euclidean norm on $\mathbb{R}^d$ for any $d \in \mathbb{N}$.

At multiple places, we will use the Wasserstein-1 distance[3] with respect to a cost function $\rho \colon \mathcal{Z} \times \mathcal{Z} \to [0, \infty)$, defined as

$$W_\rho(\mu, \nu) := \inf_{\pi \in \Pi(\mu, \nu)} \int \rho(z, z') d\pi(z, z'), \tag{D.1}$$

where $\Pi(\mu, \nu)$ is the set of couplings of $\mu$ and $\nu$. We refer to the textbook Villani (2008) for more details.

**Lemma D.1.** *Consider $\mathcal{Z}$ measurable space. Let $Z_1, \dots, Z_m$ be $\mathcal{Z}$-valued random variables with natural filtration $\mathcal{F}_i := \sigma(Z_1, \dots, Z_i)$. For each $i$, assume there is $Z_i'$ such that*

$$Z_i' \sim Law(Z_i \mid \mathcal{F}_{i-1}), \quad Z_i' \perp\!\!\!\perp Z_i \mid \mathcal{F}_{i-1}. \tag{D.2}$$

*Let $g \colon \mathcal{Z}^m \to \mathbb{R}$ be measurable and coordinate-wise Lipschitz with respect to cost functions $\rho_i \colon \mathcal{Z} \times \mathcal{Z} \to [0, \infty)$ such that $\rho_i(z_i, z_i) = 0$, with constants $L_i \geq 0$: for any $z, z' \in \mathcal{Z}^m$ differing only in the $i$-th coordinate,*

$$|g(z) - g(z')| \leq L_i \rho_i(z_i, z_i'). \tag{D.3}$$

*With $W_{\rho_j}(\cdot, \cdot)$ the Wasserstein-1 distance with respect to $\rho_j$, define, for $i < j$,*

$$\delta_{i,j}(z_{1:i}, z_i') = W_{\rho_j}(Law(Z_j \mid Z_{1:i} = z_{1:i}), Law(Z_j \mid Z_{1:i-1} = z_{1:i-1}, Z_i = z_i')). \tag{D.4}$$

*for $i \in \{1, \dots, m\}$,*

$$\left| \mathbb{E}[g(Z_{1:m}) \mid \mathcal{F}_i] - \mathbb{E}[g(Z_{1:i-1}, Z_i', Z_{i+1:m}) \mid \mathcal{F}_{i-1}, Z_i'] \right| \leq L_i \rho_i(Z_i, Z_i') + \sum_{j=i+1}^{m} L_j \delta_{i,j}(Z_{1:i}, Z_i') \tag{D.5}$$

*Proof.* Fix $i \in \{1, \dots, m\}$. We condition on $\mathcal{F}_{i-1}$. Let $u := Z_i$ and $u' := Z_i'$. Not to overburden notations, all expectations and probabilities in the following are conditional on $\mathcal{F}_{i-1}, Z_i = u, Z_i' = u'$. Define the tail functions

$$\psi(z_{i+1:m}) := g(Z_{1:(i-1)}, u, z_{i+1:m}), \tag{D.6}$$
$$\psi'(z_{i+1:m}) := g(Z_{1:(i-1)}, u', z_{i+1:m}). \tag{D.7}$$

Denote $Z_{(i+1):m} \sim Law(Z_{(i+1):m} \mid \mathcal{F}_{i-1}, Z_i = u)$ and $Z_{(i+1):m}' \sim Law(Z_{(i+1):m} \mid \mathcal{F}_{i-1}, Z_i = u')$. We decompose

$$\left| \mathbb{E}[g(Z_{1:m})] - \mathbb{E}[g(Z_{1:(i-1)}, Z_{i:m}')] \right| \tag{D.8}$$
$$= \left| \mathbb{E}[\psi(Z_{(i+1):m})] - \mathbb{E}[\psi'(Z_{(i+1):m}')] \right| \tag{D.9}$$
$$\leq \mathbb{E}\left[ \left| \psi(Z_{(i+1):m}) - \psi'(Z_{(i+1):m}) \right| \right] + \left| \mathbb{E}\left[ \psi'(Z_{(i+1):m}) \right] - \mathbb{E}\left[ \psi'(Z_{(i+1):m}') \right] \right|. \tag{D.10}$$

---

[3]This is a slight abuse of terminology, since the Wasserstein-1 distance is usually defined for metric spaces, while we only assume $\rho$ to be a cost function. However, this slight abuse of terminology will not cause any confusion in the following.

We bound the two terms separately.

By the coordinate-wise Lipschitz condition at $i$,

$$\mathbb{E}_P\left[|\psi(Z_{(i+1):m}) - \psi'(Z_{(i+1):m})|\right] \le L_i \rho_i(u, u') = L_i \rho_i(Z_i, Z'_i). \tag{D.11}$$

We write the following telescoping decomposition:

$$\left|\mathbb{E}\left[\psi'(Z_{(i+1):m})\right] - \mathbb{E}\left[\psi'(Z'_{(i+1):m})\right]\right| \le \sum_{j=i}^{m-1} \left|\mathbb{E}\left[\psi'(Z'_{(i+1):j}, Z_{(j+1):m})\right] - \mathbb{E}\left[\psi'(Z'_{(i+1):(j+1)}, Z_{(j+1):m})\right]\right|. \tag{D.12}$$

By the definition of the Wasserstein-1 distance, there exists a coupling of $(Z_{j+1}, Z'_{j+1})$ such that

$$\mathbb{E}\left[\rho_{j+1}(Z_{j+1}, Z'_{j+1}) \,\Big|\, \mathcal{F}_i, Z'_i\right] = W_{\rho_{j+1}}(\mathrm{Law}(Z_{j+1} \mid \mathcal{F}_i), \mathrm{Law}(Z_{j+1} \mid \mathcal{F}_{i-1}, Z'_i)) \le \delta_{i,j+1}(Z_{1:i-1}, Z'_i). \tag{D.13}$$

We obtain a bound on the increment at coordinate $j$ by combining the coupling with the coordinate-wise Lipschitz condition at $j$:

$$\left|\mathbb{E}\left[\psi'(Z'_{(i+1):j}, Z_{(j+1):m})\right] - \mathbb{E}\left[\psi'(Z'_{(i+1):(j+1)}, Z_{(j+1):m})\right]\right| \tag{D.14}$$

$$\le \mathbb{E}\left[|\psi'(Z'_{(i+1):j}, Z_{(j+1):m}) - \psi'(Z'_{(i+1):(j+1)}, Z_{(j+1):m})|\right] \tag{D.15}$$

$$\le L_{j+1} \mathbb{E}\left[\rho_{j+1}(Z_{j+1}, Z'_{j+1})\right] \tag{D.16}$$

$$= L_{j+1} W_{\rho_{j+1}}(\mathrm{Law}(Z_{j+1} \mid \mathcal{F}_i), \mathrm{Law}(Z_{j+1} \mid \mathcal{F}_{i-1}, Z'_i)) = L_{j+1}\delta_{i,j+1}(Z_{1:i}, Z'_i). \tag{D.17}$$

Combining the above estimates gives

$$\left|\mathbb{E}\left[\psi'(Z_{(i+1):m})\right] - \mathbb{E}\left[\psi'(Z'_{(i+1):m})\right]\right| \le \sum_{j=i}^{m-1} L_{j+1}\delta_{i,j+1}(Z_{1:i}, Z'_i). \tag{D.18}$$

which yields the desired result. ∎

We now state a classic convex domination lemma which is a slight variant of Ledoux & Talagrand (2013, Lem. 4.6).

**Lemma D.2** (Convex domination)**.** *Consider $X, Z$ a zero-mean symmetric random variables such that*

$$\mathbb{P}(|X| > t) \le C \, \mathbb{P}(|Z| > t), \tag{D.19}$$

*for some $C > 0$ and all $t > 0$.*

*Then, for any convex function $h \colon \mathbb{R} \to \mathbb{R}$,*

$$\mathbb{E}[h(X)] \le \mathbb{E}[h(CZ)]. \tag{D.20}$$

*Proof.* Let $\delta \sim \mathrm{Bernoulli}(1/C)$ be independent of $(X, Z)$. Then, for all $t > 0$, $\mathbb{P}(|Z| > t) \ge \frac{1}{C} \, \mathbb{P}(|X| > t) = \mathbb{P}(|\delta X| > t)$. Hence $|\delta X|$ is stochastically dominated by $|Z|$ and we may construct a coupling such that

$$|\delta X| \le |Z| \qquad \text{a.s.} \tag{D.21}$$

Since $X$ is symmetric, we may write in distribution $X \stackrel{d}{=} \varepsilon |X|$ where $\varepsilon$ is a Rademacher variable independent of $|X|$. Likewise, $Z \stackrel{d}{=} \varepsilon' |Z|$ with an independent Rademacher $\varepsilon'$.

Condition on $(\delta, X, Z)$ and define

$$\Phi(a) := \mathbb{E}\left[h(a \, \varepsilon \, |Z|) \mid \delta, X, Z\right], \qquad a \in [-1, 1]. \tag{D.22}$$

The map $a \mapsto \Phi(a)$ is convex (as an average of convex functions). By convexity, its maximum on $[-1, 1]$ is attained at an extreme point $\{-1, 1\}$. On the coupling where (D.21) holds, define

$$a := \begin{cases} \dfrac{\delta |X|}{|Z|}, & \text{if } Z \neq 0, \\ 0, & \text{if } Z = 0, \end{cases} \tag{D.23}$$

so that $a \in [-1, 1]$ almost surely thanks to $|X| \leq |\delta Z|$. Therefore,

$$\mathbb{E}\big[ h(\varepsilon \,|X| \,\delta) \,\big|\, \delta, X, Z \big] \;=\; \Phi(a) \;\leq\; \max\{\Phi(-1), \Phi(1)\} \;=\; \mathbb{E}\big[ h(\varepsilon \,|Z|) \,\big|\, \delta, |X|, Z \big]. \tag{D.24}$$

Taking expectations and using $X \overset{d}{=} \varepsilon |X|$ and $Z \overset{d}{=} \varepsilon |Z|$,

$$\mathbb{E}[h(\delta X)] \;\leq\; \mathbb{E}[h(Z)]. \tag{D.25}$$

Since $h$ is convex and $\mathbb{E}[\delta \mid X, Z] = 1/C$, we have, by Jensen's inequality,

$$\mathbb{E}[h(X/C)] \;=\; \mathbb{E}\big[ h\big( \mathbb{E}[\delta X \mid X, Z] \big) \big] \;\leq\; \mathbb{E}\big[ \mathbb{E}[h(\delta X) \mid X, Z] \big] \;=\; \mathbb{E}[h(\delta X)] \leq \mathbb{E}[h(Z)], \tag{D.26}$$

Finally, apply the previous inequality with the convex function $u \mapsto h(Cu)$ to obtain

$$\mathbb{E}[h(X)] \;=\; \mathbb{E}[h(C \cdot (X/C))] \;\leq\; \mathbb{E}[h(CZ)]. $$

This is exactly the desired bound.

∎

We now state a fact of subGaussian random variables, which can be found in Wainwright (2019, Thm. 2.6) for instance.

**Lemma D.3** (Convex domination). *Consider $X$ a zero-mean real-valued $\sigma^2$-sub-Gaussian random variable, which is, in addition, symmetric, i.e., $X \overset{d}{=} -X$. Then, for $Z \sim \mathcal{N}(0, \sigma^2)$,*

$$\mathbb{P}(|X| > t) \leq 8\, \mathbb{P}(|Z| > t). \tag{D.27}$$

**Lemma D.4** (Causal symmetrization). *Let $m \in \mathbb{N}$ and $(\mathcal{Z}, \mathcal{A})$ be a standard Borel measurable space. Let $Z_1, \ldots, Z_m$ be $\mathcal{Z}$-valued random with natural filtration $(\mathcal{F}_i)_{i=0,\ldots,m}$ Let $h : \mathbb{R} \to \mathbb{R}$ be convex.*

*Consider $g : \mathcal{Z}^m \to \mathbb{R}$ be measurable. Set $S := g(Z_1, \ldots, Z_m)$. For each $i \in \{1, \ldots, m\}$, assume there exists a conditionally independent resample*

$$Z_i' \sim \text{Law}(Z_i \mid \mathcal{F}_{i-1}), \quad Z_i' \perp\!\!\!\perp Z_i \mid \mathcal{F}_{i-1}. \tag{D.28}$$

*Let $\varepsilon_{1:m}, \varepsilon'_{1:m}$ be independent Rademacher variables, independent of all $Z, Z'$ and $\mathcal{F}_m$.*

*Assume there exist measurable functions $c_i : \mathcal{Z} \times \mathcal{Z} \to [0, \infty)$, $d_i : \mathcal{Z} \to [0, \infty)$ and $J \subset \{1, \ldots, m\}$ such that, the following conditions hold:*

*(i) For any $i$, there exists $j(i) \in J$, such that, for any $z_{1:i-1} \in \mathcal{Z}^{i-1}$ and $z_i, z_i' \in \mathcal{Z}$,*

$$\big| \mathbb{E}[S \mid Z_{1:i} = z_{1:i}] - \mathbb{E}[S \mid Z_{1:i-1} = z_{1:i-1}, Z_i = z_i'] \big| \leq c_i(z_i, z_i') + d_i(z_{j(i)}) \mathbb{1}\{i \notin J\}. \tag{D.29}$$

*(ii) For any $i \notin J$, $\varepsilon_i c_i(Z_i, Z_i')$ is $\sigma_i^2$-sub-Gaussian conditionally on $\mathcal{F}_{i-1}$.*

*(iii) For any $j \in J$, $Z_j$ is independent of $\mathcal{F}_{j-1}$.*

*Then, there are Gausssian random variables $G_j, G_j' \sim \mathcal{N}(0, 8\sigma_j^2)$ independent and independent of all $Z, Z', \varepsilon, \mathcal{F}_m$ such that*

$$\mathbb{E}[h(S - \mathbb{E}[S])] \leq \mathbb{E}\left[ h\left( \sum_{i \notin J} \text{Sym}_{j(i)}\big(\varepsilon_i(|G_i| + d_i(Z_{j(i)}))\big) + \sum_{j \in J} \varepsilon_j c_j(Z_j, Z_j') \right) \right], \tag{D.30}$$

*where we use the notation:*

$$\text{Sym}_{j(i)}\big(\varepsilon_i(|G_i| + d_i(Z_{j(i)}))\big) := \varepsilon_{j(i)}\Big( \varepsilon_i(|G_i| + d_i(Z_{j(i)})) - \varepsilon_i'(|G_i'| + d_i(Z_{j(i)}')) \Big). \tag{D.31}$$

*Proof.* Define $\mathcal{G} = \sigma(\varepsilon_{1:m}, G_{1:m})$.

We show the result by induction on $k$: our goal is to show that, for any $k \in \{0, \ldots, m\}$,

$$\mathbb{E}[h(S - \mathbb{E}[S])] \leq \mathbb{E}\Bigg[ h\Big( \sum_{\substack{i \notin J \\ i \geq k+1}} \big(\mathbb{1}\{j(i) \leq k\}\varepsilon_i(|G_i| + d_i(Z_{j(i)})) + \mathbb{1}\{j(i) \geq k+1\} \operatorname{Sym}_{j(i)}\big(\varepsilon_i(|G_i| + d_i(Z_{j(i)}))\big)\big) \tag{D.32}$$

$$+ \sum_{\substack{i \in J \\ i \geq k+1}} \varepsilon_i c_i(Z_i, Z_i') + \mathbb{E}[S \mid Z_{1:k}] - \mathbb{E}[S]\Big)\Bigg], \tag{D.33}$$

where $G_i, G_i' \sim \mathcal{N}(0, 8\sigma_i^2)$ are independent and independent of all $Z, Z', \varepsilon, \varepsilon', \mathcal{F}_m$. (D.33) holds trivially for $k = m$. We now show that if it holds for some $k \in \{1, \ldots, m\}$, then it also holds for $k - 1$.

Note that we can rewrite

$$\sum_{\substack{i \notin J \\ i \geq k+1}} \big(\mathbb{1}\{j(i) \leq k\}\varepsilon_i(|G_i| + d_i(Z_{j(i)})) + \mathbb{1}\{j(i) \geq k+1\} \operatorname{Sym}_{j(i)}\big(\varepsilon_i(|G_i| + d_i(Z_{j(i)}))\big)\big) \tag{D.34}$$

$$+ \sum_{\substack{i \in J \\ i \geq k+1}} \varepsilon_i c_i(Z_i, Z_i') \tag{D.35}$$

$$= \underbrace{\sum_{\substack{i \notin J \\ i \geq k+1}} \mathbb{1}\{j(i) \geq k+1\} \operatorname{Sym}_{j(i)}\big(\varepsilon_i(|G_i| + d_i(Z_{j(i)}))\big) + \sum_{\substack{i \in J \\ i \geq k+1}} \varepsilon_i c_i(Z_i, Z_i')}_{=:Y_{\perp\!\!\!\perp}} \tag{D.36}$$

$$+ \underbrace{\sum_{\substack{i \notin J \\ i \geq k+1}} \mathbb{1}\{j(i) \leq k\}\varepsilon_i\big(|G_i| + d_i(Z_{j(i)})\big)}_{=:Y_k} \tag{D.37}$$

$$= Y_{\perp\!\!\!\perp} + Y_k, \tag{D.38}$$

where $Y_{\perp\!\!\!\perp}$ is independent of $\mathcal{F}_k$ and $Y_k$ is $\mathcal{F}_k$-measurable. More precisely, we show that

$$\mathbb{E}[h(Y_{\perp\!\!\!\perp} + Y_k + \mathbb{E}[S \mid Z_{1:k}] - \mathbb{E}[S]) \mid Y_{\perp\!\!\!\perp}] \tag{D.39}$$

$$\leq \mathbb{E}\Big[ h\big(Y_{\perp\!\!\!\perp} + Y_{k-1} + \mathbb{1}\{k \notin J\}\varepsilon_k(|G_k| + d_k(Z_{j(k)})) \tag{D.40}$$

$$+ \mathbb{1}\{k \in J\}(\varepsilon_k c_k(Z_k, Z_k') \tag{D.41}$$

$$+ \sum_{\substack{i \notin J \\ i \geq k+1 \\ j(i)=k}} \operatorname{Sym}_k(\varepsilon_i(|G_i| + d_i(Z_k)))\big) \mathbb{E}[S \mid Z_{1:k-1}] - \mathbb{E}[S]) \mid Y_{\perp\!\!\!\perp}], , \tag{D.42}$$

with $Y_{k-1} := \sum_{i \notin J, i \geq k+1} \varepsilon_i \mathbb{1}\{j(i) \leq k-1\}(|G_i| + d_i(Z_{j(i)}))$, which will imply the induction step (D.33) with $k \leftarrow k - 1$ by taking expectations over $Y_{\perp\!\!\!\perp}$. Since $Y_{\perp\!\!\!\perp}$ is considered constant in (D.42), we may assume without loss of generality that $Y_{\perp\!\!\!\perp} = 0$, at the potential cost of replacing $h$ by $h(\cdot + Y_{\perp\!\!\!\perp})$, which is still convex. Therefore, it suffices to show

$$\mathbb{E}[h(Y_k + \mathbb{E}[S \mid Z_{1:k}] - \mathbb{E}[S]) \mid Y_{\perp\!\!\!\perp}] \tag{D.43}$$

$$\leq \mathbb{E}\Big[ h\big(Y_{k-1} + \mathbb{1}\{k \notin J\}\varepsilon_k(|G_k| + d_k(Z_{j(k)})) \tag{D.44}$$

$$+ \mathbb{1}\{k \in J\}(\varepsilon_k c_k(Z_k, Z_k') \tag{D.45}$$

$$+ \sum_{\substack{i \notin J \\ i \geq k+1 \\ j(i)=k}} \operatorname{Sym}_k(\varepsilon_i(|G_i| + d_i(Z_k)))\big) \mathbb{E}[S \mid Z_{1:k-1}] - \mathbb{E}[S]) \mid Y_{\perp\!\!\!\perp}], , \tag{D.46}$$

We first consider the case of $k \notin J$. Define $\Phi(z_{1:k}) := \mathbb{E}[S \mid Z_{1:k} = z_{1:k}]$. We rewrite the right-hand side (RHS) of (D.46) as

$$\mathbb{E}[h(Y_k + \mathbb{E}[S \mid Z_{1:k}] - \mathbb{E}[S]) \mid Y_{\perp\!\!\!\perp}] \tag{D.47}$$

$$= \mathbb{E}\big[h\big(Y_k + \Phi(Z_{1:k}) - \mathbb{E}\big[\Phi(Z_{1:k-1}, Z'_k)\,\big|\,Z_{1:k-1}\big] + \mathbb{E}[S\,|\,Z_{1:k-1}] - \mathbb{E}[S]\big)|Y_{\perp\!\!\!\perp}\big] \tag{D.48}$$

$$= \mathbb{E}\big[h\big(Y_k + \mathbb{E}\big[\Phi(Z_{1:k}) - \Phi(Z_{1:k-1}, Z'_k)\,\big|\,Z_{1:k}\big] + \mathbb{E}[S\,|\,Z_{1:k-1}] - \mathbb{E}[S]\big)|Y_{\perp\!\!\!\perp}\big] \tag{D.49}$$

$$= \mathbb{E}\big[h\big(Y_k + \mathbb{E}\big[\Phi(Z_{1:k}) - \Phi(Z_{1:k-1}, Z'_k)\,\big|\,Z_{1:k}, \mathcal{G}\big] + \mathbb{E}[S\,|\,Z_{1:k-1}] - \mathbb{E}[S]\big)|Y_{\perp\!\!\!\perp}\big] \tag{D.50}$$

$$\tag{D.51}$$

where we used the fact that $\mathbb{E}[S\,|\,Z_{1:k-1}] = \mathbb{E}[\Phi(Z_{1:k-1}, Z'_k)\,|\,Z_{1:k-1}] = \mathbb{E}[\Phi(Z_{1:k_1}, Z'_k)\,|\,Z_{1:k}] = \mathbb{E}[\Phi(Z_{1:k-1}, Z'_k)\,|\,Z_{1:k}, \mathcal{G}]$, since $Z'_k \sim \mathrm{Law}(Z_k\,|\,Z_{1:k-1})$ and $Z'_k \perp\!\!\!\perp Z_k\,|\,Z_{1:k-1}$ and $\mathcal{G}$ is independent of all $Z, Z'$. Since both $Y_k$ and $\mathbb{E}[S\,|\,Z_{1:k-1}] - \mathbb{E}[S]$ are $\sigma(\mathcal{F}_k, \mathcal{G})$-measurable, by Jensen's inequality (convexity of $h$) applied to the conditional expectation w.r.t. $Z_{1:k}, \mathcal{G}$, we have

$$\mathbb{E}[h(Y_k + \mathbb{E}[S\,|\,Z_{1:k}] - \mathbb{E}[S])\,|\,Y_{\perp\!\!\!\perp}] \tag{D.52}$$

$$\leq \mathbb{E}\big[h\big(Y_k + \Phi(Z_{1:k}) - \Phi(Z_{1:k-1}, Z'_k) + \mathbb{E}[S\,|\,Z_{1:k-1}] - \mathbb{E}[S]\big)\,\big|\,Y_{\perp\!\!\!\perp}\big]. \tag{D.53}$$

Since $k \notin J$, then $Y_k$ is $\sigma(\mathcal{F}_{k-1}, \mathcal{G})$-measurable. The following argument will now be made conditionally on $\mathcal{F}_{k-1}, \mathcal{G}, Y_{\perp\!\!\!\perp}$.

We have that $\Phi(Z_{1:k}) - \Phi(Z_{1:k-1}, Z'_k)$ is symmetric. Moreover, since $|\Phi(Z_{1:k}) - \Phi(Z_{1:k-1}, Z'_k)| \leq c_k(Z_k, Z'_k) + d_k(Z_{j(k)})$ by assumption (i), we have that, for any $t > 0$,

$$\mathbb{P}\big(|\Phi(Z_{1:k}) - \Phi(Z_{1:k-1}, Z'_k)| > t\,\big|\,\mathcal{F}_{k-1}, \mathcal{G}, Y_{\perp\!\!\!\perp}\big) \tag{D.54}$$

$$\leq \mathbb{P}\big(c_k(Z_k, Z'_k) + d_k(Z_{j(k)}) > t\,\big|\,\mathcal{F}_{k-1}, \mathcal{G}, Y_{\perp\!\!\!\perp}\big) \tag{D.55}$$

$$\leq \mathbb{P}\big(c_k(Z_k, Z'_k) > t - d_k(Z_{j(k)})\,\big|\,\mathcal{F}_{k-1}, \mathcal{G}, Y_{\perp\!\!\!\perp}\big) \tag{D.56}$$

$$\leq 8\,\mathbb{P}\big(|G_k| > t - d_k(Z_{j(k)})\,\big|\,\mathcal{F}_{k-1}, \mathcal{G}, Y_{\perp\!\!\!\perp}\big), \tag{D.57}$$

where we used that $\varepsilon_k c_k(Z_k, Z'_k)$ is $\sigma_k^2$-sub-Gaussian conditionally on $\mathcal{F}_{k-1}$ by assumption (ii) and Lem. D.3. Therefore, we can apply Lem. D.2 with $X \leftarrow \Phi(Z_{1:k}) - \Phi(Z_{1:k-1}, Z'_k)$ and $Z \leftarrow \varepsilon_k(|G_k| + d_k(Z_{j(k)}))$ with $C = 8$ conditionally on $\mathcal{F}_{k-1}, Y_{\perp\!\!\!\perp}$ to obtain

$$\mathbb{E}[h(Y_k + \mathbb{E}[S\,|\,Z_{1:k}] - \mathbb{E}[S])\,|\,Y_{\perp\!\!\!\perp}] \tag{D.58}$$

$$\leq \mathbb{E}\big[h\big(Y_k + \varepsilon_k(|G_k| + d_k(Z_{j(k)})) + \mathbb{E}[S\,|\,Z_{1:k-1}] - \mathbb{E}[S]\big)\,\big|\,Y_{\perp\!\!\!\perp}\big], \tag{D.59}$$

which is (D.46) in the case $k \notin J$.

For the case $k \in J$, we use a similar argument. We now have, as before,

$$\mathbb{E}[S\,|\,Z_{1:k-1}] = \mathbb{E}[\Phi(Z_{1:k-1}, Z'_k)\,|\,Z_{1:k-1}] \tag{D.60}$$

$$= \mathbb{E}\big[\Phi(Z_{1:k-1}, Z'_k) + \sum_{\substack{i \notin J \\ i \geq k+1 \\ j(i)=k}} \varepsilon'_i\big(|G'_i| + d_i(Z_k)\big)\,|\,Z_{1:k-1}\big] \tag{D.61}$$

$$= \mathbb{E}\big[\Phi(Z_{1:k-1}, Z'_k) + \sum_{\substack{i \notin J \\ i \geq k+1 \\ j(i)=k}} \varepsilon'_i\big(|G'_i| + d_i(Z_k)\big)\,|\,Z_{1:k}, \mathcal{G}\big], \tag{D.62}$$

by construction.

Since both $Y_k$ and $\mathbb{E}[S\,|\,Z_{1:k-1}] - \mathbb{E}[S]$ are $\sigma(\mathcal{F}_k, \mathcal{G})$-measurable, by Jensen's inequality (convexity of $h$) applied to the conditional expectation w.r.t. $Z_{1:k}, \mathcal{G}$, we have

$$\mathbb{E}[h(Y_k + \mathbb{E}[S\,|\,Z_{1:k}] - \mathbb{E}[S])\,|\,Y_{\perp\!\!\!\perp}] \tag{D.63}$$

$$\leq \mathbb{E}\Big[h\Big(Y_k + \Phi(Z_{1:k}) - \Phi(Z_{1:k-1}, Z'_k) - \sum_{\substack{i \notin J \\ i \geq k+1 \\ j(i)=k}} \varepsilon'_i\big(|G'_i| + d_i(Z_k)\big) + \mathbb{E}[S\,|\,Z_{1:k-1}] - \mathbb{E}[S]\Big)\Big|Y_{\perp\!\!\!\perp}\Big]. \tag{D.64}$$

We write $Y_k$ as

$$Y_k = Y_{k-1} + \sum_{\substack{i \notin J \\ i \geq k+1 \\ j(i)=k}} \varepsilon_i(|G_i| + d_i(Z_k)), \tag{D.65}$$

where $Y_{k-1}$ is $\sigma(\mathcal{F}_{k-1}, \mathcal{G})$-measurable and obtain,

$$\mathbb{E}[h(Y_{k-1} + \mathbb{E}[S \mid Z_{1:k}] - \mathbb{E}[S]) \mid Y_\perp] \tag{D.66}$$

$$\leq \mathbb{E}\left[h\left(Y_{k-1} + \Phi(Z_{1:k}) - \Phi(Z_{1:k-1}, Z'_k) + \sum_{\substack{i \notin J \\ i \geq k+1 \\ j(i)=k}} \varepsilon_i(|G_i| + d_i(Z_k)) - \varepsilon'_i(|G'_i| + d_i(Z_k)) + \mathbb{E}[S \mid Z_{1:k-1}] - \mathbb{E}[S]\right) \middle| Y_\perp\right]. \tag{D.67}$$

We now make the following domination argument conditionally on $\mathcal{F}_{k-1}, Y_{k-1}, Y_\perp$. The random variable

$$\Phi(Z_{1:k}) - \Phi(Z_{1:k-1}, Z'_k) + \sum_{\substack{i \notin J \\ i \geq k+1 \\ j(i)=k}} \varepsilon_i(|G_i| + d_i(Z_k)) - \varepsilon'_i(|G'_i| + d_i(Z_k)) \tag{D.68}$$

is symmetric and, by assumption (i) and the triangle inequality, bounded in absolute value by

$$\left| \varepsilon_k c_k(Z_k, Z'_k) + \sum_{\substack{i \notin J \\ i \geq k+1 \\ j(i)=k}} \mathrm{Sym}_k(\varepsilon_i(|G_i| + d_i(Z_k))) \right|. \tag{D.69}$$

Applying Lem. D.2 conditionally on $\mathcal{F}_{k-1}, Y_{k-1}, Y_\perp$ with $C = 1$ (hence no constant appears) yields the desired result.

■

We can now combine Lem. D.1 and Lem. D.4 to obtain the main moment bound of this section.

**Theorem D.1** (Causal symmetrization). *Let $m \in \mathbb{N}$ and $(\mathcal{Z}, \mathcal{A})$ be a standard Borel measurable space. Let $Z_1, \ldots, Z_m$ be $\mathcal{Z}$-valued random with natural filtration $(\mathcal{F}_i)_{i=0,\ldots,m}$. Let $h \colon \mathbb{R} \to \mathbb{R}$ be convex.*

*Let $g \colon \mathcal{Z}^m \to \mathbb{R}$ be measurable and coordinate-wise Lipschitz with respect to cost functions $\rho_i \colon \mathcal{Z} \times \mathcal{Z} \to [0, \infty)$ such that $\rho_i(z_i, z_i) = 0$ with constants $L_i \geq 0$: for any $z, z' \in \mathcal{Z}^m$ differing only in the $i$-th coordinate,*

$$|g(z) - g(z')| \leq L_i \rho_i(z_i, z'_i). \tag{D.70}$$

*Set $S := g(Z_1, \ldots, Z_m)$ and*

*For each $i \in \{1, \ldots, m\}$, assume there exists a conditionally independent resample*

$$Z'_i \sim \mathrm{Law}(Z_i \mid \mathcal{F}_{i-1}), \quad Z'_i \perp\!\!\!\perp Z_i \mid \mathcal{F}_{i-1}. \tag{D.71}$$

*Let $\varepsilon_{1:m}, \varepsilon'_{1:m}$ be independent Rademacher variables, independent of all $Z, Z'$ and $\mathcal{F}_m$.*

*Assume there exist constants $c_{ik} \geq 0$, measurable functions $d_{ik} \colon \mathcal{Z} \to [0, \infty)$ and $J \subset \{1, \ldots, m\}$ such that, the following conditions hold:*

*(i) For any $i < k$, there exists $j(i) \in J$, such that, for any $z_{1:i-1} \in \mathcal{Z}^{i-1}$ and $z_i, z'_i \in \mathcal{Z}$,*

$$W_{\rho_k}\big(\mathrm{Law}(Z_k \mid Z_{1:i} = z_{1:i}), \mathrm{Law}(Z_k \mid Z_{1:i-1} = z_{1:i-1}, Z_i = z'_i)\big) \leq c_{ik}\rho_i(z_i, z'_i) + d_{ik}(z_{j(i)}) \mathbb{1}\{i \notin J\}. \tag{D.72}$$

*(ii) For any $i \notin J$, $\varepsilon_i \rho_i(Z_i, Z'_i)$ is $\sigma_i^2$-sub-Gaussian conditionally on $\mathcal{F}_{i-1}$.*

*(iii) For any $j \in J$, $Z_j$ is independent of $\mathcal{F}_{j-1}$.*

*Then, there are Gausssian random variables $G_j, G'_j \sim \mathcal{N}(0, 8\sigma_j^2)$ independent and independent of all $Z, Z', \varepsilon, \mathcal{F}_m$ such that*

$$\mathbb{E}[h(S - \mathbb{E}[S])] \tag{D.73}$$

$$\leq \mathbb{E}\left[h\left(\sum_{i \notin J} \mathrm{Sym}_{j(i)}\left(\varepsilon_i\left(L_i|G_i| + \sum_{k>i} L_k c_{ik}|G_i| + L_k d_{ik}(Z_{j(i)})\right)\right) + \sum_{j \in J} \varepsilon_j\left(L_j \rho_j(Z_j, Z'_j) + \sum_{k>j} L_k c_{jk} \rho_j(Z_j, Z'_j)\right)\right)\right], \tag{D.74}$$

*where we use the notation:*

$$\mathrm{Sym}_{j(i)}\left(\varepsilon_i\left(L_i|G_i| + \sum_{k>i} L_k c_{ik}|G_i| + L_k d_{ik}(Z_{j(i)})\right)\right) := \tag{D.75}$$

$$\varepsilon_{j(i)}\left(\varepsilon_i\left(L_i|G_i| + \sum_{k>i} L_k c_{ik}|G_i| + L_k d_{ik}(Z_{j(i)})\right) - \varepsilon'_i\left(L_i|G'_i| + \sum_{k>i} L_k c_{ik}|G'_i| + L_k d_{ik}(Z_{j(i)})\right)\right). \tag{D.76}$$

### D.2. Technical lemmas

We will make use of the following elementary lemma.

**Lemma D.5.** *Let $Z$ be a real-valued random variable. Assume there exist $c \geq 1$, $f, g \colon \mathbb{R} \to \mathbb{R}_+$ non-decreasing and $p \geq 2$ integer such that, for any integer $q \in [2, p]$,*

$$\mathbb{E}[|Z|^q]^{1/q} \leq f(q) + c^{1/q} g(q) \tag{D.77}$$

*Then, for any $\delta \in (0, e^{-2}]$, with probability at least $1 - \delta$,*

$$|Z| \leq \begin{cases} ef(\log(1/\delta) + 1) + g(\log(1/\delta) + 1)e & \text{if } \delta \geq ce^{-p} \\ \frac{f(p) + c^{1/p}g(p)}{\delta^{1/p}} & \text{if } \delta < ce^{-p} . \end{cases} \tag{D.78}$$

*Proof.* By Markov's inequality, for any integer $q \in [2, p]$,

$$\mathbb{P}(|Z| \geq t) \leq \frac{\mathbb{E}[|Z|^q]}{t^q} \leq \left(\frac{f(q) + c^{1/q} g(q)}{t}\right)^q. \tag{D.79}$$

Setting the right-hand side to $\delta$ and solving for $t$ gives

$$t = \frac{f(q) + c^{1/q} g(q)}{\delta^{1/q}}, \tag{D.80}$$

If $\delta < ce^{-p}$, we can take $q = p$ to obtain the second case of the result. If $\delta \geq ce^{-p}$, we take $q$ the smallest integer such that $q \geq \log(c/\delta)$. Note that $q$ is in $[2, p]$ and $q \leq \log(c/\delta) + 2$.

Since $c \geq 1$ and $\delta \leq 1$, we have $\log(c/\delta) \geq 0$ and thus $\left(\frac{c}{\delta}\right)^{1/q} \leq \left(\frac{c}{\delta}\right)^{1/\log(c/\delta)} = e$. Plugging this into (D.80) gives the bound in the first case. ∎

We state the following lemma about norm-sub-Gaussian random vectors that will be useful later.

**Lemma D.6.** *Let $X \in \mathbb{R}^m$ satisfy the norm-sub-Gaussian tail condition of Jin et al. (2019): for any $\alpha \geq 0$,*

$$\mathbb{P}(\|X - \mathbb{E}[X]\| \geq \alpha) \leq 2\exp\left(-\frac{\alpha^2}{2\sigma^2}\right). \tag{D.81}$$

*Then, for $X'$ an i.i.d. copy of $X$ and $\varepsilon$ a Rademacher random variable independent of $X, X'$, the random variable $\varepsilon\|X - X'\|$ is sub-Gaussian with parameter at most $64\sigma^2$.*

*Proof.* For any $\alpha \geq 0$, by the triangle inequality and a union bound,

$$\mathbb{P}(\|X - X'\| \geq \alpha) \leq \mathbb{P}(\|X - \mathbb{E}[X]\| \geq \alpha/2) + \mathbb{P}(\|X' - \mathbb{E}[X]\| \geq \alpha/2) \leq 4 \exp\left(-\frac{\alpha^2}{8\sigma^2}\right). \tag{D.82}$$

Since $Z := \varepsilon\|X - X'\|$ is symmetric, this tail bound implies the scalar sub-Gaussian moment generating function bound

$$\log \mathbb{E}[e^{\lambda Z}] \leq \frac{64\sigma^2\lambda^2}{2}, \qquad \lambda \in \mathbb{R}, \tag{D.83}$$

by the standard tail-to-MGF conversion for symmetric random variables (see, e.g., Wainwright, 2019, Chap. 2). ∎

We will require the following chaining lemma for processes with $L^p$-Lipschitz increments. This result is a variant of the famous Dudley's entropy integral bound for sub-Gaussian processes, adapted to the $L^p$-Lipschitz setting.

This lemma is a direct consequence of the general chaining theory of Talagrand (2022) (see Talagrand (2022, Thm. B.2.3) with $\phi(x) = x^p$). Let us also mention Dirksen (2015) refined these ideas in the context of subpexponential processes while Latała & Tkocz (2015) further developed these tools for processes with heavier tails but still admitting a control over all moments. In our setting, the increments are assumed to be controlled only in $L^p$, which requires a different treatment of the maximal inequalities at each scale.

**Lemma D.7** (Dudley–type entropy integral under $L^p$ increments). *Let $(X_t)_{t \in T}$ be a real-valued process indexed by a pseudometric space $(T, d)$. Assume $T$ is totally bounded with diameter $\Delta := \mathrm{diam}_d(T) \in (0, \infty)$ and that for some $p > 1$ and $L > 0$,*

$$\|X_t - X_s\|_p \leq L\, d(t, s) \qquad \forall s, t \in T. \tag{D.84}$$

*Then*

$$\mathbb{E}\left[\sup_{s, t \in T} (X_t - X_s)\right] \leq C L \int_0^\Delta \left(\mathcal{N}(T, d, \varepsilon)\right)^{1/p} d\varepsilon, \tag{D.85}$$

*where $\mathcal{N}(T, d, \varepsilon)$ is the $\varepsilon$-covering number and $C < \infty$ is an absolute constant.*

### D.3. Concentration bounds for ICL

We now apply the moment symmetrization results to derive concentration bounds for ICL in the dependent data setting. These concentration bounds will then be translated into generalization bounds in the next subsection.

Let us recall ICL notations.

We denote by $\Theta \subset \mathbb{R}^d$ the space of tasks $\theta$ and by $\pi(\theta)$ the density of the pretraining task distribution. Given a task $\theta$, the data is generated according to a task-specific distribution with density $\mathrm{p}(\cdot \mid \theta)$. The training data is then generated by first sampling a task $\theta$ from the task distribution $\pi$, and then sampling data points $(x_t)_{t \geq 1}$ according to

$$x_{t+1} \sim \mathrm{p}_{t+1}(\cdot \mid x_{1:t}, \theta). \tag{D.86}$$

where $x_{1:t} = (x_1, \ldots, x_t)$.

Given a dataset of tasks $\theta_1, \ldots, \theta_N$ and associated samples $x_{1:T}^{(1)}, \ldots, x_{1:T}^{(N)}$, a model $f$ is trained by minimizing the next-sample prediction loss

$$\widehat{L}(f, (\theta_n, x_{1:T}^n)_{n \leq N}) = \frac{1}{NT} \sum_{n=1}^N \sum_{t=1}^T \ell_t(f(x_{1:t-1}^n), x_t^n), \tag{D.87}$$

where $\ell_t : \mathcal{X} \times \mathcal{X} \to [0, +\infty)$ is a loss function at step $t$.

We now provide a detailed version of Asm. 2.

**Assumption 5** (Weak dependence). We assume that there are deterministic coefficients $(A_t)_{t \geq 1}$ and $(B_{s,t})_{t \geq s \geq 1}$ such that, for any $t \geq s \geq 1$, $\theta, \theta' \in \Theta$, any $x_{1:(s-1)} \in \mathcal{X}^{s-1}$, and any $x_t, x_t' \in \mathcal{X}$,

$$W_1(\mathrm{p}_t(dx_t \mid \theta), \mathrm{p}_t(dx_t' \mid \theta')) \leq A_t \|\theta - \theta'\| \tag{D.88}$$

$$W_1(\mathrm{p}_t(dx_t \mid x_{1:s}, \theta), \mathrm{p}_t(dx_t' \mid x_{1:(s-1)}, x_s', \theta)) \leq B_{s,t} \|\theta\|. \tag{D.89}$$

In the second assumption, the Wasserstein distance between the conditional distributions of $x_t$ given $x_s$ and $x'_s$ is assumed to be controlled by the norm of the task $\theta$. This is a slight difference with Asm. 2 where we assumed a dependence on $1 + \|\theta\|$. This is however without loss of generality as we can always consider $\widetilde{\theta} = (1, \theta) \in \mathbb{R}^{d+1}$ and redefine the task distribution accordingly and this cosmetic change simplifies the presentation. We could also consider a dependence on $\|x_s - x'_s\|$, see Thm. D.1, but we omit this for simplicity.

We restate Asm. 1.

**Assumption 6** (Finite moments of the task distribution)**.** There exists $q \geq 2$ integer such that $\mathbb{E}[\|\theta\|^q] < +\infty$.

The next three assumptions are refined versions of Asm. 3. Our theory could be extended to more general assumptions on the distributions of sample, but, for simplicity, we will make the following norm-sub-Gaussian assumption on the data, conditionally on the past data and the task. Hence, this assumption does not restrict the task distribution in any way.

**Assumption 7** (Norm-sub-Gaussian data)**.** There exists $\sigma > 0$ such that, for any $t \geq 1$, $\theta \in \Theta$, and any $x_{1:(t-1)} \in \mathcal{X}^{t-1}$, $x_t \sim p_t(\cdot \mid x_{1:(t-1)}, \theta)$ satisfies the norm-sub-Gaussian tail condition, i.e.,, for any $\alpha \geq 0$,

$$\mathbb{P}_{x_t \sim p_t(\cdot \mid x_{1:(t-1)}, \theta)}\left(\left\|x_t - \mathbb{E}_{x_t \sim p_t(\cdot \mid x_{1:(t-1)}, \theta)}[x_t]\right\| \geq \alpha\right) \leq 2\exp\left(-\frac{\alpha^2}{2\sigma^2}\right). \tag{D.90}$$

**Assumption 8** (Lipschitz model and loss)**.** The models $f \in \mathcal{F}$ are uniformly Lipschitz in the following sense: there exists $L_T > 0$ such that, for any $f \in \mathcal{F}$, any $x_{1:T}, x'_t$,

$$\frac{1}{T}\sum_{s=1}^{T}\|f(x_{1:s-1}) - f(x_{1:t-1}, x'_t, x_{t+1:s-1})\| \leq L_T\|x_t - x'_t\|, \tag{D.91}$$

The losses $\ell_t$ are uniformly 1-Lipschitz: for any $t \geq 1$, any $x, x' \in \mathcal{X}$,

$$|\ell_t(x, x') - \ell_t(x, x')| \leq \|x - x'\|. \tag{D.92}$$

We will consider the following assumption on the function class $\mathcal{F}$.

**Assumption 9.** Assume that the hypothesis class $\mathcal{F}$ is bounded for w.r.t. some distance dist on $\mathcal{F}$ and that, the following extended Lipschitz condition holds: for any $f, f' \in \mathcal{F}$, any $x_{1:T}$, any $t \geq 1$, any $x'_t$, for any $f \in \mathcal{F}$, any $x_{1:T}, x'_t$,

$$\frac{1}{T}\sum_{s=1}^{T}\|f(x_{1:s-1}) - f(x_{1:t-1}, x'_t, x_{t+1:s-1}) - \left(f'(x_{1:s-1}) - f'(x_{1:t-1}, x'_t, x_{t+1:s-1})\right)\| \tag{D.93}$$

$$\leq M_T\|x_t - x'_t\| \operatorname{dist}(f, f'). \tag{D.94}$$

Note that Asm. 8 is implied of Asm. 9 when the constant function equal to zero is in $\mathcal{F}$ with $L_T = M_T \sup_{f \in \mathcal{F}} \operatorname{dist}(f, 0)$.

We denote by $\|X\|_h$ the $L^h$ norm of a random variable $X$, i.e., $\|X\|_h = (\mathbb{E}[\|X\|^h])^{1/h}$.

**Lemma D.8.** *For any $r \in [2, q]$ integer, under Asms. 5–8, we have*

$$\left\|\sup_{f \in \mathcal{F}}\left\{\mathbb{E}\left[\widehat{L}(f, (\theta_n, x_{1:T}^n)_{n \leq N})\right] - \widehat{L}(f, (\theta_n, x_{1:T}^n)_{n \leq N})\right\}\right. \tag{D.95}$$

$$\left. - \mathbb{E}\left[\sup_{f \in \mathcal{F}}\left\{\mathbb{E}\left[\widehat{L}(f, (\theta_n, x_{1:T}^n)_{n \leq N})\right] - \widehat{L}(f, (\theta_n, x_{1:T}^n)_{n \leq N})\right\}\right]\right\|_r \tag{D.96}$$

$$\leq c\sigma L_T\sqrt{\frac{Tr}{N}} \tag{D.97}$$

$$+ c\sqrt{r}\frac{L_T}{\sqrt{N}}\sqrt{\sum_{t=1}^{T}\left(\sum_{s>t} B_{t,s}\right)^2}\|\theta_1\|_2 + cr^{3/2}\frac{L_T}{N^{1-1/r}}\sqrt{\sum_{t=1}^{T}\left(\sum_{s>t} B_{t,s}\right)^2}\|\theta_1\|_q \tag{D.98}$$

$$+ c\sqrt{r}\frac{L_T}{\sqrt{N}}\left(\sum_{t=1}^{T} A_t\right)\|\theta_1 - \mathbb{E}[\theta_1]\|_2 + cr\frac{L_T}{N^{1-1/r}}\left(\sum_{t=1}^{T} A_t\right)\|\theta_1 - \mathbb{E}[\theta_1]\|_q, \tag{D.99}$$

*where $c > 0$ is a universal constant.*

*Proof.* We apply Thm. D.1 with

$$(Z_1, \ldots, Z_m) = (\theta_1, x_1^{(1)}, \ldots, x_T^{(1)}, \ldots, \theta_N, x_1^{(N)}, \ldots, x_T^{(N)}), \tag{D.100}$$

and

$$g(\theta_1, x_{1:T}^{(1)}, \ldots, \theta_N, x_{1:T}^{(N)}) \tag{D.101}$$

$$= \sup_{f \in \mathcal{F}} \left\{ \mathbb{E}\left[\widehat{L}(f, (\theta_n, x_{1:T}^n)_{n \leq N})\right] - \widehat{L}(f, (\theta_n, x_{1:T}^n)_{n \leq N}) \right\} \tag{D.102}$$

$$= \sup_{f \in \mathcal{F}} \frac{1}{NT} \left\{ \mathbb{E}\left[\sum_{n=1}^{N}\sum_{t=1}^{T} \ell_t(f(x_{1:t-1}^n), x_t^n)\right] - \sum_{n=1}^{N}\sum_{t=1}^{T} \ell_t(f(x_{1:t-1}^n), x_t^n) \right\}. \tag{D.103}$$

By Asm. 8, $g$ is coordinate-wise Lipschitz with respect to $x_t^n$ with constant $L_{N,T} := L_T/N$ and formally constant with respect to $\theta_n$.

By Lem. D.6 and Asm. 7, $\varepsilon_t^n \|x_t^n - x_t'^n\|$ is $64\sigma^2$-sub-Gaussian conditionally on $x_{1:(t-1)}, \theta_n$, for $\varepsilon_t^n$ a Rademacher variable independent of all data.

We now apply Thm. D.1 with $h(x) = |x|^r$ for $r$ integer such that $2 \leq r \leq q$ and $J$ corresponding to the indices of the tasks $\theta_1, \ldots, \theta_N$. We obtain that

$$\|f - \mathbb{E}[f]\|_r \tag{D.104}$$

$$\leq \left\| \sum_{n=1}^{N}\sum_{t=1}^{T} \mathrm{Sym}_n\left(\varepsilon_t^n\left(L_{N,T}|G_t^n| + \sum_{s>t} L_{N,T}B_{t,s}\|\theta_n\|\right)\right) + \sum_{n=1}^{N}\sum_{t=1}^{T} L_{N,T}\varepsilon_n A_t \|\theta_n - \theta_n'\| \right\|_r, \tag{D.105}$$

where

$$\mathrm{Sym}_n\left(\varepsilon_t^n\left(L_{N,T}|G_t^n| + \sum_{s>t} L_{N,T}B_{t,s}\|\theta_n\|\right)\right) := \tag{D.106}$$

$$\varepsilon_n\left(\varepsilon_t^n\left(L_{N,T}|G_t^n| + \sum_{s>t} L_{N,T}B_{t,s}\|\theta_n\|\right) - \varepsilon_t^{n\prime}\left(L_{N,T}|G_t^{n\prime}| + \sum_{s>t} L_{N,T}B_{t,s}\|\theta_n\|\right)\right), \tag{D.107}$$

and $G_t^n, G_t'^n \sim \mathcal{N}(0, 512\sigma^2)$ independent of all data and Rademacher variables.

Using Minkowski's inequality, we have

$$\|f - \mathbb{E}[f]\|_r \tag{D.108}$$

$$\leq \left\| \sum_{n=1}^{N} \varepsilon_n \sum_{t=1}^{T} L_{N,T}(\varepsilon_t^n |G_t^n| - \varepsilon_t^{n\prime}|G_t^{n\prime}|) \right\|_r \tag{D.109}$$

$$+ \left\| \sum_{n=1}^{N} \varepsilon_n\left(\|\theta_n\| \sum_{t=1}^{T} L_{N,T} \sum_{s>t} B_{t,s}\varepsilon_t^n - \|\theta_n'\| \sum_{t=1}^{T} L_{N,T} \sum_{s>t} B_{t,s}\varepsilon_t^{n\prime}\right) \right\|_r \tag{D.110}$$

$$+ \left\| \sum_{n=1}^{N} \varepsilon_n \|\theta_n - \theta_n'\| \sum_{t=1}^{T} L_{N,T} A_t \right\|_r. \tag{D.111}$$

We now bound each term (D.109)–(D.111) separately.

We begin with (D.109). By independence of the Rademacher variables and the Gaussian variables, we have that (D.109) can be rewritten as

$$(D.109) = \sqrt{2}L_{N,T} \left\| \sum_{n=1}^{N}\sum_{t=1}^{T} G_t^n \right\|_r \tag{D.112}$$

$$= 8\sigma L_{N,T}\sqrt{NT}\|G\|_r, \tag{D.113}$$

where $G \sim \mathcal{N}(0, 1)$. Using standard bounds on subGaussian random variables, we have that $\|G\|_r \leq c\sqrt{r}$ for some universal constant $c > 0$ (see e.g. Vershynin (2018, Chap. 2)). Hence, we have

$$(D.109) \leq c\sigma L_{N,T}\sqrt{NTr}, \tag{D.114}$$

for some universal constant $c > 0$.

We now turn to (D.110). By Boucheron et al. (2005, Thm. 15.11), applied to each independent and zero-mean term

$$\varepsilon_n\left(\|\theta_n\|\sum_{t=1}^{T}\varepsilon_t{}^n\sum_{s>t}B_{t,s} - \|\theta_n'\|\sum_{t=1}^{T}\varepsilon_t{}^{n'}\sum_{s>t}B_{t,s}\right), \tag{D.115}$$

we have

$$(D.110) \leq c\sqrt{r}L_{N,T}\sqrt{N}\left\|\|\theta_1\|\sum_{t=1}^{T}\varepsilon_t{}^1\sum_{s>t}B_{t,s} - \|\theta_1'\|\sum_{t=1}^{T}\varepsilon_t{}^{1'}\sum_{s>t}B_{t,s}\right\|_2 \tag{D.116}$$

$$+ \quad crL_{N,T}N^{1/r}\left\|\|\theta_1\|\sum_{t=1}^{T}\varepsilon_t{}^1\sum_{s>t}B_{t,s} - \|\theta_1'\|\sum_{t=1}^{T}\varepsilon_t{}^{1'}\sum_{s>t}B_{t,s}\right\|_r, \tag{D.117}$$

where $c > 0$ is a universal constant.

Using Minkowski's inequality again, we have

$$(D.110) \leq c\sqrt{r}L_{N,T}\sqrt{N}\left\|\|\theta_1\|\sum_{t=1}^{T}\varepsilon_t{}^1\sum_{s>t}B_{t,s}\right\|_2 \tag{D.118}$$

$$+ \quad crL_{N,T}N^{1/r}\left\|\|\theta_1\|\sum_{t=1}^{T}\varepsilon_t{}^1\sum_{s>t}B_{t,s}\right\|_r \tag{D.119}$$

$$\leq c\sqrt{r}L_{N,T}\sqrt{N}\|\theta_1\|_2\left\|\sum_{t=1}^{T}\varepsilon_t{}^1\sum_{s>t}B_{t,s}\right\|_2 \tag{D.120}$$

$$+ \quad crL_{N,T}N^{1/r}\|\theta_1\|_r\left\|\sum_{t=1}^{T}\varepsilon_t{}^1\sum_{s>t}B_{t,s}\right\|_r, \tag{D.121}$$

where we used that $\theta_1$ and $(\varepsilon_t{}^1)_{t\geq 1}$ are independent. Now, $\sum_{t=1}^{T}\varepsilon_t{}^1\sum_{s>t}B_{t,s}$ is a zero-mean sub-Gaussian random variable with parameter $\sum_{t=1}^{T}\left(\sum_{s>t}B_{t,s}\right)^2$ by Hoeffding's lemma (see e.g. Wainwright (2019, Exercise 2.4)) and we have, for some universal constant $c > 0$, for any integer $h$

$$\left\|\sum_{t=1}^{T}\varepsilon_t{}^1\sum_{s>t}B_{t,s}\right\|_h \leq c\sqrt{h}\left(\sum_{t=1}^{T}\left(\sum_{s>t}B_{t,s}\right)^2\right)^{1/2}. \tag{D.122}$$

Plugging this into (D.121) with $h = 2$ and $h = r$ gives

$$(D.110) \leq c\sqrt{r}L_{N,T}\sqrt{N}\sqrt{\sum_{t=1}^{T}\left(\sum_{s>t}B_{t,s}\right)^2}\|\theta_1\|_2 + cr^{3/2}L_{N,T}N^{1/r}\sqrt{\sum_{t=1}^{T}\left(\sum_{s>t}B_{t,s}\right)^2}\|\theta_1\|_r \tag{D.123}$$

$$\leq c\sqrt{r}L_{N,T}\sqrt{N}\sqrt{\sum_{t=1}^{T}\left(\sum_{s>t}B_{t,s}\right)^2}\|\theta_1\|_2 + cr^{3/2}L_{N,T}N^{1/r}\sqrt{\sum_{t=1}^{T}\left(\sum_{s>t}B_{t,s}\right)^2}\|\theta_1\|_q \tag{D.124}$$

$$\tag{D.125}$$

where we used that $r \leq q$ to obtain the last inequality.

Finally, we proceed similarly for (D.111). By Boucheron et al. (2005, Thm. 15.11) applied to each independent and zero-mean term

$$\varepsilon_n\|\theta_n - \theta_n'\|\sum_{t=1}^{T}L_{N,T}A_t, \tag{D.126}$$

we have

$$(D.111) \leq c\sqrt{r}L_{N,T}\sqrt{N}\left(\sum_{t=1}^{T}A_t\right)\|\theta_1 - \theta_1'\|_2 + crL_{N,T}N^{1/r}\left(\sum_{t=1}^{T}A_t\right)\|\theta_1 - \theta_1'\|_r \tag{D.127}$$

$$\leq c\sqrt{r}L_{N,T}\sqrt{N}\left(\sum_{t=1}^{T}A_t\right)\|\theta_1 - \mathbb{E}[\theta_1]\|_2 + crL_{N,T}N^{1/r}\left(\sum_{t=1}^{T}A_t\right)\|\theta_1 - \mathbb{E}[\theta_1]\|_q , \tag{D.128}$$

where we use Minkowski's inequality and the fact that $r \leq q$ to obtain the last inequality.

Combining (D.114), (D.125), and (D.128) and replacing $L_{N,T}$ by $L_T/N$ gives the result. ∎

**Proposition D.1** (Concentration bound for ICL). *Under Asms. 5–8, for any $\delta \in (0, e^{-2}]$, with probability at least $1 - \delta$,*

$$\left|\sup_{f\in\mathcal{F}}\left\{\mathbb{E}\left[\widehat{L}(f, (\theta_n, x_{1:T}^n)_{n\leq N})\right] - \widehat{L}(f, (\theta_n, x_{1:T}^n)_{n\leq N})\right\} - \mathbb{E}\left[\sup_{f\in\mathcal{F}}\left\{\mathbb{E}\left[\widehat{L}(f, (\theta_n, x_{1:T}^n)_{n\leq N})\right] - \widehat{L}(f, (\theta_n, x_{1:T}^n)_{n\leq N})\right\}\right]\right| \tag{D.129}$$

*is bounded by*

*(a) If $\delta \geq Ne^{-q}$,*

$$c\sigma\frac{L_T}{\sqrt{N}}\sqrt{T(\log(N/\delta) + 1)} \tag{D.130}$$

$$+ c\sqrt{(\log(N/\delta) + 1)}\frac{L_T}{\sqrt{N}}\sqrt{\sum_{t=1}^{T}\left(\sum_{s>t}B_{t,s}\right)^2}\|\theta_1\|_2 + c(\log(N/\delta) + 1)^{3/2}\frac{L_T}{N}\sqrt{\sum_{t=1}^{T}\left(\sum_{s>t}B_{t,s}\right)^2}\|\theta_1\|_q \tag{D.131}$$

$$+ c\sqrt{(\log(N/\delta) + 1)}\frac{L_T}{\sqrt{N}}\left(\sum_{t=1}^{T}A_t\right)\|\theta_1 - \mathbb{E}[\theta_1]\|_2 + c(\log(N/\delta) + 1)\frac{L_T}{N}\left(\sum_{t=1}^{T}A_t\right)\|\theta_1 - \mathbb{E}[\theta_1]\|_q \tag{D.132}$$

*(b) If $\delta < Ne^{-q}$,*

$$\frac{1}{\delta^{1/q}}\left(c\sigma L_{N,T}\sqrt{\frac{Tq}{N}}\right. \tag{D.133}$$

$$+ c\sqrt{q}\frac{L_T}{\sqrt{N}}\sqrt{\sum_{t=1}^{T}\left(\sum_{s>t}B_{t,s}\right)^2}\|\theta_1\|_2 + cq^{3/2}\frac{L_T}{N^{1-1/q}}\sqrt{\sum_{t=1}^{T}\left(\sum_{s>t}B_{t,s}\right)^2}\|\theta_1\|_q \tag{D.134}$$

$$\left.+ c\sqrt{q}\frac{L_T}{\sqrt{N}}\left(\sum_{t=1}^{T}A_t\right)\|\theta_1 - \mathbb{E}[\theta_1]\|_2 + cq\frac{L_T}{N^{1-1/q}}\left(\sum_{t=1}^{T}A_t\right)\|\theta_1 - \mathbb{E}[\theta_1]\|_q\right) \tag{D.135}$$

*Proof.* We apply Lem. D.5 to the moment bound from Lem. D.8.

For Lem. D.5, we use:

$$f(r) = c\sigma L_T\sqrt{\frac{Tr}{T}} + c\sqrt{r}\frac{L_T}{\sqrt{N}}\sqrt{\sum_{t=1}^{T}\left(\sum_{s>t}B_{t,s}\right)^2}\|\theta_1\|_2 + c\sqrt{r}\frac{L_T}{\sqrt{N}}\left(\sum_{t=1}^{T}A_t\right)\|\theta_1 - \mathbb{E}[\theta_1]\|_2 \tag{D.136}$$

$$g(r) = cr^{3/2}\frac{L_T}{N^{1-1/r}}\sqrt{\sum_{t=1}^{T}\left(\sum_{s>t}B_{t,s}\right)^2}\|\theta_1\|_q + cr\frac{L_T}{N^{1-1/r}}\left(\sum_{t=1}^{T}A_t\right)\|\theta_1 - \mathbb{E}[\theta_1]\|_q . \tag{D.137}$$

Applying Lem. D.5 then gives the desired concentration bound. ∎

## D.4. Complexity bounds for ICL

We now derive bounds for the analogue of the Rademacher complexity term in our setting. We will again rely on Thm. D.1.

**Lemma D.9.** *Under Asms. 5–9, we have*

$$\mathbb{E}\left[\sup_{f\in\mathcal{F}} \mathbb{E}\left[\widehat{L}(f,(\theta_n,x^n_{1:T})_{n\leq N})\right] - \widehat{L}(f,(\theta_n,x^n_{1:T})_{n\leq N})\right] \tag{D.138}$$

$$\leq c\mathcal{I}(\mathcal{F},\mathrm{dist},q)\left(\sigma M_T\sqrt{\frac{Tq}{N}}\right. \tag{D.139}$$

$$+ c\sqrt{q}\frac{M_T}{\sqrt{N}}\sqrt{\sum_{t=1}^{T}\left(\sum_{s>t}B_{t,s}\right)^2}\|\theta_1\|_2 + q^{3/2}\frac{M_T}{N^{1-1/q}}\sqrt{\sum_{t=1}^{T}\left(\sum_{s>t}B_{t,s}\right)^2}\|\theta_1\|_q \tag{D.140}$$

$$+ \sqrt{q}\frac{M_T}{\sqrt{N}}\left(\sum_{t=1}^{T}A_t\right)\|\theta_1 - \mathbb{E}[\theta_1]\|_2 + cq\frac{M_T}{N^{1-1/q}}\left(\sum_{t=1}^{T}A_t\right)\|\theta_1 - \mathbb{E}[\theta_1]\|_q\right), \tag{D.141}$$

*where $c > 0$ is a universal constant and where the Dudley-type integral $\mathcal{I}_{\mathrm{dist}}(\mathcal{F})$ is defined as*

$$\mathcal{I}(\mathcal{F},\mathrm{dist},q) = \int_0^{\Delta} (\mathcal{N}(\mathcal{F},\mathrm{dist},u))^{1/q}\,du\,, \quad \text{with } \Delta = \mathrm{diam}_{\mathrm{dist}}(\mathcal{F}) = \sup_{f,f'\in\mathcal{F}}\mathrm{dist}(f,f')\,. \tag{D.142}$$

*Proof.* The main idea of the proof is to use Lem. D.7 and to rely on Thm. D.1 to control the moments of the increments of the process $\sup_{f\in\mathcal{F}}\widehat{L}(f,(\theta_n,x^n_{1:T})_{n\leq N}) - \mathbb{E}\left[\widehat{L}(f,(\theta_n,x^n_{1:T})_{n\leq N})\right]$. Fix $f, f' \in \mathcal{F}$. We apply Thm. D.1 with

$$(Z_1,\ldots,Z_m) = (\theta_1,x^{(1)}_1,\ldots,x^{(1)}_T,\ldots,\theta_N,x^{(N)}_1,\ldots,x^{(N)}_T)\,, \tag{D.143}$$

and

$$g(\theta_1,x^{(1)}_{1:T},\ldots,\theta_N,x^{(N)}_{1:T}) \tag{D.144}$$

$$= \mathbb{E}\left[\widehat{L}(f,(\theta_n,x^n_{1:T})_{n\leq N})\right] - \widehat{L}(f,(\theta_n,x^n_{1:T})_{n\leq N}) \tag{D.145}$$

$$- \left(\mathbb{E}\left[\widehat{L}(f',(\theta_n,x^n_{1:T})_{n\leq N})\right] - \widehat{L}(f',(\theta_n,x^n_{1:T})_{n\leq N})\right) \tag{D.146}$$

and proceed as in the proof of Lem. D.8 except that $g$ is now $M_T\,\mathrm{dist}(f,f')$ coordinate-wise Lipschitz by Asm. 9 to obtain that:

$$\left\|\widehat{L}(f,(\theta_n,x^n_{1:T})_{n\leq N}) - \mathbb{E}\left[\widehat{L}(f,(\theta_n,x^n_{1:T})_{n\leq N})\right] - \left(\widehat{L}(f',(\theta_n,x^n_{1:T})_{n\leq N}) - \mathbb{E}\left[\widehat{L}(f',(\theta_n,x^n_{1:T})_{n\leq N})\right]\right)\right\|_q \tag{D.147}$$

$$\leq \mathrm{dist}(f,f')\left(c\sigma M_T\sqrt{\frac{Tq}{N}}\right. \tag{D.148}$$

$$+ c\sqrt{q}\frac{M_T}{\sqrt{N}}\sqrt{\sum_{t=1}^{T}\left(\sum_{s>t}B_{t,s}\right)^2}\|\theta_1\|_2 + cq^{3/2}\frac{M_T}{N^{1-1/q}}\sqrt{\sum_{t=1}^{T}\left(\sum_{s>t}B_{t,s}\right)^2}\|\theta_1\|_q \tag{D.149}$$

$$+ c\sqrt{q}\frac{M_T}{\sqrt{N}}\left(\sum_{t=1}^{T}A_t\right)\|\theta_1 - \mathbb{E}[\theta_1]\|_2 + cq\frac{M_T}{N^{1-1/q}}\left(\sum_{t=1}^{T}A_t\right)\|\theta_1 - \mathbb{E}[\theta_1]\|_q\right). \tag{D.150}$$

Applying Lem. D.7 then gives that

$$\mathbb{E}\left[\sup_{f,f'\in\mathcal{F}} \mathbb{E}\left[\widehat{L}(f,(\theta_n,x^n_{1:T})_{n\leq N})\right] - \widehat{L}(f,(\theta_n,x^n_{1:T})_{n\leq N}) - \left(\mathbb{E}\left[\widehat{L}(f',(\theta_n,x^n_{1:T})_{n\leq N})\right] - \widehat{L}(f',(\theta_n,x^n_{1:T})_{n\leq N})\right)\right] \tag{D.151}$$

is bounded by the RHS of the statement of the lemma. To conclude, it suffices to notice that, for any $f_0 \in \mathcal{F}$ fixed,

$$\mathbb{E}\left[\sup_{f\in\mathcal{F}} \mathbb{E}\left[\widehat{L}(f,(\theta_n,x^n_{1:T})_{n\leq N})\right] - \widehat{L}(f,(\theta_n,x^n_{1:T})_{n\leq N})\right] \tag{D.152}$$

$$= \mathbb{E}\left[ \sup_{f \in \mathcal{F}} \mathbb{E}\left[\widehat{L}(f, (\theta_n, x_{1:T}^n)_{n \leq N})\right] - \widehat{L}(f, (\theta_n, x_{1:T}^n)_{n \leq N}) - \left(\mathbb{E}\left[\widehat{L}(f_0, (\theta_n, x_{1:T}^n)_{n \leq N})\right] - \widehat{L}(f_0, (\theta_n, x_{1:T}^n)_{n \leq N})\right) \right] \tag{D.153}$$

$$\leq \mathbb{E}\left[ \sup_{f, f' \in \mathcal{F}} \mathbb{E}\left[\widehat{L}(f, (\theta_n, x_{1:T}^n)_{n \leq N})\right] - \widehat{L}(f, (\theta_n, x_{1:T}^n)_{n \leq N}) - \left(\mathbb{E}\left[\widehat{L}(f', (\theta_n, x_{1:T}^n)_{n \leq N})\right] - \widehat{L}(f', (\theta_n, x_{1:T}^n)_{n \leq N})\right) \right], \tag{D.154}$$

which concludes the proof. ∎

## D.5. Generalization bounds for ICL

Putting together the concentration bound from Proposition D.1 and the complexity bound from Lem. D.9, we obtain the following generalization bound for ICL:

**Theorem D.2** (Generalization bound for ICL). *Under Asms. 5–9, for any $\delta \in (0, e^{-2}]$, for any $\delta \in (0, Ne^{-q}]$, with probability at least $1 - \delta$, the generalization gap*

$$\sup_{f \in \mathcal{F}} \mathbb{E}\left[\widehat{L}(f, (\theta_n, x_{1:T}^n)_{n \leq N})\right] - \widehat{L}(f, (\theta_n, x_{1:T}^n)_{n \leq N}) \tag{D.155}$$

*is bounded by*

*(a) If $\delta \geq Ne^{-q}$,*

$$c\sigma \sqrt{\frac{T}{N}} \left( L_T \sqrt{(\log(N/\delta) + 1)} + M_T \mathcal{I}(\mathcal{F}, \text{dist}, q) \sqrt{q} \right) \tag{D.156}$$

$$+ c\left( L_T \sqrt{(\log(N/\delta) + 1)} + M_T \mathcal{I}(\mathcal{F}, \text{dist}, q) \sqrt{q} \right) \frac{1}{\sqrt{N}} \sqrt{\sum_{t=1}^{T} \left(\sum_{s>t} B_{t,s}\right)^2} \|\theta_1\|_2 \tag{D.157}$$

$$+ c\left( (\log(N/\delta) + 1)^{3/2} L_T + q^{3/2} N^{1/q} M_T \mathcal{I}(\mathcal{F}, \text{dist}, q) \right) \frac{1}{N} \sqrt{\sum_{t=1}^{T} \left(\sum_{s>t} B_{t,s}\right)^2} \|\theta_1\|_q \tag{D.158}$$

$$+ c\left( L_T \sqrt{(\log(N/\delta) + 1)} + M_T \mathcal{I}(\mathcal{F}, \text{dist}, q) \sqrt{q} \right) \frac{1}{\sqrt{N}} \left(\sum_{t=1}^{T} A_t\right) \|\theta_1 - \mathbb{E}[\theta_1]\|_2 \tag{D.159}$$

$$+ c\left( (\log(N/\delta) + 1) L_T + q N^{1/q} M_T \mathcal{I}(\mathcal{F}, \text{dist}, q) \right) \frac{1}{N} \left(\sum_{t=1}^{T} A_t\right) \|\theta_1 - \mathbb{E}[\theta_1]\|_q \tag{D.160}$$

*(b) If $\delta < Ne^{-q}$,*

$$\left( \frac{L_T}{\delta^{1/q}} + M_T \mathcal{I}(\mathcal{F}, \text{dist}, q) \right) \left( c\sigma \sqrt{\frac{Tq}{N}} \right. \tag{D.161}$$

$$+ c\sqrt{q} \frac{L_T}{\sqrt{N}} \sqrt{\sum_{t=1}^{T} \left(\sum_{s>t} B_{t,s}\right)^2} \|\theta_1\|_2 + cq^{3/2} \frac{L_T}{N^{1-1/q}} \sqrt{\sum_{t=1}^{T} \left(\sum_{s>t} B_{t,s}\right)^2} \|\theta_1\|_q \tag{D.162}$$

$$+ c\sqrt{q} \frac{L_T}{\sqrt{N}} \left(\sum_{t=1}^{T} A_t\right) \|\theta_1 - \mathbb{E}[\theta_1]\|_2 + cq \frac{L_T}{N^{1-1/q}} \left(\sum_{t=1}^{T} A_t\right) \|\theta_1 - \mathbb{E}[\theta_1]\|_q \right), \tag{D.163}$$

*where $c > 0$ is a universal constant and where the Dudley-type integral $\mathcal{I}_{\text{dist}}(\mathcal{F})$ is defined as*

$$\mathcal{I}(\mathcal{F}, \text{dist}, q) = \int_0^{\Delta} (\mathcal{N}(\mathcal{F}, \text{dist}, u))^{1/q} du, \quad \text{with } \Delta = \text{diam}_{\text{dist}}(\mathcal{F}) = \sup_{f, f' \in \mathcal{F}} \text{dist}(f, f'). \tag{D.164}$$

*Proof.* The result is obtained by combining Proposition D.1 and Lem. D.9: we write the decomposition

$$\sup_{f \in \mathcal{F}} \left\{ \mathbb{E}\left[ \widehat{L}(f, (\theta_n, x_{1:T}^n)_{n \leq N}) \right] - \widehat{L}(f, (\theta_n, x_{1:T}^n)_{n \leq N}) \right\} \tag{D.165}$$

$$= \mathbb{E}\left[ \sup_{f \in \mathcal{F}} \left\{ \mathbb{E}\left[ \widehat{L}(f, (\theta_n, x_{1:T}^n)_{n \leq N}) \right] - \widehat{L}(f, (\theta_n, x_{1:T}^n)_{n \leq N}) \right\} \right] \tag{D.166}$$

$$+ \sup_{f \in \mathcal{F}} \left\{ \mathbb{E}\left[ \widehat{L}(f, (\theta_n, x_{1:T}^n)_{n \leq N}) \right] - \widehat{L}(f, (\theta_n, x_{1:T}^n)_{n \leq N}) \right\} - \mathbb{E}\left[ \sup_{f \in \mathcal{F}} \left\{ \mathbb{E}\left[ \widehat{L}(f, (\theta_n, x_{1:T}^n)_{n \leq N}) \right] - \widehat{L}(f, (\theta_n, x_{1:T}^n)_{n \leq N}) \right\} \right],$$
$$\tag{D.167}$$

and we bound (D.166) using Lem. D.9 and (D.167) with high probability using Proposition D.1.

$\blacksquare$

## D.6. In-distribution vs. out-of-distribution generalization

Thms. 1 and D.2 focuses on in-distribution generalization, i.e., when the tasks at test time are sampled from the same prior $\pi$ as during training. However, the key challenge of ICL is often to generalize to out-of-distribution tasks, i.e., tasks that are sampled from a different task distribution $\rho$ at test time. In particular, in § 4, we will evaluate ICL on test tasks samples from distributions of the form $\rho = \mathcal{N}(\theta^*, \sigma^2 I_d)$ with $\theta^*$ increasingly far from the mode of the training prior $\pi$. This yields a principled way to evaluate the robustness of ICL to distribution shifts.

In that case, assuming $\pi$ has a density, the test error w.r.t. $\rho$ can be controlled using the test error w.r.t. $\pi$:

$$\mathbb{E}_{\theta \sim \rho}\left[ \mathbb{E}_{x_{1:T} \sim p_T(\cdot|\theta)}\left[ \frac{1}{T} \sum_{t=1}^T \ell_t(\hat{f}(x_{1:t-1}), x_t) \right] \right]$$

$$= \mathbb{E}_{\theta \sim \pi}\left[ \frac{d\rho(\theta)}{d\pi(\theta)} \mathbb{E}_{x_{1:T} \sim p_T(\cdot|\theta)}\left[ \frac{1}{T} \sum_{t=1}^T \ell_t(\hat{f}(x_{1:t-1}), x_t) \right] \right]$$

$$\leq \left\| \frac{d\rho}{d\pi} \right\|_\infty \mathbb{E}_{\theta \sim \pi}\left[ \mathbb{E}_{x_{1:T} \sim p_T(\cdot|\theta)}\left[ \frac{1}{T} \sum_{t=1}^T \ell_t(\hat{f}(x_{1:t-1}), x_t) \right] \right].$$

The right-most term is exactly the in-distribution test error w.r.t. $\pi$ controlled by Thm. 1, while the multiplicative factor $\|d\rho/d\pi\|_\infty$ quantifies the distribution shift between $\rho$ and $\pi$. For $\rho = \mathcal{N}(\theta^*, \sigma^2 I_d)$ with small $\sigma$, this factor $\|d\rho/d\pi\|_\infty$ is proportional to $1/\pi(\theta^*)$. As expected, the further the test task $\theta^*$ is from the mode of the training prior $\pi$, the worse the out-of-distribution generalization. Heavier-tailed priors $\pi$ mitigate this effect: since $\pi(\theta^*)$ decays more slowly as $\theta^*$ moves away from the mode for priors, the distribution shift factor $\|d\rho/d\pi\|_\infty$ grows more slowly, leading to better out-of-distribution generalization. Already, the trade-off in the choice of the prior $\pi$ starts to appear: heavier-tailed priors improve out-of-distribution generalization but harm in-distribution generalization.

## D.7. Extension: repeated tasks

In some ICL settings, tasks may be repeated multiple times in the training set. In this section, we extend our generalization bound Thm. D.2 to this setting.

We introduce $M > 0$, the number of times each task is repeated in the training set. The training data is now generated by first sampling a set of tasks $\theta_1, \ldots, \theta_N$ independently and identically according to the task distribution $\pi$, and then, for each task $\theta_n$, independently sampling $M$ sequences of data points $(x_t^{n,m})_{t \geq 1}$ for $m = 1, \ldots, M$ according to

$$x_{t+1}^{n,m} \sim p_{t+1}(\cdot \mid x_{1:t}^{n,m}, \theta_n), \tag{D.168}$$

where $x_{1:t}^{n,m} = (x_1^{n,m}, \ldots, x_t^{n,m})$.

Given such a dataset, a model $f$ is trained by minimizing the next-sample prediction loss

$$\widehat{L}(f, (\theta_n, (x_{1:T}^{n,m})_{m \leq M})_{n \leq N}) = \frac{1}{NTM} \sum_{n=1}^N \sum_{m=1}^M \sum_{t=1}^T \ell_t(f(x_{1:t-1}^{n,m}), x_t^n). \tag{D.169}$$

Applying the same proof as Lem. D.8, we obtain the following moment bound.

**Lemma D.10.** *For any $r \in [2, q]$ integer, under Asms. 5–8, we have*

$$\left\| \sup_{f \in \mathcal{F}} \left\{ \mathbb{E}\left[ \widehat{L}(f, (\theta_n, (x_{1:T}^{n,m})_{m \leq M})_{n \leq N}) \right] - \widehat{L}(f, (\theta_n, (x_{1:T}^{n,m})_{m \leq M})_{n \leq N}) \right\} \right. \tag{D.170}$$

$$\left. - \mathbb{E}\left[ \sup_{f \in \mathcal{F}} \left\{ \mathbb{E}\left[ \widehat{L}(f, (\theta_n, (x_{1:T}^{n,m})_{m \leq M})_{n \leq N}) \right] - \widehat{L}(f, (\theta_n, (x_{1:T}^{n,m})_{m \leq M})_{n \leq N}) \right\} \right] \right\|_r \tag{D.171}$$

$$\leq c \sigma L_T \sqrt{\frac{Tr}{NM}} \tag{D.172}$$

$$+ c \sqrt{r} \frac{L_T}{\sqrt{NM}} \sqrt{\sum_{t=1}^{T} \left( \sum_{s>t} B_{t,s} \right)^2} \|\theta_1\|_2 + c r^{3/2} \frac{L_T}{N^{1-1/r} \sqrt{M}} \sqrt{\sum_{t=1}^{T} \left( \sum_{s>t} B_{t,s} \right)^2} \|\theta_1\|_q \tag{D.173}$$

$$+ c \sqrt{r} \frac{L_T}{\sqrt{NM}} \left( \sum_{t=1}^{T} A_t \right) \|\theta_1 - \mathbb{E}[\theta_1]\|_2 + c r \frac{L_T}{N^{1-1/r} M} \left( \sum_{t=1}^{T} A_t \right) \|\theta_1 - \mathbb{E}[\theta_1]\|_q , \tag{D.174}$$

*where $c > 0$ is a universal constant.*

*Proof sketch.* The analogue of $g$ in the proof of Lem. D.8 is now coordinate-wise Lipschitz with respect to $x_t^{n,m}$ with constant $\frac{L_T}{NM}$. The proof proceeds as in Lem. D.8 with minor modifications to account for the $M$ independent repetitions. When going from (D.109) to (D.113), an additional factor $\sqrt{M}$ appears due to the sum of the independent repetitions. In the Hoeffding bound (D.122), a factor $\sqrt{M}$ also appears. Finally, when bounding (D.111), an additional $M$ factor also appears in (D.127). ∎

We now proceed with an analogue of Proposition D.1.

**Proposition D.2** (Concentration bound for ICL). *Under Asms. 5–8, for any $\delta \in (0, e^{-2}]$, with probability at least $1 - \delta$,*

$$\left| \sup_{f \in \mathcal{F}} \left\{ \mathbb{E}\left[ \widehat{L}(f, (\theta_n, (x_{1:T}^{n,m})_{m \leq M})_{n \leq N}) \right] - \widehat{L}(f, (\theta_n, (x_{1:T}^{n,m})_{m \leq M})_{n \leq N}) \right\} \right. \tag{D.175}$$

$$\left. - \mathbb{E}\left[ \sup_{f \in \mathcal{F}} \left\{ \mathbb{E}\left[ \widehat{L}(f, (\theta_n, (x_{1:T}^{n,m})_{m \leq M})_{n \leq N}) \right] - \widehat{L}(f, (\theta_n, (x_{1:T}^{n,m})_{m \leq M})_{n \leq N}) \right\} \right] \right| \tag{D.176}$$

*is bounded by*

(a) If $\delta \geq N e^{-q}$,

$$c \sigma \frac{L_T}{\sqrt{NM}} \sqrt{T(\log(N/\delta) + 1)} \tag{D.177}$$

$$+ c \sqrt{(\log(N/\delta) + 1)} \frac{L_T}{\sqrt{NM}} \sqrt{\sum_{t=1}^{T} \left( \sum_{s>t} B_{t,s} \right)^2} \|\theta_1\|_2 + c (\log(N/\delta) + 1)^{3/2} \frac{L_T}{N \sqrt{M}} \sqrt{\sum_{t=1}^{T} \left( \sum_{s>t} B_{t,s} \right)^2} \|\theta_1\|_q \tag{D.178}$$

$$+ c \sqrt{(\log(N/\delta) + 1)} \frac{L_T}{\sqrt{N}} \left( \sum_{t=1}^{T} A_t \right) \|\theta_1 - \mathbb{E}[\theta_1]\|_2 + c (\log(N/\delta) + 1) \frac{L_T}{N} \left( \sum_{t=1}^{T} A_t \right) \|\theta_1 - \mathbb{E}[\theta_1]\|_q \tag{D.179}$$

(b) If $\delta < N e^{-q}$,

$$\frac{1}{\delta^{1/q}} \left( c \sigma L_{N,T} \sqrt{\frac{Tq}{NM}} \right. \tag{D.180}$$

$$+ c \sqrt{q} \frac{L_T}{\sqrt{NM}} \sqrt{\sum_{t=1}^{T} \left( \sum_{s>t} B_{t,s} \right)^2} \|\theta_1\|_2 + c q^{3/2} \frac{L_T}{N^{1-1/q} \sqrt{M}} \sqrt{\sum_{t=1}^{T} \left( \sum_{s>t} B_{t,s} \right)^2} \|\theta_1\|_q \tag{D.181}$$

$$+ c\sqrt{q}\frac{L_T}{\sqrt{N}}\left(\sum_{t=1}^{T} A_t\right)\|\theta_1 - \mathbb{E}[\theta_1]\|_2 + cq\frac{L_T}{N^{1-1/q}}\left(\sum_{t=1}^{T} A_t\right)\|\theta_1 - \mathbb{E}[\theta_1]\|_q\right) \tag{D.182}$$

*Proof sketch.* As for [Proposition D.1](), we apply [Lem. D.5]() to the moment bound from [Lem. D.10](). ∎

We now proceed with the analogue of [Lem. D.9]() whose proof is similar.

**Lemma D.11.** *Under [Asms. 5–9](), we have*

$$\mathbb{E}\left[\sup_{f\in\mathcal{F}} \mathbb{E}\left[\widehat{L}(f, (\theta_n, (x_{1:T}^{n,m})_{m\le M})_{n\le N})\right] - \widehat{L}(f, (\theta_n, (x_{1:T}^{n,m})_{m\le M})_{n\le N})\right] \tag{D.183}$$

$$\le c\mathcal{I}(\mathcal{F}, \text{dist}, q)\left(\sigma M_T \sqrt{\frac{Tq}{NM}}\right. \tag{D.184}$$

$$+ c\sqrt{q}\frac{M_T}{\sqrt{NM}}\sqrt{\sum_{t=1}^{T}\left(\sum_{s>t} B_{t,s}\right)^2}\|\theta_1\|_2 + q^{3/2}\frac{M_T}{N^{1-1/q}\sqrt{M}}\sqrt{\sum_{t=1}^{T}\left(\sum_{s>t} B_{t,s}\right)^2}\|\theta_1\|_q \tag{D.185}$$

$$+ \sqrt{q}\frac{M_T}{\sqrt{N}}\left(\sum_{t=1}^{T} A_t\right)\|\theta_1 - \mathbb{E}[\theta_1]\|_2 + cq\frac{M_T}{N^{1-1/q}}\left(\sum_{t=1}^{T} A_t\right)\|\theta_1 - \mathbb{E}[\theta_1]\|_q\right), \tag{D.186}$$

*where $c > 0$ is a universal constant.*

Putting together [Proposition D.2]() and [Lem. D.11](), we obtain the following generalization bound for ICL with repeated tasks.

**Theorem D.3** (Generalization bound for ICL). *Under [Asms. 5–9](), for any $\delta \in (0, e^{-2}]$, for any $\delta \in (0, Ne^{-q}]$, with probability at least $1 - \delta$, the generalization gap*

$$\sup_{f\in\mathcal{F}} \mathbb{E}\left[\widehat{L}(f, (\theta_n, (x_{1:T}^{n,m})_{m\le M})_{n\le N})\right] - \widehat{L}(f, (\theta_n, (x_{1:T}^{n,m})_{m\le M})_{n\le N}) \tag{D.187}$$

*is bounded by*

(a) *If $\delta \ge Ne^{-q}$,*

$$c\sigma\sqrt{\frac{T}{NM}}\left(L_T\sqrt{(\log(N/\delta) + 1)} + M_T\mathcal{I}(\mathcal{F}, \text{dist}, q)\sqrt{q}\right) \tag{D.188}$$

$$+ c\left(L_T\sqrt{(\log(N/\delta) + 1)} + M_T\mathcal{I}(\mathcal{F}, \text{dist}, q)\sqrt{q}\right)\frac{1}{\sqrt{NM}}\sqrt{\sum_{t=1}^{T}\left(\sum_{s>t} B_{t,s}\right)^2}\|\theta_1\|_2 \tag{D.189}$$

$$+ c\left((\log(N/\delta) + 1)^{3/2}L_T + q^{3/2}N^{1/q}M_T\mathcal{I}(\mathcal{F}, \text{dist}, q)\right)\frac{1}{N\sqrt{M}}\sqrt{\sum_{t=1}^{T}\left(\sum_{s>t} B_{t,s}\right)^2}\|\theta_1\|_q \tag{D.190}$$

$$+ c\left(L_T\sqrt{(\log(N/\delta) + 1)} + M_T\mathcal{I}(\mathcal{F}, \text{dist}, q)\sqrt{q}\right)\frac{1}{\sqrt{N}}\left(\sum_{t=1}^{T} A_t\right)\|\theta_1 - \mathbb{E}[\theta_1]\|_2 \tag{D.191}$$

$$+ c\left((\log(N/\delta) + 1)L_T + qN^{1/q}M_T\mathcal{I}(\mathcal{F}, \text{dist}, q)\right)\frac{1}{N}\left(\sum_{t=1}^{T} A_t\right)\|\theta_1 - \mathbb{E}[\theta_1]\|_q \tag{D.192}$$

(b) *If $\delta < Ne^{-q}$,*

$$\left(\frac{L_T}{\delta^{1/q}} + M_T\mathcal{I}(\mathcal{F}, \text{dist}, q)\right)\left(c\sigma\sqrt{\frac{Tq}{NM}}\right. \tag{D.193}$$

$$+ c\sqrt{q}\frac{L_T}{\sqrt{NM}}\sqrt{\sum_{t=1}^{T}\left(\sum_{s>t} B_{t,s}\right)^2}\|\theta_1\|_2 + cq^{3/2}\frac{L_T}{N^{1-1/q}\sqrt{M}}\sqrt{\sum_{t=1}^{T}\left(\sum_{s>t} B_{t,s}\right)^2}\|\theta_1\|_q \tag{D.194}$$

$$+ c\sqrt{q}\frac{L_T}{\sqrt{N}}\left(\sum_{t=1}^{T} A_t\right)\|\theta_1 - \mathbb{E}[\theta_1]\|_2 + cq\frac{L_T}{N^{1-1/q}}\left(\sum_{t=1}^{T} A_t\right)\|\theta_1 - \mathbb{E}[\theta_1]\|_q\Bigg),, \tag{D.195}$$

where $c > 0$ is a universal constant and where the Dudley-type integral $\mathcal{I}_{\mathrm{dist}}(\mathcal{F})$ is defined as

$$\mathcal{I}(\mathcal{F}, \mathrm{dist}, q) = \int_0^{\Delta} (\mathcal{N}(\mathcal{F}, \mathrm{dist}, u))^{1/q} du, \quad with\ \Delta = \mathrm{diam}_{\mathrm{dist}}(\mathcal{F}) = \sup_{f,f'\in\mathcal{F}} \mathrm{dist}(f, f'). \tag{D.196}$$

The proof of Thm. D.3 is the same as that of Thm. D.2, using Proposition D.2 instead of Proposition D.1 and Lem. D.11 instead of Lem. D.9.

We also provide a simplified version of Thm. D.3 in the spirit of Thm. 1.

**Theorem D.4.** *Under Asms. 1–3, for any $\delta \in (0, e^{-2})$, with probability at least $1 - \delta$, it holds:*

*(a) If $\delta \geq Ne^{-q}$, then*

$$\widehat{\mathrm{gen}} \leq \mathcal{O}\left(\frac{(\log 1/\delta)^{3/2} L_T \sqrt{T}}{\sqrt{NM}}\left(1 + A_T\sqrt{TM} + B_T T\right)\right), \tag{D.197}$$

*(b) If $\delta < Ne^{-q}$, then*

$$\widehat{\mathrm{gen}} \leq \mathcal{O}\left(\frac{L_T \sqrt{T}}{\delta^{1/q}\sqrt{NM}}\left(1 + A_T\sqrt{TM} + B_T T\right)\right), \tag{D.198}$$

*where the terms in $\mathcal{O}(\cdot)$ depend polynomially on $q$, $\log N$, the scale of $\pi$ and the size of $\mathcal{F}$.*

# E. Task Selection

In this section, we study how tasks are selected at test time in ICL. This section is structured as follows. First we consider an abstract setting for Apps. E.1 and E.2 where in App. E.1 we state a few preliminary lemmas that will be useful in the analysis, and in App. E.2 we prove a template task selection bound under minimal assumptions. Then, in App. E.3, we reintroduce the ICL setting along with the detailed assumptions before proving the main task selection bound in App. E.4, which is where the main contribution of this section lies.

## E.1. Preliminary Lemmas

**Definition 1** (Kullback-Leibler divergence). For $\mathbb{P}$ and $\mathbb{Q}$ two probability measures on a measurable space $\mathcal{X}$, the *Kullback-Leibler (KL) divergence* from $\mathbb{P}$ to $\mathbb{Q}$ is defined as

$$\mathrm{KL}(\mathbb{P} \parallel \mathbb{Q}) = \begin{cases} \int_{\mathcal{X}} \log\left(\frac{d\mathbb{P}}{d\mathbb{Q}}(x)\right) d\mathbb{P}(x) & \text{if } \mathbb{P} \ll \mathbb{Q} \\ +\infty & \text{otherwise.} \end{cases} \tag{E.1}$$

We now state the Donsker-Varadhan lemma, also known as the Gibbs variational principle.

**Lemma E.1** (Donsker-Varadhan lemma, Gibbs variational principle). *Consider $\mathbb{P}$ probability measure on a measurable $\mathcal{X}$ and $g\colon \mathcal{X} \to \mathbb{R}$ a measurable function such that $\mathbb{E}_{\mathbb{P}}[\exp(g)] < \infty$. Then, we have*

$$\log \mathbb{E}_{\mathbb{P}}[e^{g(x)}] = \sup_{\mathbb{Q}}\left\{\mathbb{E}_{\mathbb{Q}}[g(x)] - \mathrm{KL}(\mathbb{Q} \parallel \mathbb{P})\right\}, \tag{E.2}$$

*with equality attained in particular for $\frac{d\mathbb{Q}}{d\mathbb{P}}(x) \propto e^{g(x)}$.*

See for instance Hellström et al. (2025); Rodríguez-Gálvez et al. (2024) for original references and proofs.

Let us state a technical consequence of this lemma that essentially corresponds to Zhang (2003, Lem. 3.1).

**Lemma E.2.** *Consider $X$ a random variable on $\mathcal{X}$ distributed according to $\mathbb{P}_X$ and $\theta$ a random variable on $\Theta$ with prior distribution $\pi(d\theta)$ and with posterior distribution such that, conditionally on $X$,*

$$\widehat{\mathbb{P}}(d\theta \mid X) = \frac{d\mathbb{P}(X \mid \theta)}{d\mathbb{P}(X)}\pi(d\theta)\,. \tag{E.3}$$

*Consider $L\colon \mathcal{X} \times \Theta \to \mathbb{R}$ a measurable function. Then,*

$$\mathbb{E}_{X,\theta\sim\widehat{\mathbb{P}}(\cdot|X)}[L(X,\theta) - \log \mathbb{E}_X[\exp(L(X,\theta))]] \leq \mathbb{E}_X[\mathrm{KL}(\mathbb{P}_\theta(\cdot \mid X) \parallel \pi)]\,. \tag{E.4}$$

*Proof.* We apply Lem. E.1 with $g(\theta) = L(X,\theta) - \log \mathbb{E}_X[\exp(L(X,\theta))]$ conditionally on $X$ to obtain

$$\mathbb{E}_{\theta\sim\widehat{\mathbb{P}}(\cdot|X)}[L(X,\theta) - \log \mathbb{E}_X[\exp(L(X,\theta))] - \mathrm{KL}(\mathbb{P}_\theta(\cdot \mid X) \parallel \pi)] \tag{E.5}$$

$$\leq \log \mathbb{E}_{\theta\sim\pi}[\exp(L(X,\theta) - \log \mathbb{E}_X[\exp(L(X,\theta))])]\,. \tag{E.6}$$

We then have

$$\mathbb{E}_X\left[\exp \mathbb{E}_{\theta\sim\widehat{\mathbb{P}}(\cdot|X)}[L(X,\theta) - \log \mathbb{E}_X[\exp(L(X,\theta))] - \mathrm{KL}(\mathbb{P}_\theta(\cdot \mid X) \parallel \pi)]\right] \tag{E.7}$$

$$\leq \mathbb{E}_{X,\theta\sim\pi}[\exp(L(X,\theta) - \log \mathbb{E}_X[\exp(L(X,\theta))])] = 1\,, \tag{E.8}$$

and the result follows by Jensen's inequality with the convex function exp. ∎

## E.2. Template Task Selection Bound

Let us start with a template task selection bound under minimal assumptions. This proof is adapted from Zhang (2003, Thm. 4.1) to the case of non-i.i.d. data and when the true task is not necessarily in the support of the prior.

**Proposition E.1** (Template task selection bound). *Consider $X$ a random variable on $\mathcal{X}$ distributed according to $\mathbb{P}_X$ and $\theta$ a random variable on $\Theta$ with prior distribution $\pi(d\theta)$ such that, conditionally on $X$, $\theta$ is distributed according to*

$$\widehat{\mathbb{P}}(d\theta \mid X) = \frac{d\,\mathbb{P}(X \mid \theta)}{d\,\mathbb{P}(X)}\pi(d\theta)\,. \tag{E.9}$$

*Then, we have, for any $\theta_0 \in \Theta$, for any $\rho \in (0,1)$, $\alpha > 1$,*

$$\mathbb{E}_{X,\theta \sim \widehat{\mathbb{P}}(\cdot \mid X)}\left[-\log \mathbb{E}_X\left[\left(\frac{d\,\mathbb{P}_X(\cdot \mid \theta)}{d\,\mathbb{P}_X(\cdot)}\right)^{\rho}\right]\right] \tag{E.10}$$

$$\leq -\alpha \log \mathbb{E}_{\theta \sim \pi}\left[\exp\left(-\mathbb{E}_X \log \frac{d\,\mathbb{P}_X(\cdot \mid \theta_0)}{d\,\mathbb{P}_X(\cdot \mid \theta)}\right)\right] + \alpha \mathrm{KL}(\mathbb{P}_X(\cdot) \parallel \mathbb{P}_X(\cdot \mid \theta_0)) \tag{E.11}$$

$$+ (\alpha - 1)\,\mathbb{E}_X\left[\log \mathbb{E}_{\theta \sim \pi}\left[\exp\left(-\frac{\alpha - \rho}{\alpha - 1}\log \frac{d\,\mathbb{P}_X(\cdot \mid \theta_0)}{d\,\mathbb{P}_X(\cdot \mid \theta)}\right)\right]\right] \tag{E.12}$$

*Proof.* To simplify notations in this proof, unless otherwise specified, $\theta$ indicates a random variable distributed according to $\widehat{\mathbb{P}}(\cdot \mid X)$. We start from Lem. E.2 with $L(X,\theta) = \rho \log \frac{d\,\mathbb{P}_X(\cdot \mid \theta)}{d\,\mathbb{P}_X(\cdot)}$ and rearrange to obtain:

$$\mathbb{E}_\theta\left[-\log \mathbb{E}_X\left[\left(\frac{d\,\mathbb{P}_X(\cdot \mid \theta)}{d\,\mathbb{P}_X(\cdot)}\right)^{\rho}\right]\right] \leq \mathbb{E}_{X,\theta}\left[\rho \log \frac{d\,\mathbb{P}_X(\cdot)}{d\,\mathbb{P}_X(\cdot \mid \theta)}\right] + \mathbb{E}_X[\mathrm{KL}(\mathbb{P}_\theta(\cdot \mid X) \parallel \pi)]\,. \tag{E.13}$$

The left-hand side (LHS) is the quantity we want to bound. We now only need to bound the RHS. Making $\theta_0 \in \Theta$ appear in the bound, we have

$$\mathbb{E}_{X,\theta}\left[\rho \log \frac{d\,\mathbb{P}_X(\cdot)}{d\,\mathbb{P}_X(\cdot \mid \theta)}\right] + \mathbb{E}_X[\mathrm{KL}(\mathbb{P}_\theta(\cdot \mid X) \parallel \pi)] \tag{E.14}$$

$$= \rho\,\mathbb{E}_X\left[\log \frac{d\,\mathbb{P}_X(\cdot)}{d\,\mathbb{P}_X(\cdot \mid \theta_0)}\right] + \mathbb{E}_{X,\theta}\left[\rho \log \frac{d\,\mathbb{P}_X(\cdot \mid \theta_0)}{d\,\mathbb{P}_X(\cdot \mid \theta)}\right] + \mathbb{E}_X[\mathrm{KL}(\mathbb{P}_\theta(\cdot \mid X) \parallel \pi)] \tag{E.15}$$

$$= \rho \mathrm{KL}(\mathbb{P}_X(\cdot) \parallel \mathbb{P}_X(\cdot \mid \theta_0)) \tag{E.16}$$

$$+ \mathbb{E}_{X,\theta}\left[\rho \log \frac{d\,\mathbb{P}_X(\cdot)}{d\,\mathbb{P}_X(\cdot \mid \theta)}\right] + \mathbb{E}_X[\mathrm{KL}(\mathbb{P}_\theta(\cdot \mid X) \parallel \pi)]\,. \tag{E.17}$$

Introducing $\alpha > 1$ and defining $\mu = \frac{\alpha - 1}{\alpha - \rho} < 1$, we now bound the last two terms in (E.17) as follows:

$$\mathbb{E}_{X,\theta}\left[\rho \log \frac{d\,\mathbb{P}_X(\cdot \mid \theta_0)}{d\,\mathbb{P}_X(\cdot \mid \theta)}\right] + \mathbb{E}_X[\mathrm{KL}(\mathbb{P}_\theta(\cdot \mid X) \parallel \pi)] \tag{E.18}$$

$$= \alpha\left(\mathbb{E}_{X,\theta}\left[\log \frac{d\,\mathbb{P}_X(\cdot \mid \theta_0)}{d\,\mathbb{P}_X(\cdot \mid \theta)}\right] + \mathbb{E}_X[\mathrm{KL}(\mathbb{P}_\theta(\cdot \mid X) \parallel \pi)]\right) \tag{E.19}$$

$$- (\alpha - \rho)\left(\mathbb{E}_{X,\theta}\left[\log \frac{d\,\mathbb{P}_X(\cdot \mid \theta_0)}{d\,\mathbb{P}_X(\cdot \mid \theta)}\right] + \mu\,\mathbb{E}_X[\mathrm{KL}(\mathbb{P}_\theta(\cdot \mid X) \parallel \pi)]\right)\,. \tag{E.20}$$

Let us first focus on the first term. By the equality case in Lem. E.1 and the definition of $\mathbb{P}(\theta \mid X)$, we have, almost surely,

$$\mathbb{E}_{\theta \sim \mathbb{P}(\cdot \mid X)}\left[\log \frac{d\,\mathbb{P}(X \mid \theta_0)}{d\,\mathbb{P}(X \mid \theta)}\right] + \mathrm{KL}(\mathbb{P}_\theta(\cdot \mid X) \parallel \pi) = \inf_{\mathbb{Q}}\left\{\mathbb{E}_{\theta \sim \mathbb{Q}}\left[\log \frac{d\,\mathbb{P}(X \mid \theta_0)}{d\,\mathbb{P}(X \mid \theta)}\right] + \mathrm{KL}(\mathbb{Q} \parallel \pi)\right\}\,. \tag{E.21}$$

Passing to the expectation over $X$ we obtain that,

$$\mathbb{E}\left[\log \frac{d\,\mathbb{P}(X)}{d\,\mathbb{P}(X \mid \theta)}\right] + \mathbb{E}_X[\mathrm{KL}(\mathbb{P}_\theta(\cdot \mid X) \parallel \pi)] \tag{E.22}$$

$$= \mathbb{E}_X\left[\inf_{\mathbb{Q}}\left\{\mathbb{E}_{\theta \sim \mathbb{Q}}\left[\log \frac{d\,\mathbb{P}(X \mid \theta_0)}{d\,\mathbb{P}(X \mid \theta)}\right] + \mathrm{KL}(\mathbb{Q} \parallel \pi)\right\}\right] \tag{E.23}$$

$$\leq \inf_{\mathbb{Q}}\left\{\mathbb{E}_{\theta \sim \mathbb{Q}}\left[\mathbb{E}_X\left[\log \frac{d\,\mathbb{P}(X \mid \theta_0)}{d\,\mathbb{P}(X \mid \theta)}\right]\right] + \mathrm{KL}(\mathbb{Q} \parallel \pi)\right\} \tag{E.24}$$

$$= -\log \mathbb{E}_{\theta \sim \pi} \left[ \exp\left( -\mathbb{E}_X \left[ \log \frac{d\,\mathbb{P}_X(\cdot \mid \theta_0)}{d\,\mathbb{P}_X(\cdot \mid \theta)} \right] \right) \right], \tag{E.25}$$

where the last line follows from Lem. E.1 again with $g(\theta) = -\mathbb{E}_X \left[ \log \frac{d\,\mathbb{P}_X(\cdot \mid \theta_0)}{d\,\mathbb{P}_X(\cdot \mid \theta)} \right]$. Let us now bound the second term in (E.20). We have, by Lem. E.1 again,

$$\mathbb{E}_{X,\theta} \left[ \log \frac{d\,\mathbb{P}_X(\cdot \mid \theta_0)}{d\,\mathbb{P}_X(\cdot \mid \theta)} \right] + \mu\,\mathbb{E}_X[\mathrm{KL}(\mathbb{P}_\theta(\cdot \mid X) \,\|\, \pi)] \tag{E.26}$$

$$\geq -\mu\,\mathbb{E}_X \left[ \log \mathbb{E}_{\theta \sim \pi} \left[ \exp\left( -\frac{1}{\mu} \log \frac{d\,\mathbb{P}_X(\cdot \mid \theta_0)}{d\,\mathbb{P}_X(\cdot \mid \theta)} \right) \right] \right]. \tag{E.27}$$

Putting together (E.20), (E.25), and (E.27) concludes the proof.

∎

### E.3. ICL setting

Let us now re-introduce the ICL setting from § 3.1 along with the detailed assumptions.

$\|\cdot\|$ denotes the Euclidean norm on $\mathbb{R}^d$ for any $d \in \mathbb{N}$. Assume that task vectors live in $\Theta \subset \mathbb{R}^d$ the space of tasks $\theta$ and by $\pi(\theta)$ the density of the pretraining task distribution. The context sequence is then generated by first sampling a task $\theta$ from the task distribution $\pi$, and then sampling data points $(x_t)_{t \geq 1}$ according to

$$x_{t+1} \sim \mathrm{p}_{t+1}(\cdot \mid x_{1:t}, \theta). \tag{E.28}$$

where $x_{1:t} = (x_1, \ldots, x_t)$.

We denote the posterior $\widehat{p}_t(\theta \mid x_{1:t-1})$ the posterior distribution over tasks given the input sequence $x_{1:t-1}$

Assumption 10 combined with Asm. 11 are the detailed version of Asm. 4 from § 3.1. Recall that we write $\mathrm{poly}(x)$ to denote a quantity that is polynomial in $x$ with coefficients independent of the prior $\pi$ and the number of samples $T$. We also denote by $\overline{\mathbb{B}}(0, R)$ the closed ball of radius $R$ centered at 0 in $\mathbb{R}^d$ for the Euclidean norm $\|\cdot\|$.

**Assumption 10** (Data generation). Fix $\theta^* \in \Theta$ the true task and $\theta_0 \in \Theta$ a reference task such that $\pi(\theta_0) > 0$.

- Tail behaviour of $(x_t)_{t \geq 1}$: there is $k \geq 1$ such that for any $T \geq 1$, $R \geq T$,

$$\mathbb{P}_{X \sim \mathrm{p}_T(\cdot \mid \theta^*)} \left( \sup_{\theta : \|\theta\| \geq R} \mathrm{p}_T(X \mid \theta) \geq \mathrm{p}_T(X \mid \theta_0) \right) \leq \frac{\mathrm{poly}(T)}{1 + R^{1/k}} \tag{E.29}$$

$$\mathbb{P}_{X \sim \mathrm{p}_T(\cdot \mid \theta^*)} \left( \exists t \leq T, \|x_t\| \geq R \right) \leq \frac{\mathrm{poly}(T)}{1 + R^{1/k}} + \tag{E.30}$$

- Moment bound on $(x_t)_{t \geq 1}$: for any $T \geq 1$

$$\mathbb{E}_{X \sim \mathrm{p}_T(\cdot \mid \theta^*)} \left[ \log^2 \left( \sup_{\theta \in \Theta} \frac{\mathrm{p}_T(X \mid \theta)}{\mathrm{p}_T(X \mid \theta_0)} \right) \right] \leq \mathrm{poly}(T). \tag{E.31}$$

- Regularity of the likelihood: for any $t \geq 1$, $\theta, \theta' \in \Theta \cap \overline{\mathbb{B}}(0, R)$,

$$\sup_{x_{1:t} \in \overline{\mathbb{B}}(0,R)^t} \log \frac{\mathrm{p}_t(x_t \mid x_{1:t-1}, \theta)}{\mathrm{p}_t(x_t \mid x_{1:t-1}, \theta')} \leq \mathrm{poly}(R)\|\theta - \theta'\|. \tag{E.32}$$

For a sequence $(x_t)_{t \geq 1}$, we denote by $x_{a:b}$ the subsequence $(x_a, x_{a+1}, \ldots, x_b)$ for $1 \leq a \leq b$ with the convention that $x_{a:b} = x_{1:t}$ if $a < 1$.

### E.4. Task Selection Bound for ICL

We begin with a discretization argument and first we generalize the bracketing numbers to the non-i.i.d. case. This definition generalizes the bracketing numbers used in Barron et al. (1999); Zhang (2003; 2006) to the non-i.i.d case and the following result generalises the results of Zhang (2006) to the non-i.i.d. case.

**Definition 2.** Given a sequence of random variables $(x_t)_{t \leq T}$ on a measurable space $\mathcal{X}$, with parametric densities $p_t(\cdot|\theta)$ parameterized by $\theta \in \Theta$, compact sets $\Theta' \subset \Theta$ and $\mathcal{X}' \subset \mathcal{X}$, the $\varepsilon$-upper bracketing number of $\Theta'$, denoted by $\mathcal{B}(\Theta', \varepsilon, \mathcal{X}', T)$ is the minimum number of sets $U_j$ that cover $\Theta'$ such that, for any $t \leq T - 1$, any $x_{1:t+1} \in \mathcal{X}'^{t+1}$, any $j$,

$$\int_{\mathcal{X}'} \sup_{\theta \in U_j} p_{t+1}(x_{t+1} \mid x_{1:t}, \theta) dx_{t+1} \leq 1 + \varepsilon. \tag{E.33}$$

**Lemma E.3.** *For $\mu \in (0, 1)$, for any $\varepsilon > 0$ and any compact set $\Theta' \subset \Theta$, any set $\mathcal{X}' \subset \mathcal{X}$, it holds*

$$\mu \, \mathbb{E}_{x_{1:T}} \left[ \log \mathbb{E}_{\theta \sim \pi} \left[ \exp\left( -\frac{1}{\mu} \log \frac{p_T(x_{1:T} \mid \theta_0)}{p_T(x_{1:T} \mid \theta)} \right) \right] \right] \tag{E.34}$$

$$\leq 2 \log(\mathcal{B}(\Theta', \varepsilon, \mathcal{X}', T)) + 6T\varepsilon + \pi(\theta \notin \Theta')^\mu \tag{E.35}$$

$$+ \mathbb{E}_{x_{1:T}} \left[ \mathbb{1} \left\{ \sup_{\theta \notin \Theta'} \frac{p_T(x_{1:T} \mid \theta)}{p_T(x_{1:T} \mid \theta_0)} \geq 1 \right\} \cdot \log\left( 1 + \sup_{\theta \notin \Theta'} \frac{p_T(x_{1:T} \mid \theta)}{p_T(x_{1:T} \mid \theta_0)} \right) \right] \tag{E.36}$$

$$+ \mathbb{E}_{x_{1:T}} \left[ \mathbb{1} \{x_{1:T} \notin \mathcal{X}'^T\} \cdot \log\left( \sup_{\theta \in \Theta} \frac{p_T(x_{1:T} \mid \theta)}{p_T(x_{1:T} \mid \theta_0)} \right) \right]. \tag{E.37}$$

*Proof.* First, let us consider $\theta \in \Theta'$ and $X = x_{1:T} \in \mathcal{X}'^T$. We have

$$\exp\left( -\frac{1}{\mu} \log \frac{p_T(X \mid \theta_0)}{p_T(X \mid \theta)} \right) = \exp\left( \frac{1}{\mu} \sum_{t=0}^{T-1} \log \frac{p_{t+1}(x_{t+1} \mid x_{1:t}, \theta)}{p_{t+1}(x_{t+1} \mid x_{1:t}, \theta_0)} \right) \tag{E.38}$$

Invoking the bracketing definition (Definition 2), we obtain sets $U_j$, for $j = 1, \ldots, \mathcal{B}(\Theta', \varepsilon, \mathcal{X}', T)$ such that, for any $t \leq T - 1$, any $x_{1:t+1} \in \mathcal{X}'^{t+1}$, any $j$, with $g_j(\cdot \mid \cdot) := \sup_{\theta \in U_j} p_{t+1}(\cdot \mid \cdot, \theta)$,

$$\int_{\mathcal{X}'} g_j(x_{t+1} \mid x_{1:t}) dx_{t+1} \leq 1 + \varepsilon. \tag{E.39}$$

Therefore, for any $\theta \in \Theta'$, any $t \geq 1$, any $x_{1:t+1} \in \mathcal{X}'^{t+1}$, there exists $i \in \{1, \ldots, \mathcal{B}(\Theta', \varepsilon, \mathcal{X}', T)\}$ such that

$$p_{t+1}(x_{t+1} \mid x_{1:t}, \theta) \leq g_i(x_{t+1} \mid x_{1:t}). \tag{E.40}$$

Hence, we can bound

$$\exp\left( -\frac{1}{\mu} \log \frac{p_T(X \mid \theta_0)}{p_T(X \mid \theta)} \right) \leq \exp\left( \frac{1}{\mu} \sum_{t=0}^{T-1} \log \frac{g_i(x_{t+1} \mid x_{1:t})}{p_{t+1}(x_{t+1} \mid x_{1:t}, \theta_0)} + \frac{T}{\mu} \log \frac{1+\varepsilon}{1-\varepsilon} \right). \tag{E.41}$$

We now control the contribution from $\theta \notin \Theta'$ by simply taking the supremum over this set. We have

$$\mathbb{E}_{\theta \sim \pi} \left[ \mathbb{1}\{\theta \notin \Theta'\} \cdot \exp\left( -\frac{1}{\mu} \log \frac{p_T(X \mid \theta_0)}{p_T(X \mid \theta)} \right) \right] \tag{E.42}$$

$$= \pi(\theta \notin \Theta') \sup_{\theta \notin \Theta'} \left( \frac{p_T(X \mid \theta)}{p_T(X \mid \theta_0)} \right)^{1/\mu}. \tag{E.43}$$

Combining (E.41) and (E.43), we bound the LHS of the statement as

$$\mu \, \mathbb{E}_X \left[ \mathbb{1}\{X \in \mathcal{X}'^T\} \log \mathbb{E}_{\theta \sim \pi} \left[ \exp\left( -\frac{1}{\mu} \log \frac{p_T(X \mid \theta_0)}{p_T(X \mid \theta)} \right) \right] \right] \tag{E.44}$$

$$= \mu \, \mathbb{E}_X \left[ \mathbb{1}\{X \in \mathcal{X}'^T\} \log \mathbb{E}_{\theta \sim \pi} \left[ \mathbb{1}\{\theta \in \Theta'\} \exp\left(-\frac{1}{\mu} \log \frac{p_T(X \mid \theta_0)}{p_T(X \mid \theta)}\right) \right] + \mathbb{1}\{\theta \notin \Theta'\} \exp\left(-\frac{1}{\mu} \log \frac{p_T(X \mid \theta_0)}{p_T(X \mid \theta)}\right) \right] \quad \text{(E.45)}$$

$$\leq \mu \, \mathbb{E}_X \left[ \mathbb{1}\{X \in \mathcal{X}'^T\} \log \left( \sum_{i=1}^{\mathcal{B}(\Theta', \varepsilon, \mathcal{X}', T)} \exp\left(\frac{1}{\mu} \sum_{t=0}^{T-1} \log \frac{g_i(x_{t+1} \mid x_{1:t})}{p_{t+1}(x_{t+1} \mid x_{1:t}, \theta_0)} + \frac{T}{\mu} \log \frac{1+\varepsilon}{1-\varepsilon}\right) \right. \right. \quad \text{(E.46)}$$

$$\left. \left. + \pi(\theta \notin \Theta') \cdot \sup_{\theta \notin \Theta'} \left(\frac{p_T(X \mid \theta)}{p_T(X \mid \theta_0)}\right)^{1/\mu} \right) \right]. \quad \text{(E.47)}$$

Since $\mu \in (0,1)$, for any non-negative numbers $a_1, \ldots, a_K$ we have $\left(\sum_{k=1}^{K} a_k\right)^\mu \leq \sum_{k=1}^{K} a_k^\mu$. Using this inequality and that $\log(a+b) \leq \log(1+a) + \log(1+b)$ for $a, b \geq 0$, we obtain

$$\mu \, \mathbb{E}_X \left[ \mathbb{1}\{X \in \mathcal{X}'^T\} \log \mathbb{E}_{\theta \sim \pi} \left[ \exp\left(-\frac{1}{\mu} \log \frac{p_T(X \mid \theta_0)}{p_T(X \mid \theta)}\right) \right] \right] \quad \text{(E.48)}$$

$$\leq \mathbb{E}_X \left[ \mathbb{1}\{X \in \mathcal{X}'^T\} \log \left( \sum_{i=1}^{\mathcal{B}(\Theta', \varepsilon, \mathcal{X}', T)} \exp\left(\sum_{t=0}^{T-1} \log \frac{g_i(x_{t+1} \mid x_{1:t})}{p_{t+1}(x_{t+1} \mid x_{1:t}, \theta_0)} + T \log \frac{1+\varepsilon}{1-\varepsilon}\right) \right. \right. \quad \text{(E.49)}$$

$$\left. \left. + \pi(\theta \notin \Theta')^\mu \cdot \sup_{\theta \notin \Theta'} \left(\frac{p_T(X \mid \theta)}{p_T(X \mid \theta_0)}\right) \right) \right] \quad \text{(E.50)}$$

$$\leq \mathbb{E}_X \left[ \mathbb{1}\{X \in \mathcal{X}'^T\} \log \left(1 + \sum_{i=1}^{\mathcal{B}(\Theta', \varepsilon, \mathcal{X}', T)} \exp\left(\sum_{t=0}^{T-1} \log \frac{g_i(x_{t+1} \mid x_{1:t})}{p_{t+1}(x_{t+1} \mid x_{1:t}, \theta_0)} + T \log \frac{1+\varepsilon}{1-\varepsilon}\right)\right) \right. \quad \text{(E.51)}$$

$$\left. + \log\left(1 + \pi(\theta \notin \Theta')^\mu \cdot \sup_{\theta \notin \Theta'} \left(\frac{p_T(X \mid \theta)}{p_T(X \mid \theta_0)}\right)\right) \right]. \quad \text{(E.52)}$$

Using Jensen's inequality on the first term, we have

$$\mu \, \mathbb{E}_X \left[ \mathbb{1}\{X \in \mathcal{X}'^T\} \log \mathbb{E}_{\theta \sim \pi} \left[ \exp\left(-\frac{1}{\mu} \log \frac{p_T(X \mid \theta_0)}{p_T(X \mid \theta)}\right) \right] \right] \quad \text{(E.53)}$$

$$\leq \log\left(1 + \mathbb{E}_X \left[ \sum_{i=1}^{\mathcal{B}(\Theta', \varepsilon, \mathcal{X}', T)} \exp\left(\sum_{t=0}^{T-1} \log \frac{g_i(x_{t+1} \mid x_{1:t})}{p_{t+1}(x_{t+1} \mid x_{1:t}, \theta_0)} + T \log \frac{1+\varepsilon}{1-\varepsilon}\right) \right]\right) \quad \text{(E.54)}$$

$$+ \mathbb{E}_X \left[ \log\left(1 + \pi(\theta \notin \Theta')^\mu \cdot \sup_{\theta \notin \Theta'} \left(\frac{p_T(X \mid \theta)}{p_T(X \mid \theta_0)}\right)\right) \right] \quad \text{(E.55)}$$

$$\leq \log\left(1 + \mathcal{B}(\Theta', \varepsilon, \mathcal{X}', T)(1+\varepsilon)^T \left(\frac{1+\varepsilon}{1-\varepsilon}\right)^T\right) + \mathbb{E}_X \left[ \log\left(1 + \pi(\theta \notin \Theta')^\mu \cdot \mathbb{E}_X \left[ \sup_{\theta \notin \Theta'} \left(\frac{p_T(X \mid \theta)}{p_T(X \mid \theta_0)}\right) \right]\right) \right], \quad \text{(E.56)}$$

where we used the definition of the bracketing number [Definition 2](#) in the last line. To obtain the final result, we perform additional manipulations on each term. For the first term, we use that $\frac{1}{1-x} \leq 1 + 2x$ for $x \in (0, 1/2)$ so that

$$\log\left((1+\varepsilon)^T \left(\frac{1+\varepsilon}{1-\varepsilon}\right)^T\right) \leq \log\left((1+2\varepsilon)^{3T}\right) \leq 6T\varepsilon, \quad \text{(E.57)}$$

so that

$$\log\left(1 + \mathcal{B}(\Theta', \varepsilon, \mathcal{X}', T)(1+\varepsilon)^T \left(\frac{1+\varepsilon}{1-\varepsilon}\right)^T\right) \leq \log(1 + \mathcal{B}(\Theta', \varepsilon, \mathcal{X}', T)) + 6T\varepsilon \quad \text{(E.58)}$$

$$\leq 2\log(\mathcal{B}(\Theta', \varepsilon, \mathcal{X}', T)) + 6T\varepsilon. \quad \text{(E.59)}$$

For the second term, we use that $\log(1+x) \leq x$ and distinguish two cases to obtain

$$\mathbb{E}_X \left[ \log\left(1 + \pi(\theta \notin \Theta')^\mu \cdot \mathbb{E}_X \left[ \sup_{\theta \notin \Theta'} \left(\frac{p_T(X \mid \theta)}{p_T(X \mid \theta_0)}\right) \right]\right) \right] \quad \text{(E.60)}$$

$$\leq \pi(\theta \notin \Theta')^\mu + \mathbb{E}_X \left[ \mathbb{1}\left\{\sup_{\theta \notin \Theta'} \frac{p_T(X \mid \theta)}{p_T(X \mid \theta_0)} \geq 1\right\} \cdot \log\left(1 + \sup_{\theta \notin \Theta'} \frac{p_T(X \mid \theta)}{p_T(X \mid \theta_0)}\right) \right]. \quad \text{(E.61)}$$

All that is left to do is to deal with the case $X \notin \mathcal{X}'^T$. We have, as above,

$$\mu \mathbb{E}_X\left[\mathbb{1}\{X \notin \mathcal{X}'^T\} \log \mathbb{E}_{\theta \sim \pi}\left[\exp\left(-\frac{1}{\mu} \log \frac{\mathrm{p}_T(X \mid \theta_0)}{\mathrm{p}_T(X \mid \theta)}\right)\right]\right] \le \mathbb{E}_X\left[\mathbb{1}\{X \notin \mathcal{X}'^T\} \log\left(\sup_{\theta \in \Theta} \frac{\mathrm{p}_T(X \mid \theta)}{\mathrm{p}_T(X \mid \theta_0)}\right)\right]. \tag{E.62}$$

$\blacksquare$

We now leverage Asm. 10 to control the different terms of Lem. E.3.

**Lemma E.4.** *For $\mu \in (0, 1)$, under Asm. 10, for any $T \ge 1$, it holds that*

$$\mu \mathbb{E}_{x_{1:T}}\left[\log \mathbb{E}_{\theta \sim \pi}\left[\exp\left(-\frac{1}{\mu} \log \frac{\mathrm{p}_T(x_{1:T} \mid \theta_0)}{\mathrm{p}_T(x_{1:T} \mid \theta)}\right)\right]\right] \le \pi(\theta \notin \Theta')^\mu + \mathcal{O}(\log(T)), \tag{E.63}$$

*where the $\mathcal{O}(\cdot)$ hides constants that do not depend on $\pi$ or $T$.*

*Proof.* Fix $R > 0$ that will be chosen later and take $\mathcal{X}' = \overline{\mathbb{B}}(0, R)$ and $\Theta' = \overline{\mathbb{B}}(0, R)$. Let us consider a $\delta$-cover of $\Theta'$ with $\delta > 0$ that will be chosen later: there are $K$ sets $U_j$, $j = 1, \ldots, K$ that cover $\Theta'$ such that for any $\theta, \theta' \in U_j$, we have $\|\theta - \theta'\| \le \delta$. By e.g., Wainwright (2019, Ex. 5.2), we can take $K$ such that $\log K \le d \log(1 + 2R/\delta)$.

Assumption 10 ensures that the sets $U_j$ satisfy the bracketing condition of Definition 2 with $\varepsilon = \exp(\mathrm{poly}(R)\delta) - 1$. Therefore, we have, with this choice of $\varepsilon$,

$$\log \mathcal{B}(\Theta', \varepsilon, \mathcal{X}', T) \le d \log(1 + 2R/\delta). \tag{E.64}$$

Using Cauchy-Schwarz inequality and Asm. 10, we have that, both

$$\mathbb{E}_{x_{1:T}}\left[\mathbb{1}\left\{\sup_{\theta \notin \Theta'} \frac{\mathrm{p}_T(x_{1:T} \mid \theta)}{\mathrm{p}_T(x_{1:T} \mid \theta_0)} \ge 1\right\} \cdot \log\left(1 + \sup_{\theta \notin \Theta'} \frac{\mathrm{p}_T(x_{1:T} \mid \theta)}{\mathrm{p}_T(x_{1:T} \mid \theta_0)}\right)\right] \le \frac{\mathrm{poly}(T)}{1 + R^{1/k}} \tag{E.65}$$

$$\mathbb{E}_{x_{1:T}}\left[\mathbb{1}\left\{x_{1:T} \notin \mathcal{X}'^T\right\} \cdot \log\left(\sup_{\theta \in \Theta} \frac{\mathrm{p}_T(x_{1:T} \mid \theta)}{\mathrm{p}_T(x_{1:T} \mid \theta_0)}\right)\right] \le \frac{\mathrm{poly}(T)}{1 + R^{1/k}}. \tag{E.66}$$

Choose $R = \mathrm{poly}(T)$ so that both (E.65) and (E.66) are $\mathcal{O}(1)$. Finally, we choose $\delta = (\mathrm{poly}(T))^{-1}$ so that $\varepsilon = \exp(\mathrm{poly}(R)\delta) - 1 = \mathcal{O}(1/T)$. Combining this (E.64)–(E.66) with Lem. E.3 concludes the proof. $\blacksquare$

We can now state our main result for ICL. As a metric to asses the quality of a given retrieved task $\theta$ w.r.t. the true task $\theta^*$, we consider the Rényi divergence (Rényi, 1961) of order $\rho \in (0, 1)$ between the distributions $\mathrm{p}_T(\cdot \mid \theta)$ and $\mathrm{p}_T(\cdot \mid \theta^*)$:

$$\mathrm{D}_\rho(\theta \| \theta^*) = -\frac{1}{T(1 - \rho)} \log \mathbb{E}_{X \sim \mathrm{p}_T(\cdot \mid \theta^*)}\left[\prod_{t=1}^T \left(\frac{\mathrm{p}_t(x_t \mid x_{1:t-1}, \theta)}{\mathrm{p}_t(x_t \mid x_{1:t-1}, \theta^*)}\right)^\rho\right]. \tag{E.67}$$

**Theorem E.1.** *Under Asm. 10, for any $\rho \in (0, 1)$, $T \ge 1$, it holds that, for $x_{1:T} \sim \mathrm{p}_T(\cdot \mid \theta^*)$,*

$$\mathbb{E}_{x_{1:T}}\left[\mathbb{E}_{\theta \sim \widehat{p}_T(\cdot \mid x_{1:T})}\left[\mathrm{D}_\rho(\theta \| \theta^*)\right]\right] \tag{E.68}$$

$$\le -\frac{1 + \rho}{(1 - \rho)T} \log\left(\mathbb{E}_{\theta \sim \pi}\left[\exp\left(-\mathbb{E}_{x_{1:T}}\left[\log \frac{\mathrm{p}_T(x_{1:T} \mid \theta_0)}{\mathrm{p}_T(x_{1:T} \mid \theta)}\right]\right)\right]\right) \tag{E.69}$$

$$+ \frac{1 + \rho}{1 - \rho} \frac{\mathrm{KL}(\mathrm{p}_T(\cdot \mid \theta^*) \| \mathrm{p}_T(\cdot \mid \theta_0))}{T} \tag{E.70}$$

$$+ \mathcal{O}\left(\frac{\log(T)}{T}\right), \tag{E.71}$$

*where the $\mathcal{O}(\cdot)$ hides constants that do not depend on $\pi$ or $T$.*

*Proof.* This is a direct consequence of Proposition E.1 combined with Lem. E.4 with $\alpha = 1 + \rho$ and bounding $\pi(\theta \notin \Theta')^\mu \le 1$. $\blacksquare$

A few comments are in order. The first term of (E.69) captures how much the prior $\pi$ covers the reference task $\theta_0$. When $\theta_0 = \theta^*$, this term thus quantifies how well the prior covers the true task $\theta^*$. When $\theta_0$ is inside the support of $\pi$, this term is vanishing as $T$ grows large, see the next results below.

The second term of (E.70) captures how well the reference task $\theta_0$ approximates the true task $\theta^*$. When $\theta_0 = \theta^*$, the term of (E.70) is 0. Otherwise, consider the case the KL will typically be of order $T$ so that this term is $\mathcal{O}(1)$: it represents the best ICL error one can hope for when the true task $\theta^*$ is not in the support of the prior $\pi$.

## E.5. Laplace Approximation

We will make use of the following version of the Laplace approximation, see Wong (2001, Chap. 9, Thm. 3) for a proof.

**Lemma E.5** (Laplace approximation). *Let $\mu$ be a probability measure on $\mathbb{R}^d$ with density $g : \mathbb{R}^d \to [0, \infty)$. Fix $x^* \in \mathbb{R}^d$ such that $g$ is continuous at $x^*$ and $g(x^*) > 0$. Then, as $\varepsilon \to 0$,*

$$\int_{\mathbb{R}^d} \exp\left(-\tfrac{1}{2\varepsilon} \|x - x^*\|\right) g(x) \, dx, \; = \; g(x^*) \, C \, \varepsilon^d \; + \; o(\varepsilon^d).$$

*where $C := \int_{\mathbb{R}^d} \exp\left(-\tfrac{1}{2} \|y\|\right) dy \in (0, \infty)$.*

**Assumption 11.** Consider the following additional assumptions to Asm. 10:

- Tail behaviour: for any $T \geq 1$, $R > 0$,

$$\mathbb{P}_{X \sim p_T(\cdot \mid \theta^*)}\left(\sup_{\theta : \|\theta\| \geq R} p_T(X \mid \theta) \geq p_T(X \mid \theta_0)\right) \leq \mathrm{poly}(T) e^{-R} \tag{E.72}$$

$$\mathbb{P}_{X \sim p_T(\cdot \mid \theta^*)}\left(\exists t \leq T, \|x_t\| \geq R\right) \leq \mathrm{poly}(T) e^{-R}. \tag{E.73}$$

- Regularity of $\pi$: $\pi$ is continuous and positive at $\theta_0$.

- Second moment of $\pi$:

$$\mathbb{E}_{\theta \sim \pi}\left[\|\theta\|^2\right] < \infty. \tag{E.74}$$

**Proposition E.2.** *Under Asms. 10 and 11, then, for $T$ large enough,*

$$-\log\left(\mathbb{E}_{\theta \sim \pi}\left[\exp\left(-\mathbb{E}_{x_{1:T}}\left[\log \frac{p_T(x_{1:T} \mid \theta_0)}{p_T(x_{1:T} \mid \theta)}\right]\right)\right]\right) \leq \log 1/\pi(\theta_0) + \mathcal{O}(\mathrm{poly}(\log T)). \tag{E.75}$$

*Proof.* For some $R_T \geq r_T > 0$, we split the term as

$$-\log\left(\mathbb{E}_{\theta \sim \pi}\left[\exp\left(-\mathbb{E}_{x_{1:T}}\left[\log \frac{p_T(x_{1:T} \mid \theta_0)}{p_T(x_{1:T} \mid \theta)}\right]\right)\right]\right) \tag{E.76}$$

$$= -\log\left(\mathbb{E}_{\theta \sim \pi}\left[\mathbb{1}\{\|\theta\| \leq R_T\} \exp\left(-\mathbb{E}_{x_{1:T}}\left[\log \frac{p_T(x_{1:T} \mid \theta_0)}{p_T(x_{1:T} \mid \theta)}\right]\right) + \mathbb{1}\{\|\theta\| > R_T\} \exp\left(-\mathbb{E}_{x_{1:T}}\left[\log \frac{p_T(x_{1:T} \mid \theta_0)}{p_T(x_{1:T} \mid \theta)}\right]\right)\right]\right) \tag{E.77}$$

$$\leq -\log\left(\mathbb{E}_{\theta \sim \pi}\left[\mathbb{1}\{\|\theta\| \leq r_T\} \exp\left(-\mathbb{E}_{x_{1:T}}\left[\log \frac{p_T(x_{1:T} \mid \theta_0)}{p_T(x_{1:T} \mid \theta)}\right]\right) + \mathbb{1}\{\|\theta\| > R_T\} \exp\left(-\mathbb{E}_{x_{1:T}}\left[\log \frac{p_T(x_{1:T} \mid \theta_0)}{p_T(x_{1:T} \mid \theta)}\right]\right)\right]\right) \tag{E.78}$$

Using Cauchy-Schwarz inequality and Asm. 10 and its refinement in the statement, we bound the second term as, for $\theta$ such that $\|\theta\| > R_T$, so that

$$\left|\mathbb{E}_{x_{1:T}}\left[\log \frac{p_T(x_{1:T} \mid \theta_0)}{p_T(x_{1:T} \mid \theta)}\right]\right| \leq e^{-R_T/2} \mathrm{poly}(T). \tag{E.79}$$

so that

$$\mathbb{E}_{\theta \sim \pi}\left[\mathbb{1}\{\|\theta\| > R_T\} \exp\left(-\mathbb{E}_{x_{1:T}}\left[\log \frac{p_T(x_{1:T} \mid \theta_0)}{p_T(x_{1:T} \mid \theta)}\right]\right)\right] \tag{E.80}$$

$$\leq \exp\left(e^{-R_T/2} \operatorname{poly}(T)\right) \pi(\|\theta\| > R_T) \tag{E.81}$$

$$\leq \exp\left(e^{-R_T/2} \operatorname{poly}(T)\right) \frac{\mathbb{E}_{\theta \sim \pi}\left[\|\theta\|^2\right]}{R_T^2}, \tag{E.82}$$

where we used Markov's inequality in the last line. Take $R_T = T^{(d+1)}/2$ so that (E.82) is $\mathcal{O}(1/T^{d+1})$.

We now focus on the first term of (E.78) and bound it as:

$$\mathbb{E}_{x_{1:T}}\left[\log \frac{p_T(x_{1:T} \mid \theta_0)}{p_T(x_{1:T} \mid \theta)}\right] = \mathbb{E}_{x_{1:T}}\left[\mathbb{1}\left\{\max_t \|x_t\| \leq r_T\right\} \log \frac{p_T(x_{1:T} \mid \theta_0)}{p_T(x_{1:T} \mid \theta)}\right] + \mathbb{E}_{x_{1:T}}\left[\mathbb{1}\left\{\max_t \|x_t\| > r_T\right\} \log \frac{p_T(x_{1:T} \mid \theta_0)}{p_T(x_{1:T} \mid \theta)}\right] \tag{E.83}$$

$$\leq \operatorname{poly}(r_T)T\|\theta - \theta_0\| + \operatorname{poly}(T)e^{-r_T/2} \tag{E.84}$$

where we used the regularity assumption of Asm. 10 for the first term and Cauchy-Schwarz inequality combined with Asm. 11 for the second term.

Take $r_T = \operatorname{poly}(\log T)$ so that $\operatorname{poly}(T)e^{-r_T/2} = \mathcal{O}(1)$ and assume that $T$ is large enough so that $r_T \geq \|\theta_0\| + 1$.

Putting everything together, we have

$$-\log\left(\mathbb{E}_{\theta \sim \pi}\left[\exp\left(-\mathbb{E}_{x_{1:T}}\left[\log \frac{p_T(x_{1:T} \mid \theta_0)}{p_T(x_{1:T} \mid \theta)}\right]\right)\right]\right) \tag{E.85}$$

$$\leq -\log\left(\mathbb{E}_{\theta \sim \pi}\left[\mathbb{1}\{\|\theta\| \leq r_T\} \exp\left(-\operatorname{poly}(r_T)T\|\theta - \theta_0\| + \mathcal{O}(1)\right) + \mathcal{O}\left(\frac{1}{T^{d+1}}\right)\right]\right) \tag{E.86}$$

$$\leq -\log\left(\mathbb{E}_{\theta \sim \pi}\left[\mathbb{1}\{\|\theta\| \leq \|\theta_0\| + 1\} \exp\left(-\operatorname{poly}(\log T)T\|\theta - \theta_0\| + \mathcal{O}(1)\right) + \mathcal{O}\left(\frac{1}{T^{d+1}}\right)\right]\right), \tag{E.87}$$

where we used that we assumed that $r_T = \operatorname{poly}(\log T) \geq \|\theta_0\| + 1$.

Applying Lem. E.5 with $\varepsilon = 1/(\operatorname{poly}(\log T)T)$ yields:

$$\mathbb{E}_{\theta \sim \pi}[\mathbb{1}\{\|\theta\| \leq \|\theta_0\| + 1\} \exp(-\operatorname{poly}(\log T)T\|\theta - \theta_0\|)] = \operatorname{poly}(\log T)T^{-d}(\pi(\theta_0)C + o(1)), \tag{E.88}$$

where $C$ is the constant of Lem. E.5 and this concludes the proof.

∎

We can now combine Thm. E.1 and Proposition E.2 to obtain the final result in the main text.

**Theorem E.2.** *Under Asms. 10 and 11, for any $\rho \in (0,1)$, $T \geq 1$, it holds that, for $x_{1:T} \sim p_T(\cdot \mid \theta^*)$,*

$$\mathbb{E}_{x_{1:T}}\left[\mathbb{E}_{\theta \sim \widehat{p}_T(\cdot \mid x_{1:T})}\left[D_\rho(\theta \| \theta^*)\right]\right] \tag{E.89}$$

$$\leq \frac{1+\rho}{(1-\rho)T} \log 1/\pi(\theta_0) \tag{E.90}$$

$$+ \frac{1+\rho}{1-\rho} \frac{\operatorname{KL}(p_T(\cdot \mid \theta^*) \| p_T(\cdot \mid \theta_0))}{T} \tag{E.91}$$

$$+ \mathcal{O}\left(\frac{\log(T)}{T}\right), \tag{E.92}$$

*where the $\mathcal{O}(\cdot)$ hides constants that do not depend on $\pi$ or $T$.*

*Proof.* This is a direct consequence of Thm. E.1 and Proposition E.2. ∎

# F. Additional details on examples

### F.1. Example: Volterra equation model

We discuss the Volterra equation model to explicit the dependence of the generalization bounds on the memory decay parameter $\alpha > 0$.

**Setup.** Let $(W_t)_{t \geq 1}$ be noise sequence taking values in $\mathbb{R}^d$. Given a Lipschitz drift $b : \mathbb{R}^d \to \mathbb{R}^d$ with Lipschitz constant $L \geq 0$, we consider the discretized Volterra equation: for $t \geq 0$,

$$X_{t+1} = \sum_{u=1}^{t} K(t, u) \left( b(X_u) + W_u \right), \qquad K(t, u) = \frac{1}{(t - u + 1)^\alpha}, \quad \alpha > 0. \tag{F.1}$$

When applying the generalization framework, we would consider the augmented sequence $(X_1, W_1, X_2, W_2, \ldots)$. To satisfy the weak dependence assumption Asm. 5, we need to bound the effect of perturbations in either the state or the noise or the drift. We begin with perturbations in the state or noise, and we discuss drift perturbations at the end of this section. For perturbations in the state or noise, we will obtain bounds on the Wasserstein distance between the conditional laws of $X_t$ and $X'_t$ given the past, where $X_t$ and $X'_t$ are two versions of the process (F.1) that differ by a perturbation at some time $s < t$.

The coefficient $\alpha$ will play a key role in the dependence structure through the sums:

$$H_\alpha(n) = \sum_{r=1}^{n} \frac{1}{r^\alpha}. \tag{F.2}$$

We also use $\zeta(\alpha) = \sum_{r=1}^{\infty} r^{-\alpha}$ for $\alpha > 1$ and we have the following bounds on $H_\alpha(n)$

$$H_\alpha(n) \leq \begin{cases} 1 + \log n, & \alpha = 1, \\ \zeta(\alpha), & \alpha > 1. \end{cases} \tag{F.3}$$

We will make use of the following technical lemma.

**Lemma F.1.** *Let $(a_n)_{n \geq 0}$ be nonnegative numbers and suppose that for $n \geq 1$,*

$$a_n \leq L \sum_{r=1}^{n} r^{-\alpha} a_{n-r} + g_n, \tag{F.4}$$

*with non-decreasing $(g_n)_{n \geq 1}$ and given $a_0 \geq 0$. Define, for $N \geq 1$,*

$$\lambda_N := \begin{cases} L(1 + \log N) & \text{if } \alpha = 1, \\ L\zeta(\alpha) & \text{if } \alpha > 1. \end{cases} \tag{F.5}$$

*Then, for all $1 \leq n \leq N$,*

$$a_n \leq \lambda_N^n a_0 + \sum_{j=1}^{n} g_j \lambda_N^{n-j}. \tag{F.6}$$

*Proof.* Let $A_n := \max_{0 \leq m \leq n} a_m$. From (F.4), $a_n \leq L \sum_{r=1}^{n} r^{-\alpha} A_{n-r} + g_n \leq L H_\alpha(n) A_{n-1} + g_n$, so $A_n \leq L H_\alpha(n) A_{n-1} + g_n$ since $(g_n)_n$ is non-decreasing. Bounding $H_\alpha(n)$ using (F.3) gives $A_n \leq \lambda_N A_{n-1} + g_n$ for all $1 \leq n \leq N$. Iterating this inequality yields the result. ∎

**State perturbation.** Fix $s \geq 1$ and let $\mathcal{F}_s := \sigma(X_1, \ldots, X_s, W_1, \ldots, W_s)$ on which we condition. Assume the two systems agree up to $s - 1$, and at time $s$ we have

$$X'_s = X_s - h$$

with $h \neq 0$. For $t \geq s$, define $\Delta_t := X_t - X'_t$. Subtracting (F.1) for the two evolutions (they share $(W_u)$) gives for $t \geq s$:

$$\Delta_{t+1} = \sum_{u=s}^{t} \frac{b(X_u) - b(X'_u)}{(t - u + 1)^\alpha}, \qquad \|\Delta_{t+1}\| \leq L \sum_{u=s}^{t} \frac{\|\Delta_u\|}{(t - u + 1)^\alpha}. \tag{F.7}$$

Set $n := t - s + 1$, $a_n := \mathbb{E}\big(\|\Delta_{s+n}\| \,\big|\, \mathcal{F}_s\big)$ and $a_0 = \|\Delta_s\| = \|h\|$. Applying Lemma F.1 with $g_n = 0$ yields, for $n \leq N$,

$$a_n \leq \lambda_N^n \, \|h\|, \tag{F.8}$$

We now bound the Wasserstein distance between the conditional laws of $X_{s+n}$ and $X'_{s+n}$ given $\mathcal{F}_s$ by using the synchronous coupling between $X_{s+n}$ and $X'_{s+n}$ (which share the same noise sequence $(W_u)_{u>s}$):

$$W_1\big(\mathcal{L}(X_{s+n} \mid \mathcal{F}_s), \; \mathcal{L}(X'_{s+n} \mid \mathcal{F}_s)\big) \leq \mathbb{E}\big(\|X_{s+n} - X'_{s+n}\| \mid \mathcal{F}_s\big) \leq \lambda_N^n \, \|h\|.$$

Therefore, for any horizon $T \geq s + 1$,

$$\sup_{s+1 \leq t \leq T} W_1\big(\mathcal{L}(X_t \mid \mathcal{F}_s), \; \mathcal{L}(X'_t \mid \mathcal{F}_s)\big) \;\leq\; \|h\| \, \lambda_{T-s}^{T-s} = \begin{cases} \|h\| \, (L(1 + \log(T - s))^{T-s} & \text{if } \alpha = 1, \\ \|h\| \, (L\zeta(\alpha))^{T-s} & \text{if } \alpha > 1. \end{cases} \tag{F.9}$$

The behaviour of the bound crucially depends on $\alpha$ and $L$: if $\alpha > 1$ and $L\zeta(\alpha) < 1$, the effect of the perturbation decays exponentially fast with $T - s$; if $\alpha > 1$ and $L\zeta(\alpha) > 1$, the effect of the perturbation grows exponentially fast with $T - s$. In both case, higher values of $\alpha$ (faster memory decay) lead to better dependence properties.

**Noise perturbation.** Fix $s \geq 1$ and let $\mathcal{F}_{s-1} := \sigma\big(X_1, \ldots, X_{s-1}, W_1, \ldots, W_{s-1}\big)$. Assume the two systems agree up to time $s$ except that at time $s$ we have

$$W'_s \;=\; W_s + \eta$$

with $\eta \neq 0$, and $W'_u = W_u$ for $u \neq s$. Again define $\Delta_t := X_t - X'_t$ for $t \geq s$. Subtracting the two recursions gives for $t \geq s$:

$$\Delta_{t+1} = \sum_{u=s}^{t} \frac{b(X_u) - b(X'_u)}{(t - u + 1)^\alpha} \;+\; \frac{W_s - W'_s}{(t - s + 1)^\alpha}. \tag{F.10}$$

Taking norms and using Lipschitzness,

$$\|\Delta_{t+1}\| \;\leq\; L \sum_{u=s}^{t} \frac{\|\Delta_u\|}{(t - u + 1)^\alpha} \;+\; \frac{\|\eta\|}{(t - s + 1)^\alpha}.$$

Set $n := t - s + 1$ and $a_n := \mathbb{E}\big(\|\Delta_{s+n}\| \,\big|\, \mathcal{F}_{s-1}\big)$. Note $a_0 = 0$ (since $X_s = X'_s$). Apply Lemma F.1 with $g_n := \|\eta\| \, n^{-\alpha}$ to obtain, for $n \leq N$,

$$a_n \;\leq\; \sum_{j=1}^{n} \|\eta\| j^{-\alpha} \lambda_N^{n-j} \leq \|\eta\| \times \frac{\lambda_N^n - 1}{\lambda_N - 1}, \tag{F.11}$$

where we consider $\lambda_N \neq 1$ for simplicity.

Bounding the Wasserstein distance as before yields, for any horizon $T \geq s + 1$,

$$\sup_{s+1 \leq t \leq T} W_1\big(\mathcal{L}(X_t \mid \mathcal{F}_{s-1}), \; \mathcal{L}(X'_t \mid \mathcal{F}_{s-1})\big) \;\leq\; \begin{cases} \|\eta\| \, \frac{(L(1+\log(T-s)))^{T-s} - 1}{L(1+\log(T-s)) - 1}, & \text{if } \alpha = 1, \\ \|\eta\| \, \frac{(L\zeta(\alpha))^{T-s} - 1}{L\zeta(\alpha) - 1}, & \text{if } \alpha > 1. \end{cases} \tag{F.12}$$

**Drift perturbation.** To consider drift perturbations, we write the drift as $b_\theta$ where $\theta$ is a parameter. In addition to assuming that $b_\theta$ is uniformly $L$-Lipschitz for all $\theta$, we also assume that it is $M$-Lipschitz in $\theta$ uniformly in $x$, that is, for all $x, x' \in \mathbb{R}^d$ and $\theta, \theta'$,

$$\|b_\theta(x) - b_{\theta'}(x')\| \;\leq\; L \, \|x - x'\| + M \, \|\theta - \theta'\|. \tag{F.13}$$

Consider $\theta, \theta'$ and the two systems with drifts $b_\theta$ and $b_{\theta'}$ respectively:

$$X_{t+1} \;=\; \sum_{u=1}^{t} K(t, u) \, \big(b_\theta(X_u) + W_u\big), \tag{F.14}$$

$$X'_{t+1} \;=\; \sum_{u=1}^{t} K(t, u) \, \big(b_{\theta'}(X'_u) + W_u\big). \tag{F.15}$$

As before, we will bound the Wasserstein distance between $X_t$ and $X_t'$ by using the synchronous coupling. Assuming that the two sequences share the same noise sequence $(W_u)$, we define $\Delta_t = X_t - X_t'$ and obtain, using (F.13), for $t \le T$

$$\|\Delta_{t+1}\| \le L \sum_{u=1}^{t} \frac{\|\Delta_u\|}{(t - u + 1)^\alpha} + M\|\theta - \theta'\| H_\alpha(T). \tag{F.16}$$

Setting $a_n = \|\Delta_n\|$ and $g_n = M\|\theta - \theta'\| H_\alpha(T)$ with $a_0 = 0$, we can apply Lemma F.1 as before to obtain, for $t \le T$,

$$W_1\big(\mathcal{L}(X_t),\ \mathcal{L}(X_t')\big) \ \le \ M\|\theta - \theta'\| \begin{cases} (1 + \log T)\, \frac{(L(1+\log T))^t - 1}{L(1+\log T) - 1}, & \text{if } \alpha = 1, \\ \zeta(\alpha)\, \frac{(L\zeta(\alpha))^t - 1}{L\zeta(\alpha) - 1}, & \text{if } \alpha > 1 \end{cases} \tag{F.17}$$

where we used (F.3) to bound $H_\alpha(T)$.

### F.2. Examples for task selection assumptions

In this section, we check that the examples of § 3.1 in the main text satisfy Asms. 10 and 11. These are lengthy but mostly straightforward calculations, which we sketch to illustrate how to verify the assumptions in practice. We also explicit the link between the Renyi divergence that appears in Thm. 2 and the usual loss functions in these examples.

**Example F.1** (Linear regression). We consider the linear regression example of § 3.1 in the main text and check that it satisfies Asms. 10 and 11. Fix a true task $\theta^* \in \mathbb{R}^d$. For $t = 1, \ldots, T$, consider $q_t \sim \mathcal{N}(0, \sigma_q{}^2 I_d)$ and noise $\epsilon_t \sim \mathcal{N}(0, \sigma_\epsilon^2)$ i.i.d., and $y_t = q_t^\top \theta^* + \epsilon_t$, $z_t = (q_t, y_t)$, $X = \{z_t\}_{t=1}^T$. Define $Q \in \mathbb{R}^{T \times d}$ has rows $q_t^\top$ and $Y = (y_t)_{t=1}^T$, and, for any parameter $\theta \in \mathbb{R}^d$,

$$\ell_T(\theta) := \log \mathrm{p}_T(X \mid \theta) = -\frac{1}{2\sigma_\epsilon^2}\|Y - Q\theta\|_2^2 + \text{const},$$

where the constant term depends on $Q$ but not on $\theta$

Let us begin with the tail behavior. Both $q_t$ and $y_t = q_t^T \theta^* + \epsilon_t$ are sub-Gaussian; hence for some $c > 0$ and all $R \ge 1$,

$$\mathbb{P}(\exists t \le n\,, \|z_t\| \ge R) \ \le \ \mathrm{poly}(n)\, e^{-cR^2} \ \le \ \mathrm{poly}(n)\, e^{-R}.$$

For the tail condition on the likelihood, let $\Delta = \theta - \theta_0$ and $r_0 := Y - Q\theta_0$. Then

$$\ell_T(\theta) - \ell_T(\theta_0) = -\frac{1}{2\sigma_\epsilon^2}\big(\|Q\Delta\|_2^2 - 2\Delta^\top Q^\top r_0\big)$$

Now, by e.g., Wainwright (2019, Thm. 6.1), for $T$ large enough, there is $c > 0$ constant such that, with probability at least $1 - e^{-cT}$, $\|Q\Delta\| \ge c\sqrt{T}\,\|\Delta\|$ and $\|Q^\top r_0\| \le c^{-1}\sqrt{T}\,\|r_0\|$. Hence, uniformly over $\|\theta\| \ge R$ (so $\|\Delta\| \ge R - \|\theta_0\|$),

$$\ell_T(\theta) - \ell_T(\theta_0) \ \le \ -\frac{c^2 T}{2\sigma_\epsilon^2}\|\Delta\|^2 + \frac{c^{-1}\sqrt{T}}{\sigma_\epsilon^2}\|\Delta\|\,\|r_0\|.$$

For all $R$ larger than a constant multiple of $\|r_0\|/\sqrt{T} + \|\theta_0\|$, the right-hand side is negative; thus $\sup_{\|\theta\| \ge R} \mathrm{p}_T(X \mid \theta) < \mathrm{p}_T(X \mid \theta_0)$. Since $\|r_0\|$ is sub-Gaussian and the norm bounds above hold with probability at least $1 - e^{-cn} \ge 1 - e^{-cR}$ for $R \ge T$, we obtain, for all $R \ge T$,

$$\mathbb{P}\left(\sup_{\|\theta\| \ge R} \mathrm{p}_T(X \mid \theta) \ \ge \ \mathrm{p}_T(X \mid \theta_0)\right) \ \le \ \mathrm{poly}(T)\, e^{-R}.$$

We now consider the moment condition. Then, for any reference $\theta_0$,

$$\sup_\theta \frac{\mathrm{p}_T(X \mid \theta)}{\mathrm{p}_T(X \mid \theta_0)} = \exp\Big(\sup_\theta\{\ell_T(\theta) - \ell_T(\theta_0)\}\Big) \le \exp\Big(\frac{1}{2\sigma_\epsilon^2}\|Y - Q\theta_0\|_2^2\Big),$$

Therefore, we have

$$\log^2 \sup_\theta \frac{\mathrm{p}_T(X \mid \theta)}{\mathrm{p}_T(X \mid \theta_0)} \ \le \ C\left(\|Q(\theta^* - \theta_0)\|_2^2 + \|\epsilon\|_2^2\right)^2,$$

and using Gaussian moment bounds

$$\mathbb{E}\left[\log^2 \sup_\theta \frac{p_T(X \mid \theta)}{p_T(X \mid \theta_0)}\right] \leq \text{poly}(n)\left(1 + \|\theta^* - \theta_0\|_2^4\right) = \text{poly}(n).$$

We finally check the local regularity condition. For any $t$ and $\theta, \theta'$,

$$\log \frac{p_t(y_t \mid q_{1:t}, y_{1:t-1}, \theta)}{p_t(y_t \mid q_{1:t}, y_{1:t-1}, \theta')} = -\frac{1}{2\sigma_\epsilon^2}\left[(y_t - \theta^\top q_t)^2 - (y_t - \theta'^\top q_t)^2\right].$$

Assuming that $\|q_{1:t}\|_\infty, |y_{1:t}| \leq R$ and $\|\theta\|, \|\theta'\| \leq R$ (with $R \geq 1$) and using that $(a-b)^2 - (a-c)^2 = (c-b)(2a-b-c)$, we have

$$\left|\log \frac{p_t(y_t \mid q_{1:t}, y_{1:t-1}, \theta)}{p_t(y_t \mid q_{1:t}, y_{1:t-1}, \theta')}\right| = \frac{1}{2\sigma_\epsilon^2}\left|(\theta - \theta')^\top q_t\right|\left|2y_t - (\theta + \theta')^\top q_t\right| \leq \frac{1}{\sigma_\epsilon^2}R^3\|\theta - \theta'\|,$$

so the condition holds.

Let us now explicit the Renyi divergence in this case. Since $q_t$ do not depend on $\theta$ and $(q_t, y_t)_t$ are i.i.d., we have

$$D_\rho(\theta \parallel \theta^*) = -\frac{\lfloor T/2 \rfloor}{T(1-\rho)}\log \mathbb{E}_{q,y}\left[\left(\frac{p(y \mid q, \theta)}{p(y \mid q, \theta^*)}\right)^\rho\right]. \tag{F.18}$$

We now focus on the expectation and write, using standard Gaussian integrals,

$$\mathbb{E}_{q,y}\left[\left(\frac{p(y \mid q, \theta)}{p(y \mid q, \theta^*)}\right)^\rho\right] = \mathbb{E}_q \mathbb{E}_{y|q}\left[\exp\left(\frac{\rho}{2\sigma_\epsilon^2}\left((y - q^\top\theta^*)^2 - (y - q^\top\theta)^2\right)\right)\right] \tag{F.19}$$

$$= \mathbb{E}_q \mathbb{E}_{y|q}\left[\exp\left(\frac{\rho}{2\sigma_\epsilon^2}\left(2\epsilon q^\top(\theta^* - \theta) - (q^\top(\theta^* - \theta))^2\right)\right)\right] \tag{F.20}$$

$$= \mathbb{E}_q\left[\exp\left(-\frac{\rho(1-\rho)}{2\sigma_\epsilon^2}(q^\top(\theta^* - \theta))^2\right)\right] \tag{F.21}$$

$$= \frac{1}{\sqrt{1 + \frac{\rho^2(1-\rho)^2\sigma_q^2}{\sigma_\epsilon^4}\|\theta - \theta^*\|^2}}. \tag{F.22}$$

The Renyi divergence is therefore

$$D_\rho(\theta \parallel \theta^*) = \frac{\lfloor T/2 \rfloor}{2T(1-\rho)}\log\left(1 + \frac{\rho^2(1-\rho)^2\sigma_q^2}{\sigma_\epsilon^4}\|\theta - \theta^*\|^2\right). \tag{F.23}$$

Moreover, for $\rho$ either close to 0 or 1, we have the approximation

$$D_\rho(\theta \parallel \theta^*) = \frac{\rho\lfloor T/2 \rfloor \sigma_q^2 \rho^2(1-\rho)}{2T\sigma_\epsilon^4}\|\theta - \theta^*\|^2 + \mathcal{O}\left(\rho^4(1-\rho)^3\right). \tag{F.24}$$

Hence, the quantity bounded in Thm. 2 can be related to the squared loss as follows:

$$\mathbb{E}_{\theta \sim \widehat{p}_T(\cdot|x_{1:T})}\left[D_\rho(\theta \parallel \theta^*)\right] \tag{F.25}$$

$$= \frac{\rho\lfloor T/2 \rfloor \sigma_q^2 \rho^2(1-\rho)}{2T\sigma_\epsilon^4}\mathbb{E}_{\theta \sim \widehat{p}_T(\cdot|x_{1:T})}\left[\|\theta - \theta^*\|^2\right] + \mathcal{O}\left(\rho^4(1-\rho)^3\right) \tag{F.26}$$

$$\geq \frac{\rho\lfloor T/2 \rfloor \sigma_q^2 \rho^2(1-\rho)}{2T\sigma_\epsilon^4}\|\mathbb{E}_{\theta \sim \widehat{p}_T(\cdot|x_{1:T})}[\theta] - \theta^*\|^2 + \mathcal{O}\left(\rho^4(1-\rho)^3\right) \tag{F.27}$$

$$= \frac{\rho\lfloor T/2 \rfloor \rho^2(1-\rho)}{2T\sigma_\epsilon^4}\mathbb{E}_q\left\|\mathbb{E}_{\theta \sim \widehat{p}_T(\cdot|x_{1:T})}[\mathbb{E}[y \mid q, \theta]] - \mathbb{E}[y \mid q, \theta^*]\right\|^2 \tag{F.28}$$

$$+ \mathcal{O}\left(\rho^4(1-\rho)^3\right), \tag{F.29}$$

where we used Jensen's inequality in the second line. Note that $\mathbb{E}_{\theta \sim \widehat{p}_T(\cdot | x_{1:T})}[\mathbb{E}[y \mid q, \theta]]$ is the optimal Bayesian predictor under the squared loss given the posterior distribution over $\theta$, see (3). As a conclusion, the Renyi divergence term in Thm. 2 controls the squared prediction error of the Bayesian predictor, which models the in-context learning performance.

**Example F.2** (Ornstein–Uhlenbeck process). We consider the Ornstein–Uhlenbeck (OU) process example of § 3.1 in the main text and check that it satisfies Asms. 10 and 11. For simplicity, we consider the one-dimensional case $d = 1$; the extension to $d > 1$ with diagonal diffusion is straightforward. We consider tasks $\theta = (\mu, \tau)$ where $\mu \in \mathbb{R}$ and $\tau \in [\overline{\tau}, \underline{\tau}]$ with $0 < \overline{\tau} \leq \underline{\tau} < \infty$. Given $\theta$, the Ornstein–Uhlenbeck (OU) SDE

$$\mathrm{d}X_t = \tau(\mu - X_t)\,\mathrm{d}t + \sigma\,\mathrm{d}W_t$$

is observed at regular times $t_r = r\,\Delta_t$ $(r = 1, \ldots, n)$. We write $x_r := X_{t_r}$ and $X = \{x_r\}_{r=1}^n$. The Markov transition is Gaussian with mean

$$m_\theta(x) := \mu + e^{-\tau\Delta_t}(x - \mu) = e^{-\tau\Delta_t}x + \left(1 - e^{-\tau\Delta_t}\right)\mu$$

and variance $v_\theta := \mathrm{Var}(x_r \mid x_{r-1}, \theta) = \sigma^2 \frac{1-e^{-2\tau\Delta_t}}{2\tau}$. For any path $x_{1:n}$, define $\ell_n(\theta) := \log \mathrm{p}_n(X \mid \theta)$.

Recall $\theta = (\mu, \tau)$ with $\tau \in [\overline{\tau}, \underline{\tau}]$, discretization step $\Delta_t$, and

$$m_\theta(x) = \mu + \rho_\tau(x - \mu) = \rho_\tau x + (1 - \rho_\tau)\mu, \qquad v_\theta = \sigma^2\frac{1 - \rho_\tau^2}{2\tau}, \qquad \rho_\tau := e^{-\tau\Delta_t}.$$

Fix a reference $\theta_0 = (\mu_0, \tau_0)$, write $m_0 := m_{\theta_0}$, $v_0 := v_{\theta_0}$, and let $X = (x_1, \ldots, x_n)$ with $x_r$ the OU samples at times $r\Delta_t$. The one–step densities are Gaussian, hence

$$\log\frac{\mathrm{p}_n(X \mid \theta)}{\mathrm{p}_n(X \mid \theta_0)} = \sum_{r=1}^n \left\{ -\frac{1}{2}\log\frac{v_\theta}{v_0} - \frac{(x_r - m_\theta(x_{r-1}))^2}{2v_\theta} + \frac{(x_r - m_0(x_{r-1}))^2}{2v_0} \right\}. \tag{F.30}$$

Let us begin with the tail behavior. Each one-step innovation $x_r - m_\theta(x_{r-1})$ is Gaussian with variance $v_\theta$ and

$$0 < v_{\min} \leq v_\theta \leq v_{\max} < \infty, \quad v_{\min} := \sigma^2\frac{1 - e^{-2\underline{\tau}\Delta_t}}{2\underline{\tau}}, \quad v_{\max} := \sigma^2\frac{1 - e^{-2\overline{\tau}\Delta_t}}{2\overline{\tau}}.$$

Moreover, if $x_{r-1}$ satisfies $|x_{r-1}| \leq R$, then $m_\theta(x_{r-1})$ also satisfies $|m_\theta(x_{r-1})| \leq \rho_{\underline{\tau}}R + (1 - \rho_{\underline{\tau}})|\mu|$. Hence, there exists $c > 0$ depending only on $(\Delta_t, \overline{\tau}, \underline{\tau}, \sigma)$ and the law of $x_0$ such that, for all $R \geq 1$,

$$\mathbb{P}\left(\exists r \leq n, |x_r| \geq R\right) \tag{F.31}$$

$$\leq \mathbb{P}\left(\exists r \leq n, |x_r - m_\theta(x_{r-1})| \geq (1 - \rho_{\underline{\tau}})R - |\mu|\right) \tag{F.32}$$

$$\leq \mathrm{poly}(n)\,e^{-cR^2} \leq \mathrm{poly}(n)\,e^{-R}, \tag{F.33}$$

for $R$ large enough compared to $|\mu|$.

Let us continue with the tail condition on the likelihood. We have the bound

$$\left| \sum_{r=1}^n -\tfrac{1}{2}\log\tfrac{v_\theta}{v_0} \right| \leq \tfrac{n}{2} \log\frac{v_{\max}}{v_{\min}} =: C_{\mathrm{var}}\,n. \tag{F.34}$$

For each $r$, abbreviate $m := m_\theta(x_{r-1})$ and $m_0 := m_0(x_{r-1})$. Using $v_\theta \geq v_{\min}$ and $v_0 \geq v_{\min}$,

$$-\frac{(x_r - m)^2}{2v_\theta} + \frac{(x_r - m_0)^2}{2v_0} \leq \frac{1}{2v_{\min}}\left((x_r - m_0)^2 - (x_r - m)^2\right).$$

Expanding the square,

$$(x_r - m_0)^2 - (x_r - m)^2 = -\left(m - m_0\right)^2 + 2\left(x_r - m_0\right)\left(m - m_0\right).$$

Summing over $r$ and applying Cauchy–Schwarz,

$$\sum_{r=1}^{n}\left(-\frac{(x_r - m)^2}{2v_\theta} + \frac{(x_r - m_0)^2}{2v_0}\right) \leq -\frac{1}{2v_{\min}}\sum_{r=1}^{n}\Delta_r^2 + \frac{1}{v_{\min}}\left(\sum_{r=1}^{n}(x_r - m_0)^2\right)^{1/2}\left(\sum_{r=1}^{n}\Delta_r^2\right)^{1/2}, \tag{F.35}$$

where $\delta_r := m_\theta(x_{r-1}) - m_0(x_{r-1})$.

On events where $|x_{1:n}| \leq R$, we have the conditions

$$c\|\mu - \mu_0\| - C(1 + R)|\delta_r| \leq L(1 + R)\|\theta - \theta_0\|,$$

for constants $c, C, L$ depending only on $(\bar\tau, \underline\tau, \Delta_t)$. Therefore, for $\|\mu - \mu_0\|$ larger than a constant multiple of $(1 + R)$, we have

$$\sum_{r=1}^{n}\delta_r^2 \geq nc\|\mu - \mu_0\|^2 \quad \text{and} \quad \left(\sum_{r=1}^{n}\delta_r^2\right)^{1/2} \leq \sqrt{n}\,C(1 + R)\|\theta - \theta_0\|, \tag{F.36}$$

for constants $c, C$ depending only on $(\bar\tau, \underline\tau, \Delta_t)$.

Combining (F.34), (F.35), and (F.36),

$$\log\frac{p_n(X \mid \theta)}{p_n(X \mid \theta_0)} \leq Cn - cn\|\mu - \mu_0\|^2 + \left(\sum_{r=1}^{n}(x_r - m_0(x_{r-1}))^2\right)^{1/2}\sqrt{n}C(1 + R)\|\theta - \theta_0\|,, \tag{F.37}$$

for constants $c, C$ depending only on $(\bar\tau, \underline\tau, \Delta_t)$.

Fix $R \geq 1$ and assume that $|x_{1:n}| \leq R$: we have shown that it holds with probability at least $1 - \text{poly}(n)e^{-cR^2}$.

In that case, $\left(\sum_{r=1}^{n}(x_r - m_0(x_{r-1}))^2\right)^{1/2}$ in (F.37) is bounded $\mathcal{O}(\sqrt{n}R)$ so the RHS can be made negative for all sufficiently large $\|\theta\|$: more precisely, it is negative for $\|\theta\| \geq R'$ with $R' \geq C(1 + R)^2$ for a constant $C$ depending only on $(\bar\tau, \underline\tau, \Delta_t)$. Since the event we are considering holds with probability at least $1 - \text{poly}(n)e^{-cR^2}$, it means that it holds with probability at least $1 - \text{poly}(n)e^{-R'}$. This proves the required tail bound with $R \leftarrow R'$.

Moving to the moment condition, by Gaussian moment bounds, (F.30) readily implies

$$\mathbb{E}\left[\log^2\sup_\theta\frac{p_n(X \mid \theta)}{p_n(X \mid \theta_0)}\right] \leq Cn^2 = \text{poly}(n),$$

which verifies the likelihood-ratio moment condition in Asm. 10.

Finally, we show the local regularity condition. For fixed $x_{1:r-1}$, the conditional density is

$$\log p_r(x_r \mid x_{1:r-1}, \theta) = -\tfrac{1}{2}\log(2\pi v_\theta) - \frac{(x_r - m_\theta(x_{r-1}))^2}{2v_\theta}.$$

On sets where $|x_{1:r}| \leq R$, $\|\theta\| \leq R$ (so $\mu, \tau$ bounded) and with $\tau \in [\bar\tau, \underline\tau]$, the maps

$$\theta \mapsto m_\theta(x_{r-1}) = e^{-\tau\Delta_t}x_{r-1} + \left(1 - e^{-\tau\Delta_t}\right)\mu, \qquad \theta \mapsto v_\theta = \sigma^2\frac{1 - e^{-2\tau\Delta_t}}{2\tau}$$

are smooth with bounded first derivatives: $|\partial_\mu m_\theta| \leq 1$, $|\partial_\tau m_\theta| \leq C_R$, $|\partial_\tau v_\theta| \leq C$, $\partial_\mu v_\theta = 0$. Since $x_r - m_\theta(x_{r-1})$ is also bounded by a constant multiple of $R$ on these sets, we obtain, for all $\theta, \theta'$ with $\|\theta\|, \|\theta'\| \leq R$,

$$\sup_{\substack{|x_{1:r}| \leq R \\ \|\theta\|, \|\theta'\| \leq R}}\left|\log\frac{p_r(x_r \mid x_{1:r-1}, \theta)}{p_r(x_r \mid x_{1:r-1}, \theta')}\right| \leq \text{poly}(R)\|\theta - \theta'\|.$$

