# OpenReview forum: "How Does the Pretraining Distribution Shape In-Context Learning? A Fundamental Trade-Off"
_ICML.cc/2026/Conference — ICML 2026 regular_

### Official Review · Reviewer_g4nn · 2026-03-10

**Soundness:** 3
**Presentation:** 3
**Significance:** 2
**Originality:** 3
**Overall Recommendation:** 5
**Confidence:** 3

**Summary:**

This paper studies how the statistical properties of the pretraining distribution shape in-context learning and decomposes develops a theoretical framework that explicitly accounts for heavy-tailed priors and dependent sequences. The main conclusion is a trade-off: heavier-tailed pretraining distributions can improve task selection robustness under distribution shift, but they worsen generalization when the number of pretraining tasks is limited. The paper supports this claim with controlled experiments on numerical tasks, including stochastic differential equations and processes with memory.

**Compliance With Llm Reviewing Policy:**

Affirmed.

**Final Justification:**

Thanks for the author responses and they have solved my comments well. I have modified the final score to "accept".

**Key Questions For Authors:**

1. The task-selection result appears to rely on a Bayesian-optimal view of a sufficiently powerful trained model. How should readers interpret this result for actual trained transformers, rather than for the idealized Bayesian regime?

2. The empirical study is restricted to controlled numerical tasks. Do the authors have additional evidence that the same trade-off appears in a more realistic sequence-modeling or language setting?

3. Which assumptions in the generalization analysis are most essential for the heavy-tail/generalization trade-off, and which are mainly proof conveniences?

4. The experiments mainly use prediction error under task shift to support the task-selection theory. Can the authors provide a more direct empirical proxy for task retrieval or posterior concentration?

5. The paper argues that heavy-tailed priors can help robustness under distribution shift. Could the authors clarify what concrete aspects of real pretraining data or task mixtures correspond to “tail heaviness” in practice?

6. Table 1 positions the work against prior theory papers. It would be helpful if the authors could state even more explicitly, in the main text, what new conclusions become possible only because of the heavy-tailed and dependent-sequence setting.

**Limitations:**

The paper should more clearly note that its empirical support is restricted to synthetic numerical tasks and that its conclusions about LLM pretraining are suggestive rather than directly established by text-domain experiments.

**Strengths And Weaknesses:**

Strengths: The paper addresses an important question for ICL: not just whether pretraining matters, but which properties of the pretraining distribution matter and why. The paper is technically ambitious in extending the analysis to heavy-tailed priors and dependent sequences, and it positions itself carefully relative to prior theory work. The resulting trade-off is meaningful and is supported by the controlled experiments presented in the paper.


Weaknesses: The paper’s strongest conclusions are established in an idealized theoretical/synthetic setting rather than in realistic language-model pretraining settings. In particular, the task-selection analysis relies on a Bayesian-optimal interpretation of sufficiently powerful trained models, which leaves some gap to actual transformer behavior. In addition, the experiments are limited to controlled numerical tasks, so the practical implications for real LLM pretraining remain suggestive rather than directly demonstrated. Finally, the theory is somewhat assumption-heavy, and the paper could do more to clarify which assumptions are essential to the main insight and which are mainly technical.

---

> ### Author Rebuttal · Authors · 2026-03-30
>
> Thank you for your detailed feedback! We answer your questions below and will integrate these clarifications in the revision.
>
> **Q1:**  The Bayesian-optimal view is by now a standard and well-supported lens in the ICL literature: several works have shown empirically that sufficiently expressive trained transformers closely track the Bayes-optimal predictor (Chan et al., 2022; Raventós et al., 2023; Wurgaft et al., 2025; Nguyen \& Reddy, 2025; Park et al., 2025). We therefore interpret our task-selection result as characterizing the regime that actual trained transformers are expected to approach when they are sufficiently powerful and well trained, rather than as a statement about an entirely disconnected idealization. Importantly, our empirical study is conducted with actual trained transformers, and the qualitative trends predicted by the theory are borne out in those experiments. We will revise the main text to make this interpretation more explicit.
>
> **Q2:**
> Some earlier works (Chan et al., 2022; Singh et al., 2023) have already observed some of our predictions in vision tasks ("heavier-tailed pretraining distributions can improve ICL performance only up to a point, beyond which performance
> degrades"). We agree however that studying this phenomena with natural language tasks with large language models is an important topic and it is a natural follow-up to our work.
>
> **Q3:**  Thank you for suggesting this improvement to the presentation, we will make this clear. For the generalization result Thm. 1, the crucial assumption to quantify the heavy tail character of the pretraining distribution $\pi$ is Asm.1: it defines $q$ the highest exponent such that $\mathbb{E}_{\theta \sim \pi}[\|\theta\|^q]$ is finite. Higher values  of $q$ indicate lighter tails. Asm. 2 is then needed for our predictions on the relationship between generalization error and temporal dependence. Asm. 3 concerns the regularity of the model and is mainly for technical purposes.
> For the task selection results, most assumptions are for technical reasons, the heavy tail character of $\pi$ is quantified through $\log 1/\pi(\theta^*)$ which appears in the bound.
>
> **Q4:** Thank you for this suggestion. We agree that a more direct empirical proxy for task retrieval or posterior concentration would be valuable. However, in our transformer ICL setup on numerical sequence tasks, these quantities are not directly observable: unlike in an explicit Bayesian model, the transformer does not expose a posterior over tasks or a canonical retrieved latent variable. We therefore evaluate the theory through the main quantity available: namely prediction error under task shift. We agree that this is an indirect rather than direct test of the task-selection mechanism, and we will make this limitation more explicit in the revision. Our claim is thus not that we directly measure retrieval or posterior concentration, but that the observed prediction behavior is consistent with the task-selection effect predicted by the theory. Exploring internal-state probes or other mechanism-level diagnostics would be an interesting direction for future work.
>
>
> **Q5:**
> Thank you for this suggestion. For numerical / tabular data, our conclusions about the tail heaviness of the pretraining distribution apply directly. A natural example is tabular foundation models [1,2], where one samples a class of functions and inputs from some distribution and then trains a large-scale model on this data. This clarifies how heavy tails can enter the pipeline: through the distribution used to sample pretraining tasks. Our work suggests that, in such settings, this choice can directly affect the robustness/generalization trade-off. In more realistic NLP pretraining settings, the natural analogue is a long-tailed mixture over latent tasks, domains, or input-output patterns, rather than a mixture concentrated almost entirely on highly frequent task types. We will revise the main text to make this interpretation more explicit.
>
> [1]: Hollmann, Noah, et al. "TabPFN." ICLR 23
>
> [2]: Qu, Jingang, et al. "TabICL" ICML 25
>
> **Q6**: Our contribution is not only to introduce a framework encompassing heavier-tailed priors and dependent task sequences, but to enable qualitatively new conclusions. In particular, our framework yields the three concrete predictions in \S3.4. It reveals a trade-off in pretraining distribution design for ICL: heavier tails can improve task selection at test time, especially under distribution shift, but can worsen generalization when the number of pretraining tasks is limited. This conclusion would not be accessible in a framework restricted to light-tailed distributions. Likewise, temporal dependence also has a concrete effect: stronger temporal dependence harms generalization, especially when the number of pretraining tasks is small. We will revise the main text to make these links more explicit.
>
> **Limitations**: We will add a limitation section discussing this, thank you!

---

> > ### Author Rebuttal · Reviewer_g4nn · 2026-04-03
> >
> > the authors have solved my comments well.

---

### Official Review · Reviewer_zjS1 · 2026-03-13

**Soundness:** 4
**Presentation:** 3
**Significance:** 4
**Originality:** 4
**Overall Recommendation:** 5
**Confidence:** 4

**Summary:**

This paper elucidates how the statistical properties of pre-trained distributions, including tail behavior, task coverage, and temporal dependence, affect the performance of ICL.
The authors propose a unified theoretical framework that decomposes ICL error into two distinct components: task selection and generalization.
They propose a fundamental trade-off. While heavy-tailed pre-trained distributions enhance the model's robustness and ability to retrieve tasks under distribution bias, they also reduce the efficiency of generalization samples when training data is limited.
These theoretical findings are validated through controlled experiments on complex numerical tasks, including linear regression and stochastic differential equations.

**Compliance With Llm Reviewing Policy:**

Affirmed.

**Final Justification:**

I keep my original score in support of the paper.

**Key Questions For Authors:**

see weakness

**Limitations:**

Yes

**Strengths And Weaknesses:**

Strengths
1. This paper makes a significant theoretical contribution by extending concentration inequalities to heavy-tailed priors and related sequences (non-independent and identically distributed data). This goes beyond the standard sub-Gaussian assumption prevalent in the literature and better fits the complex statistical structures found in real-world LLM pre-trained data.

2. This research clearly elucidates the trade-off between task recognition and generalization. The authors provide a principled explanation of how data management and distribution "coverage" affect the performance of ICL. They point out that heavy-tailed distributions are beneficial for task recognition but reduce label prediction efficiency.

3. The authors validated their theoretical analysis through numerical simulations, including the Ornstein-Uhlenbeck process and the Volterra equation. The theoretical error bounds agree well with empirical results for various distribution shifts and tail heaviness.


Weakness

1. While stochastic differential equations (SDEs) and linear regression can precisely control statistical variables, validation on large-scale natural language datasets is lacking.

2. This paper focuses almost entirely on sample/statistical efficiency. This tradeoff is described as a purely statistical tradeoff, but computational tradeoffs are equally crucial in practice.

---

> ### Author Rebuttal · Authors · 2026-03-30
>
> Thank you for your feedback and positive evaluation! We are grateful that you recognize that "This paper makes a significant theoretical contribution" and "clearly elucidates the trade-off between task recognition and generalization".
>
>
> We reply to your comments below.
>
> **Validation on NLP dataset**
>
> Some earlier works (Chan et al., 2022; Singh et al., 2023) have already observed some of our predictions in vision tasks ("heavier-tailed pretraining distributions can improve ICL performance only up to a point, beyond which performance
> degrades"). We agree however that studying this phenomena with natural language tasks with large language models is an important topic and it is a natural follow-up to our work. We will mention this discussion in the revision.
>
> **Computational trade-offs**
>
> Thank you for suggesting this important perspective. We agree that computational tradeoffs are crucial in practice, while our paper focuses primarily on the statistical tradeoff between task identification and generalization. That said, our results do suggest an indirect computational implication: if a pretraining distribution enables faster task identification, then fewer in-context examples may be needed at test time, which reduces the required context length. Since attention cost grows superlinearly with context length, this can translate into meaningful computational savings. We will clarify this scope in the revision and mention the combined study of statistical and computational tradeoffs as an interesting direction for future work.
>
> Thank you again for your input, and please let us know if you have any further questions.

---

> > ### Author Rebuttal · Reviewer_zjS1 · 2026-04-01
> >
> > Thank you for the clarifications. This work is timely and well executed.

---

### Official Review · Reviewer_o1CY · 2026-03-13

**Soundness:** 4
**Presentation:** 3
**Significance:** 3
**Originality:** 3
**Overall Recommendation:** 5
**Confidence:** 4

**Summary:**

This paper develops a theoretical framework under which to analyse the impact of the properties of the pretraining distribution and data sample number on the performance of in context learning's generalisation and task inference abilities. They extend existing results on generalisation bounds to heavy tailed distributions and sequences beyond iid and Markovian data (albeit with weak restrictions on their distributions) allowing for better reflection of the conditions of pretraining data in LLMs than more restrictive models. The paper proves two key theorems and summarises the implications of them in 3 intuitive takeaways: Heavier-tailed pretraining task distributions lead to better ICL performance under distribution shift (i.e. better coverage of prior), the generalization penalty of heavier-tailed pretraining distributions becomes significant when the number of pretraining tasks is small, and stronger temporal dependence harms generalization, especially when the number of pretraining tasks decreases. The authors validate their findings empirically on toy numerical tasks.

**Compliance With Llm Reviewing Policy:**

Affirmed.

**Final Justification:**

I keep my original score in support of the paper. The authors clarified questions I had during the rebuttal.

**Key Questions For Authors:**

- Are there any analysis/experiment combinations that can highlight the benefit of this framework (with its generality in sequence distribution and data structure) over other models without these generalities?

**Limitations:**

Yes

**Strengths And Weaknesses:**

**Strengths**
- The authors contextualise their work well and place it amongst the existing literature.
- Their extension of generalisation bounds to heavy tailed priors of (weakly restricted) sequence distributions allow for general results and is a useful contribution given the closer setup to LLM training in practice.
- They produce theorems that give clear interesting insights: 'Heavier-tailed priors and stronger temporal dependences increase the number of tasks required for reliable ICL generalization' and 'the ability to infer a new tasks is heavily dependent on the mass assigned to that task in the prior'.
- The theoretical framework and key assumptions are clearly stated and they are self-contained. The theorems and their implications are presented in a clear manner with a summary in 3.4.
-The lens of analysing ICL performance as a trade-off between task identification and generalisation provides a good explanation of previously observed phenomena and could be beneficial to the community beyond learning theory.


**Weaknesses**
- The uniform sub-Gaussian assumption is strong and, although explicitly stated, could have restrictions in the kinds of processes that can be modelled. It would be useful to address the strength of these assumptions in a limitations section. The sub-Gaussian assumption fail in some models, e.g. in models with parameter dependent noise.
- The paper would benefit from better self referencing to the appendix, for example, when discussing the kernel exponents (which is not defined in the main body).

---

> ### Author Rebuttal · Authors · 2026-03-30
>
> Thank you for your detailed input and positive evaluation. We are grateful that you find that we provide “theorems that give clear interesting insights”. We reply to your questions and comments below.
>
> **Uniform sub-Gaussian assumption**
>
> Thank you for raising this point. We chose this simplifying assumption so that we could focus on highlighting the impact of the properties of $\pi$ on the ICL error, and we will discuss it in a new “Limitations” section. Also note that our analysis can be extended to cover the case where the constant in the sub-Gaussian assumption grows with $\\|\theta\\|$.
> At the cost of a more intricate proof and additional constants, this yields the same theorem as Thm. 1. We will mention this as a remark while the general case is a subject of future work.
>
> **Self-referencing to the appendix and kernel exponents**
>
> Thank you for raising this issue. We will make sure the appendix is adequately referenced and add the definition of the kernel exponent, i.e., the exponent $\alpha$ in Section 4.3, to the main text to make it self-contained.
>
> **Benefits of our framework**
>
> Thank you for this question.
> Yes: one benefit of our framework (heavy-tailed distributions, temporal dependence) is that it is a better model of natural language than previous works restricted to light-tailed priors and i.i.d. or Markovian sequences. Natural language indeed displays strong temporal dependencies (Alabdulmohsin et al., 2024), and topic distributions tend to follow heavy-tailed laws (Zipf’s law; see [1]).
>
> In addition, our framework yields three concrete predictions (§3.4) that would not be accessible in more restrictive frameworks. It reveals a trade-off in pretraining distribution design for ICL: heavier tails can improve task selection at test time, especially under distribution shift, but can worsen generalization when the number of pretraining tasks is limited. This conclusion would not be accessible in frameworks restricted to light-tailed pretraining distributions. Likewise, the effect of temporal dependence cannot be captured in frameworks restricted to i.i.d. or Markovian pretraining sequences: our analysis shows that stronger temporal dependence harms generalization, especially when the number of pretraining tasks is small. We will revise the main text to make these links, and the benefit over more restrictive models, more explicit.
>
> Thank you again for your input, and please let us know if you have any further questions.
>
> [1] Piantadosi, Steven T. “Zipf’s word frequency law in natural language: A critical review and future directions.” *Psychonomic Bulletin & Review* 21.5 (2014).

---

> > ### Author Rebuttal · Reviewer_o1CY · 2026-04-02
> >
> > The reviewer's answer clarifies the questions I had.

---

### Official Review · Reviewer_kZo9 · 2026-03-18

**Soundness:** 3
**Presentation:** 3
**Significance:** 3
**Originality:** 2
**Overall Recommendation:** 4
**Confidence:** 4

**Summary:**

This work studies the role of the properties of pretraining distribution (in particular tail behaviour) on the ICL performance. They first study the generalization of the empirical bayes predictor trained on a fixed number of tasks $N$ sampled from the pretrain prior, and one sequence of length $T$ sampled from the respective task distributions under some assumptions on task and sequence dependence structure, and model regularity conditions like average lipschitzness wrt to inputs. Here, they show that heavier pre-train prior tails negatively affect the generalization error. Next, assuming a Bayes optimal predictor ($N \to \infty$) then quantify how the posterior task distribution saturates onto the true task (under some assumptions on data), therefore modelling “task selection in ICL” and show that heavy-tailed prior helps this.

**Compliance With Llm Reviewing Policy:**

Affirmed.

**Final Justification:**

During my initial review, I had concerns about whether the claimed tradeoff between generalization and task selection under heavy-tailed priors was an artifact of the two results operating in different regimes (finite vs infinite $N$), but the authors gave a fair response clarifying that this is not the case. The discussion on how this work connects to the commonly studied IWL-ICL regimes in prior literature was also fruitful, and the authors contextualized their results well. Overall, given the substantial existing work on IWL-ICL transitions, task diversity thresholds, and training dynamics in synthetic-theory ICL setups, I think this is a good paper that examines
a relatively understudied axis, formalizing the effect of the pretraining distribution on ICL performance, an effect that has been empirically observed in prior work.

One concern that I brought up in my initial review remains unresolved: prior work shows sub-linear $T^{-(1-\alpha)}$ context-length dependence, whereas Theorem 2 claims $1/T$ posterior concentration. The authors responded that Renyi divergence is necessary for their framework, but this does not explain the rate discrepancy with empirical observations. However, I believe this is a technical issue and the work has genuine strengths, so I increase my score from 2 to 4.

**Key Questions For Authors:**

See strengths and weaknesses.

**Limitations:**

yes

**Strengths And Weaknesses:**

**Strengths**

The work adopts a fairly general mixture-of-tasks framework (heavily studied in the ICL literature) and tries to theoretically understand the role of heavy-tailed priors in pretraining on ICL performance, amongst other things such as the correlation structure of samples within a sequence and the separation of the task distribution. The framework is fairly broad and encompasses common well-specified tasks such as linear regression and classification that are commonly used to study ICL, which is a strength.

While the effect of heavy-tailed priors on ICL performance has been empirically studied before, a theoretical study is new to the best of my knowledge, and is distinct from most theory works that aim to capture effects such as task diversity thresholds and training dynamics. The paper is overall well written.

**Weaknesses**:

My first concern is broad. This paper studies generalization error in a learning-theory sense, whereas much of the ICL literature, when training on a fixed set of tasks, measures two types of test performance: (i) on new sequences from training tasks, and (ii) on sequences from unseen tasks. The current framework is not suited to capture this distinction, and I believe this is because Part 1 assumes the predictor finds the empirical Bayes optimum regardless of how many sequences are sampled per task. In contrast, the IWL--ICL transition keeps model capacity fixed and increases $N$, and at some point when capacity is saturated the model falls into the ``learning mode.'' Additionally, these works with fiixed task set typically repeats task sequences, which promotes good learning of seen tasks, whereas the main part of this work assumes one sequence per task. Just prior to Takeaway 1, the paper briefly discusses this repetition; I would like to hear the authors' thoughts on whether the framework and results can be adapted to capture the IWL--ICL transition, and if not, why. In either case, a discussion of this connection should be included.


Second, the effect of the prior is captured in different ways in the generalization and task selection results. In the generalization error, the relevant quantity is a global parameter $q$ that captures how heavy-tailed the distribution is. In contrast, for task selection, it is the local prior term $\log(1/\pi(\theta^{\*}))$. I am not fully convinced by the tradeoff message: for generalization, heavy tails mean small $q$ and worse error, that is clear. But for task selection, first the dependence is logarithmic, which seems very weak, and second this specifically concerns tasks far from the bulk with small $\pi(\theta^{*})$, whereas for most tasks located in the bulk, task selection is unaffected.

I believe the two results are consistent precisely because they operate in different regimes: with limited data, a heavy-tailed prior means a large chunk of the distribution is poorly covered, making generalization hard; whereas with infinitely many tasks (the task selection part), identification is easier under heavier tails as the prior puts up less resistance, and needs less in-context evidence to concentrate the posterior on the true task.

This leads to my third concern: since the two results operate in separate regimes, it is difficult to reconcile them. For instance, if task selection were studied under finite $N$, it would capture the effect of posterior concentration as a function of both $N$ and $T$. Intuitively, with finite $N$, the empirical distribution over training tasks poorly approximates $\pi$ in the tail, and therefore task selection should be negatively affected by heavier tails rather than positively, the opposite of what Theorem 2 suggests, and the advertised tradeoff may not hold in the finite $N$ regime.



Connection to prior results Wurgaft et al. 2025, Bigelow et al. 2025: Prior work shows a sub-linear dependence on context length of the form $T^{(1-\alpha)}$ in a similar Bayesian inference framework, with sigmoidal growth (measuring correct predictions) with respect to $T^{1-\alpha}$, whereas Theorem 2 captures a $1/T$ rate. Since $1/T$ decays faster than $T^{-(1-\alpha)}$ for $\alpha \in (0,1)$, the bound appears better than empirical rates. Is this discrepancy because the paper tracks Renyi divergence, a different quantity from model prediction, and the translation between the two does not preserve the rate?

On average Lipschitzness: The paper assumes average Lipschitzness of the model wrt inputs over the sequence. There is substantial literature (Castin et al. 2023, Kim et al. 2021, see refs within these) on the Lipschitz properties of self-attention and how they typically degrade with $T$; however, those works typically study pointwise rather than average Lipschitzness. It'd be helpful to discuss this distinction and how the pointwise results affect average Lipschitzness.

Other questions/small points:

The dependence structure assumptions ($A_T, B_T$​) are stated abstractly in Wasserstein distance. How do they behave with $T$? Do they degrade? In general, I think it’d be helpful to give some worked out examples (in main text or appendix) on how the sample correlation $B_T$ and task influence $A_T$ varies for common ICL examples like regression and SDEs, and then use them in the gen. bounds as examples.

Line 214: says "we encompass non-independent and identically distributed (i.i.d)...."

**Refs**:

Wurgaft et al. 2025. In-context strategies emerge rationally.

Bigelow et al. 2025. Belief Dynamics Reveal the Dual Nature of In-Context Learning and Activation Steering

Castin et al. 2023. How smooth is attention?

Kim et al. 2021. The Lipschitz Constant of Self-Attention

---

> ### Author Rebuttal · Authors · 2026-03-30
>
> Thank you for your detailed feedback. We answer all your comments below and we will add these clarifications to the revision.
>
> **On the two types of ICL metrics**
>
>  The error (ii) you mention, ie the error of ICL on sequences from unseen tasks, is significantly more challenging than (i) and especially important for ICL.
> Our framework is thus designed to tackle this error and proceeds in two steps:
> - Thm.1 guarantees that the loss of the actual model wrt to the whole pretraining distribution π is small:
> this is equivalent to considering new tasks that are from the same pretraining
> distribution π but not from the training set. It also corresponds to the error (ii) in expectation. Note that the model $\hat f$ in this result is not the Bayes optimal predictor but
> a realistic model from the function class 𝓕
>  which minimizes the empirical risk (2). Crucially, to go further, guaranteeing that the population risk is low in turn ensures that the trained model $\hat f$ is actually close to the Bayes optimal predictor wrt π (3).
> - In §3.3, for task selection, Thm. 2 provides a guarantee for the Bayes optimal predictor on θ* a given new unseen task: this is precisely error (ii) you mention.
>
> **Repeating sequences for each task**
>
> Our work handles this case, see Thm. D.4 app. D.
>
> **IWL-ICL transition**
>
> Though it is not our focus, our work can provide some insights into this transition.
> Consider N samples $\theta_i$ from π.
> IWL corresponds to the Bayes optimal wrt the discrete distribution $\theta_i$ while ICL corresponds to the Bayes optimal wrt to the true distribution (see Raventos '23). A trained model will be closer to the ICL regime when it minimizes not only the training error but also the population loss. Our generalization guarantee can thus be seen as a guarantee as to when the model will enter the ICL regime: when the generalization error is low, it means that a trained model will be in the ICL regime.
>
> **Effect of the prior on task selection**
>
> First, even though the dependence is logarithmic in π, the heavy-tail character of π still heavily affects the bound.
> Consider a simple example: a 1D Student-t distribution π(θ) ∝ $1/\\|\theta\\|^{1+\nu}$
> The bound of Thm.2 thus depends on $$\frac{\log 1/\pi(\theta^\*)} {T} \sim \frac{\nu + 1}{T}  \log \\|\theta^\*\\|$$ so the bound depends linearly on ν/T: ν
>  thus directly influences the identification speed.  Moreover, our experiments clearly demonstrate this prediction.
>
> Second, we agree that the effect of the heavy-tail character of π mostly affects tasks far from the bulk, i.e. out-of-distribution tasks. We already highlight this point several times: see lines 287-291, right column;  l72-74;  l29-30 and §3.4:"Heavier-tailed pretraining distributions lead to better ICL performance under larger distribution shifts."
>
> **On the low vs infinite data regimes**
>
> Yes, ICL performance heavily depends on the number of tasks N. This is why the trade-off for heavy-tailed priors involves the data regime: heavy-tailed priors facilitate task selection, especially for large N, but are harmful to generalization, especially for small N.
>
> Finally, note that our framework can also encompass the finite N setting for task selection:
>
> **Task selection with finite N**
>
> We strongly disagree that studying task selection with finite N would eliminate the trade-off. When the test task θ* is kept fixed and outside of the bulk, heavier-tailed distributions always help for task identification: even if π is discrete, it is more likely that there are some samples near θ* if π is heavier-tailed.
> This conclusion is very well supported by our experiments (§4.1-4.2). and, more importantly, our framework can be used to formally analyze this situation.
> Consider a discrete pretraining distribution over N tasks $\theta_i$ from π. Thm. 1 can be applied (and the bound simplifies).
> For task selection, the more general version of Thm. 2 in the appendix, Thm. E.2, controls the error for a new unseen test task θ*: the leading error term is bounded by
> $$
> \min_{i}\\|\theta_i - \theta^\*\\|^2
> $$
> When θ* is far from the bulk, the expectation of this error is lower for π with heavier tails. For instance, for the 1D Student-t case, the expectation of this error is
> $$
> \frac{|\theta^\*|^{2 \nu + 2}}{N^2}
> $$
> Hence, when θ* is large, having heavier tails (small ν) is beneficial for task identification (with an exponential dependency).
>
> **Connection to prior results**
>
> Indeed, for our general framework to be possible, we need to consider the error in terms of Renyi divergence.
>
> **On average Lipschitzness**
>
> This assumption (see 229-237) is complementary to works on the pointwise smoothness of attention. Concretely, with L pointwise Lipschitz constant for contexts of length C, then the average Lipschitzness satisfies $L_T \leq \frac{CL}{T}$. Castin et al., Kim et al., can then be used to estimate L.
>
> **Coefficients A_T, B_T**
>
> For linear regression and SDE, they are in general bounded. For Volterra, see App. F.1.

---

> > ### Author Rebuttal · Reviewer_kZo9 · 2026-04-04
> >
> > Thanks to the authors for their detailed responses.
> >
> > On task selection with finite $N$ and whether it might kill the tradeoff: Fair point, noted. This was a major concern for me, the authors response makes sense.
> >
> > On the two types of metrics and ICL-IWL: Thanks for the comment. I have a follow up question here "if the gen. error is low the model is in ICL mode" however ICL mode implies ability to learn new tasks (OOD), and the paper says these there is a tradeoff between the two (gen. error vs ood task selection, which I view as learning a new task correct me if I'm wrong) under heavy tails. How do you reconcile this? Is it perhaps Raventos et al. 2023 (light tail), Park et al. 2025 (uniform) never consider heavy tails?
> >
> > Thanks for the other responses to my questions, I'm happy with all that. I am leaning towards increasing my score if you could resolve my final few questions.

---

> > > ### Author Response · Authors · 2026-04-05
> > >
> > > Thank you for your response and for this helpful comment.
> > >
> > > When we refer to the **ICL regime**, we follow the standard usage in the literature: the model behaves like the Bayes-optimal predictor with respect to the true pretraining distribution $\pi$ and can therefore generalize beyond the finite training set of tasks (e.g., Raventós et al.). However, this does **not** mean uniformly strong performance on all shifted or OOD tasks. Even within the ICL regime, performance can differ substantially between tasks in the bulk of $\pi$ and tasks far from it.
> > >
> > > This is precisely where our trade-off enters. Low generalization error means that the trained model is close to the Bayes-optimal predictor for $\pi$, i.e. that it is in the ICL regime rather than the IWL regime. However, the task selection under distribution shift is a separate question within that regime: a model may be Bayes-like with respect to $\pi$ and still perform poorly on shifted/OOD tasks. One of the aims of our work is precisely to understand how the shape of $\pi$, in particular its tail behavior, affects task selection under shift. This is one of the main takeaways in section 3.4.
> > >
> > > Thus, there is no contradiction: low generalization error concerns approximation to the population Bayes predictor, while our task-selection result concerns how that predictor behaves on a given test task $\theta^\star$, especially when $\theta^\star$ lies far from the bulk of $\pi$.
> > >
> > > Raventós et al. (2023) and Park et al. (2025) indeed do not vary the shape of the pretraining distribution (and Raventós et al. only consider tasks sampled from the pretraining distribution itself, rather than shifted/OOD tasks). If we consider figure 1 from our manuscript, their results would be concerned with the left side of the graph where the shift is minimal.
> > >
> > > We will add this clarification in the revision, and we hope this resolves your question.

---

### Decision · Program_Chairs · 2026-04-30

**Decision:**

Accept (regular)

**Comment:**

This paper establishes a theoretical framework to characterize how the statistical properties of the pretraining distribution, particularly tail behavior, shape ICL. It shows that heavy-tailed distributions can improve test-time task selection while harming generalization in limited-data regime during training. The technical results are solid, and the discussion of data distribution provides useful insights into ICL.

All reviewers agree that the paper is technically strong, novel, and provides meaningful theoretical contributions. They raised several concerns regarding clarification of certain results and assumptions, and the rebuttal effectively addressed these points. Reviewers updated their assessments and support acceptance.